**Year-round simulated methane emissions from a permafrost ecosystem in Northeast Siberia**

**Karel Castro-Morales[1*], Thomas Kleinen[2], Sonja Kaiser[1], Sönke Zaehle[1], Fanny Kittler[1], Min Jung Kwon[1§], Christian Beer[3,4] and Mathias Göckede[1]**

[1] Max Planck Institute for Biogeochemistry, Jena, Germany
[2] Max Planck Institute for Meteorology, Hamburg, Germany
[3] Department of Environmental Science and Analytical Chemistry, Stockholm University, Sweden
[4] Bolin Centre for Climate Research, Stockholm University, Stockholm, Sweden
[§] Present address: Korea Polar Research Institute, Incheon, Republic of Korea
*Correspondence to: Karel Castro-Morales (kcastro@bgc-jena.mpg.de)

**Abstract**

Wetlands of northern high latitudes are ecosystems highly vulnerable to climate change. Some degradation effects include soil hydrologic changes due to permafrost thaw, formation of deeper active layers, and rising topsoil temperatures that accelerate the degradation of permafrost carbon and increase in $CO_2$ and $CH_4$ emissions. In this work we present two years of modeled year-round $CH_4$ emissions to the atmosphere from a northeastern Siberian region in the Russian Far East. We use a revisited version of the process-based JSBACH-methane model that includes four $CH_4$ transport pathways: plant-mediated transport, ebullition and molecular diffusion in the presence or absence of snow. The gas is emitted through wetlands represented by grid cell inundated areas simulated with a TOPMODEL approach. The magnitude of the summertime modeled $CH_4$ emissions is comparable to ground-based $CH_4$ fluxes measured with the eddy covariance technique and flux chambers in the same area of study, whereas wintertime modeled values are underestimated by one order of magnitude. In an annual balance, the most important mechanism for transport of methane into the atmosphere is through plants (61 %). This is followed by ebullition (~35 %), while summertime molecular diffusion is negligible (0.02 %) compared to the diffusion through the snow during winter (~4 %). We investigate the relationship between temporal changes in the $CH_4$ fluxes, soil temperature, and soil moisture content. Our results highlight the heterogeneity in $CH_4$ emissions at landscape scale and suggest that further improvements to the representation of large-scale hydrological conditions in the model, will facilitate a more process-oriented land surface scheme and better simulate $CH_4$ emissions under climate change. This is especially necessary at regional scales in Arctic ecosystems influenced by permafrost thaw.

**Keywords:** methane, permafrost, carbon cycle, Arctic, wetlands, winter emissions.

## 1.    Introduction

During the last 30 years, atmospheric temperatures at northern high-latitudes have risen more than the global average (Schuur et al., 2015; Serreze et al., 2000). In consequence, many permafrost areas in these regions have experienced expedited thawing rates in recent years. Permafrost in northern high-latitude ecosystems contains twice as much carbon as the current carbon pool in the atmosphere and about half of global soil organic carbon (Hugelius et al.,

2014; Tarnocai et al., 2009). About two-thirds of the terrestrial Arctic is classified as wetlands (Liljedahl et al., 2016; Hugelius et al., 2014) and permafrost underlies most of them. Wetlands globally contribute about 25 % of the total $CH_4$ emissions (using bottom-up approaches between 2003-2012) from natural sources into the atmosphere. Nearly 4 % of the total global emissions from top-down inversions correspond to emissions in latitudes > 60 °N

(Saunois and al., 2016). The degradation of freshly available carbon from permafrost thaw is expected to contribute strongly to a positive carbon-climate feedback in Arctic ecosystems (e.g. Beer, 2008). Changes in air temperature, surface topography and projected shifts in precipitation in Arctic tundra ecosystems (Kattsov and Walsh, 2000; Lawrence et al., 2015) influence the soil hydrologic regime in permafrost areas. Also, thawing permafrost will

induce changes in the surface wetness due to surface subsidence of ice-rich soils (Christensen et al., 2004; Helbig et al., 2017b). These changes will therefore also influence the magnitude of future emissions of $CO_2$ and $CH_4$ into the atmosphere from Arctic terrestrial ecosystems (Hugelius et al., 2014; Lawrence et al., 2015; Schuur et al., 2008). Drier soil columns will enhance methane oxidation and increase $CO_2$ emissions (Kittler et al., 2016; Kwon et al.,

2016; Lawrence et al., 2015; Liljedahl et al., 2016; Sturtevant et al., 2012), also leading to changes in plant community structure (Christensen et al., 2004; Kutzbach et al., 2004; Kwon et al., 2016). Thus, it is imperative to improve our understanding of the effects of climate change in permafrost wetlands, specifically their contribution to greenhouse gases into the atmosphere.

Freeze and thaw soil processes are critical mechanisms that modulate the seasonality of $CH_4$ emissions in permafrost ecosystems of the Arctic (Panikov and Dedysh, 2000). Most of the annual $CH_4$ emissions from Arctic wetlands take place during summer (growing season). In spring, the rising air and soil temperatures promote the melt of snow and ice in the soil, stimulating the microbial production of gas within the mostly anoxic active layer (i.e. the

surface soil layer that thaws during summer and freezes again during autumn). During this season, episodic releases of large amounts of $CH_4$ in the form of bursts have been evidenced in wetlands (e.g. Friborg et al., 1997; Song et al., 2012), peatlands (e.g. Tokida et al., 2007)

and lakes (e.g. Jammet et al., 2015) of northern high-latitudes. During late autumn, $CH_4$ emissions still take place when the active layer starts to freeze gradually from the top and ice begins to fill the soil pore spaces, i.e. the so-called zero curtain period. Through this period, the remaining $CH_4$ in the soil that was produced during the growing season or in the deeper warm soil layers is squeezed out of the soil. This remaining gas is emitted to the atmosphere via molecular diffusion, and via the "pressure pumping" phenomenon due to advection enhanced by wind (Bowling and Massman, 2011; Massman et al., 1997), through the forming layer of snow (Mastepanov et al., 2008, 2013; Zona et al., 2016). Previous studies have reported that $CH_4$ emissions during the cold season in Arctic tundra ecosystems account for up to 50 % of the total annual $CH_4$ flux released in the form of gas bursts (Mastepanov et al., 2008; Zona et al., 2016).

Soil and vegetation at northern high-latitudes remain covered by snow during most of the year (October to May). Snow is an effective thermal insulator between the soil and the atmosphere, and it is a porous medium that allows the diffusive exchange of gases. Only few observational efforts have previously been made to constrain gas fluxes through the snow in tundra and permafrost environments during the long and cold Arctic winter. $CH_4$ emissions have been measured using flux chambers and eddy covariance (EC) towers in various snow-covered areas, e.g. in boreal forest soils (Kim et al., 2007; Whalen and Reeburgh, 1992), boreal peat landscapes bogs and fens (Helbig et al., 2017b; Panikov and Dedysh, 2000; Rinne et al., 2007; Smagin and Shnyrev, 2015), and in subalpine soils (Mast et al., 1998; Wickland et al., 1999). Also, in the Alaskan tundra (Zona et al., 2016) and in the Zackenberg valley in northern Greenland (Mastepanov et al., 2008; Pirk et al., 2016). In boreal peat bogs of West Siberia, cold season $CH_4$ emissions contribute from 3.5 to 11 % of the annual $CH_4$ fluxes (Panikov and Dedysh, 2000). In other Arctic permafrost tundra ecosystems, however, winter $CH_4$ emissions were one to two orders of magnitude lower than the emissions during summer, and only accumulate in the snowpack in the presence of layers of ice blocking their exit route to the atmosphere (Pirk et al., 2016). Wickland et al. (1999) concluded that in snow-covered subalpine wetland soils, $CH_4$ fluxes accounted for 25 % of the annual fluxes, similarly to the recent results shown in a boreal peat landscape of northwestern Canada (Helbig et al., 2017b). However, there are still large uncertainties in cold season $CH_4$ emissions from wetlands and permafrost ecosystems of the Arctic tundra, particularly related to projected changes in vegetation phenology due to climate warming which might also lead to changes in snow cover, e.g. more shrubs will tend to hold more snow during winter (Blanc-Betes et al., 2016; Domine et al., 2015). A thicker snow layer will insulate more the

soil column during autumn and winter, preserving the heat of the active layer after the preceding growing (zero curtain period) season. This will further impact the extent of subsequent wintertime $CH_4$ productions and emissions.

Numerical models have made much progress to better simulating the magnitude and temporal and spatial variability of $CH_4$ emissions in boreal regions. Methane models include the traditional theoretical and empirical approaches that describe the mechanistic understanding of the processes involved in the production, oxidation and transport of $CH_4$ in terrestrial ecosystems (e.g. Grant, 1998; Riley et al., 2011; Walter and Heimann, 2000). Previous

studies have also improved the scaling representation from plot to regional areas in specific locations, and also to global frameworks (Bohn et al., 2015; Lawrence et al., 2015; Melton et al., 2013; e.g. Riley et al., 2011; Ringeval et al., 2011; Tagesson et al., 2013; Wania et al., 2010). There are still many shortcomings in land surface models for boreal regions because a complex network of processes and a wide range of spatial and seasonal variation characterize

these areas. These are particularly related to the inability of accounting for methane emissions and uptake in dry non-wetland areas, capturing shifts in vegetation cover (e.g. Chen et al., 2015), representing soil thawing and freezing cycles due to a poor soil thermal physics representation (e.g. Schuldt et al., 2013; Zhu et al., 2014), varying wetland extents and water tables (Bohn et al., 2015), accounting for microtopography effects on surface water

and methane emissions dynamics (Cresto Aleina et al., 2013, 2016), upscaling to larger areas or for coupling to earth system models (e.g. Mi et al., 2014; Xu et al., 2015). Also, the majority of models lack a snow scheme that interacts with gas transport and a descriptive representation of peat soils (Xu et al., 2016). Special challenges exist for regional scale model simulations in wetland-dominated areas influenced by climate change, such as in Arctic

permafrost ecosystems, mostly because of the lack of observational constraints sufficient to understand the processes in these areas and to evaluate model outputs.

The aim of this work is to analyze the performance of an improved process-based methane model, designed for Arctic tundra and wetlands underlain by permafrost, when applied to a regional domain in Northeast Siberia. Our intention is to evaluate the potential of a refined

process-based methane model as a proof of concept, for its application to larger than site level scales. Also, a regional scale application will allow the identification of spatial heterogeneities in $CH_4$ emissions in boreal regions. To address these objectives, we simulate year-round $CH_4$ emissions during 2014 and 2015 with the process-based JSBACH-methane model (Kaiser et al., 2017) in a region dominated by low-lying wetland areas and continuous

permafrost in the Russian Far East. This model includes freeze and thaw soil cycles

associated with explicit methane production, oxidation and transport. The latter takes place through distinct pathways: plant-mediated, ebullition, and diffusion. In this work, we use an improved version of the model that explicitly simulates $CH_4$ emissions to the atmosphere in the presence of snow during the non-growing season, and also contains a revised representation of $CH_4$ transported by plants including the description of relevant features of vascular plants based on the volume of roots in the soil pore space. We present and analyze the year-round temporal variation of the $CH_4$ emissions and their relationship to the environmental controls at a regional (model domain) scale. The model performance was assessed by comparison of the simulated $CH_4$ emissions against year-round EC measurements and summertime chamber flux measurements in the same study area. Because temporal variation in the amount of inundated area is essential for the estimation of $CH_4$ emissions from wetlands (Prigent et al., 2007), our model also includes a representation of inundated areas using a TOPMODEL approach. We evaluate the modeled horizontal extent of the inundated areas against the wetland area from a high-resolution remote sensing product.

## 2. Methods

### 2.1. Site description

The target region of this study is located in Northeast Siberia, Sakha Republic. The model domain is centered on the town of Chersky and to the west is dominated by low-lying wetland areas of the Kolyma River floodplain and to the east by dry upland tundra (Fig. 1a). This is a region of continuous permafrost and active layer depths that range between 20 and 180 cm. Winter spans from October to May, with daily air temperatures that remain well below the freezing point and average daily temperatures of about 13 °C during July (Dutta et al., 2006). In this region prevail dry climate conditions, with a mean annual precipitation of 218 mm (60 % as snow and 40 % as rain; Dutta et al., 2006). At the Kolyma River floodplain, the soil profile has a top layer of organic material (~15-25 cm thick) that is located above alluvial mineral soils, i.e. silty clay (Kittler et al., 2016; Kwon et al., 2016). In this area, the vegetation is heterogeneous and representative of wet tussock tundra. There, the water-logged areas are covered by the tussock-forming sedges (*Carex appendiculata* and *Carex lugens*), and cotton grasses (*Eriophorium angunstifolium*) (Kwon et al., 2016). During the spring snowmelt (May and June), large sections of the Kolyma floodplain usually become inundated, and during summer, the extent of surface water recedes due to evapotranspiration and drainage to the river channels located nearby. However, most areas remain inundated throughout the year (Kwon et al., 2016) and microtopographic structures typical of polygonal

tundra landscapes are sparse in this region. The eastern part of the model domain has more elevated slopes and drier soils with tundra vegetation dominated by grasslands and forests, i.e. dwarf evergreen and deciduous shrubs, *Sphagnum* mosses, and lichens, and few trees (Dutta et al., 2006; Merbold et al., 2009). Loess soil deposits originating from the accumulation of aeolian and alluvial sediments characterize the soil in this region.

## 2.2.   Model configuration

The model results presented in this work were obtained with a regional configuration in offline mode of the land-surface component of the MPI-ESM (Max Planck Institute for Meteorology Earth System Model), the so-called Jena Scheme for Biosphere Atmosphere Coupling in Hamburg (JSBACH) model. We used a JSBACH version that has been extended

from the version of the CMIP5 activity (e.g. Brovkin et al., 2013; e.g. Raddatz et al., 2007; Reick et al., 2013). Modifications include the addition of a multilayer hydrology scheme (Hagemann and Stacke, 2015) and the representation of permafrost physical processes (Ekici et al., 2014). The model domain covers an area of 7 degrees in longitude (158° E to 165° E) and 3 degrees in latitude (66.5° N to 69.5° N). Using a horizontal resolution of 0.5° (Fig. 1b),

this results in a model domain with 14×6 equally spaced grid cells. The vertical structure in the model domain comprises 11 non-equidistant soil layers with thicknesses that increase from 6.5 cm at the top to 23.2 m at the bottom, reaching a maximum column depth of 40.5 m. This vertical refinement is necessary to achieve numerically stable solutions for the gas diffusion equation. In the model domain, the root zone is confined to the top five layers

(maximum depth of 1.1 m) with maximum and mean root depths of 0.88 m and 0.42 m respectively. The soil ice content is restricted to the top six layers (maximum depth of 2.0 m), with bedrock located from the 6[th] layer downwards.

In JSBACH, each grid cell has defined fractions for different types of vegetation that are assigned across a maximum of 11 non-equal tiles, hence a hospitable fraction to vegetation,

that represents the sub-grid scale heterogeneity of vegetation cover. The remaining fraction of the grid cell where vegetation is not assigned, is then associated to a land cover type that represents areas inhospitable to vegetation such as rocky surfaces and deserts (Reick et al., 2013).

In our model domain, only four land cover types were present (ordered by dominance in the

model domain): 1) C3 grasses, 2) deciduous trees, 3) evergreen trees and 4) deciduous shrubs (see Fig. S1 for the spatial distribution of the cover types in the model domain).

The model configuration contains the basic JSBACH modules with components from the Biosphere-Energy-Transfer-Hydrology model, BETHY (Knorr, 2000). The vegetation carbon

is categorized into three groups: wood, green, and reserve. The soil carbon and decomposition model Yasso07 (Tuomi et al., 2009, 2011) takes care of the transport and decomposition of carbon into the soil. It simulates the breakdown of litter and soil organic matter based on measurements of soil carbon and litterbag experiments, and has been previously implemented into JSBACH (Goll et al., 2015; Thum et al., 2011). In Yasso07, soil litter is divided into three classes: non-woody, woody, and humus. The non-woody class is subdivided into four pools representing groups of chemical compounds with an independent decomposition rate determined by changes in air temperature and precipitation, thus it has no relation to plant species (Goll et al., 2015; Tuomi et al., 2009).

Most of the $CH_4$ emissions into the atmosphere from Arctic terrestrial ecosystems are from wetland areas, thus the representation of the wetland extent in $CH_4$ models is of relevance. We use a TOPMODEL (TOPographic MODEL) approach (Beven and Kirkby, 1979; Kleinen et al., 2012; Stocker et al., 2014) to determine the fraction of any grid cell that is inundated implying it has a water table at or above the soil surface. We obtain a grid cell mean water table position from the soil hydrology scheme (Hagemann and Stacke, 2015) determined from the saturation state of the soil layers: the lowest soil layer that is not completely frozen or completely saturated contains the grid cell mean water table, with the exact location within the layer given as the layer fraction that is saturated. Details on the TOPMODEL scheme in JSBACH are shown in section 1 of supplementary material.

The position of the local water table depth $z_i$ is used to define the grid cell wetland area (Eq. S1), i.e. the grid cell wetland area is defined where $z_i \geq 0$ and it is subject to a minimum CTI (compound topographic index) threshold value $\chi_{min\_cti}$ that limits the maximum possible areas that can be flooded following the approach of Stocker et al. (2014), with lower values leading to larger wetland areas. In this configuration, the constant prescribed value of $\chi_{min\_cti}$ and the exponential decay of transmissivity with depth $f$ (Eq. S1) are tunable parameters of the TOPMODEL module used to expand or reduce the fraction of inundated surface areas in a model grid cell. Within the inundated fraction of the grid cell, a constant relative soil moisture saturation of 0.95 is assumed. The decomposition of soil organic matter is reduced to 35 % of the aerobic decomposition in line with Wania et al. (2010).

The $CH_4$ production and emission processes in the model are tightly linked to the volumetric soil porosity to allocate gas transport (Kaiser et al., 2017). This model configuration contains a permafrost module to explicitly simulate soil freeze and thaw processes coupled to the hydrological and thermal regimes in the soil column (Ekici et al., 2014). This is a relevant

process in permafrost regions where changes in the soil ice content drive the seasonal changes in the volumetric soil pore space, and changes in the soil moisture ultimately determine whether the soil pores are filled with water or air.

### 2.2.1. Methane module

In this work, the JSBACH-methane configuration presented in Kaiser et al. (2017) underwent several modifications. Besides being coupled to TOPMODEL and the soil carbon Yasso07 components, the $CH_4$ module itself acquired several extensions: i) a refined description of plant-mediated transport, ii) allowance of gas transport via diffusion through the snow during the non-growing season, and iii) change in the order of transport processes. Details on each of these changes are listed in section 2 of the supplementary material.

In the process-based JSBACH-methane module, the equilibrium between the concentrations in free atmosphere, soil air, and soil moisture is assumed for the initialization of the methane and oxygen concentrations in the soil. During each model time step, $CH_4$ is produced in the soil column depending on the soil hydrological conditions (i.e. ice content and soil moisture), soil temperatures, soil pore space, and the available decomposed carbon. The fraction of $CH_4$ produced from the total carbon decomposition under anaerobic conditions for mineral soils ($f_{CH4anox}$) is prescribed as 0.5 (i.e. 50 % of the anaerobically decomposed carbon is used to produce $CH_4$). Since this setting is highly uncertain, the model response to a range of $f_{CH4anox}$ values is tested in sensitivity experiments as part of this work.T

The JSBACH-methane module contains two explicitly modeled $CH_4$ oxidation processes: bulk soil oxidation and rhizospheric oxidation of methane (plant oxidation). These oxidation pathways interact iteratively in the model with the methane transport processes, reducing the methane pool when oxidation takes place. Only part of the oxygen in the soil is available for methane oxidation, and this discrimination relates to the amount of carbon dioxide produced during heterotrophic respiration, which uses up to a maximum value of 40 % of the total oxygen content in the soil. An additional 10 % of the available oxygen is assumed to be unavailable because it is used in other processes such as respiration by microbes. This leads to only 50 % of the total oxygen in the soil to be available for $CH_4$ oxidation.

To facilitate the interaction between the $CH_4$ and TOPMODEL modules, the ice-free pores of the soil column are prescribed at a saturation level of 95 % in the fraction of the grid cell that was determined as inundated. This concept mimics the lateral distribution of water that creates water-logged conditions, depending on the topographic profile. However, the soil temperatures, ice content and available carbon for $CH_4$ production are not changed in the model during this process. Thus, $CH_4$ emissions from a grid cell happen under a combination

of soil temperatures, ice content, and available carbon decomposition characteristic of an unsaturated soil column on the one hand, and ice-free soil pores with soil moisture at 95 % saturation on the other. Ultimately, the methane production, oxidation, and transport processes only take place in the saturated portion of the grid cell (Fig. S2). The transport of

the gases to and from the atmosphere is distributed across four explicitly modeled transport processes: plant-mediated transport, ebullition, and molecular diffusion without snow and through the snow. The transport pathways follow a sequential order based on the expected priority with their efficiency based upon prevailing soil moisture content (set to constant 95 % saturation in the inundated areas) taking into account the ice-corrected volumetric soil

porosity, which in turn depends on the soil temperature.

The *plant-mediated transport* in the model only takes place in areas with C3 grasses and follows Fick's first law, including the diffusion of gas between the roots of plants and the surrounding soil pores. In wetland ecosystems, many plants have developed an efficient aerenchyma system that functions as a transport mechanism of gases between the atmosphere

and their roots. Plants need oxygen for metabolic processes and the root exodermis is an efficient barrier that keeps the oxygen inside the plant roots and, at the same time, slows down the diffusion of gas from the soil into the roots; thus, the gas flow is restricted by the thickness of the exodermis tissue. In the JSBACH-methane module, the root exodermis has a prescribed diffusivity value of 80 % of the total diffusivity of the same gas in water, for the

gas transport from soil into the plant. *Ebullition* takes place when excess gas that has not been dissolved in the available soil pore liquid water forms bubbles that are rapidly transported upwards from their source in the deep soil layers through the water and into the atmosphere, successfully bypassing the oxic areas in the soil. *Diffusion* is the molecular transfer of gas from high to low concentration gradients between soil layers and the atmosphere following

Fick's second law. In this model version, diffusion is now also allowed to take place through a layer of snow using a simplified formulation that does not take into account the enhanced advection of gas in the snowpack due to wind, i.e. pressure pumping.

Between the model time steps, the amount of gas is constant, whereas the gas concentrations change in relation to the varying ice-free pore space. Further details on how these schemes

are included in the model are shown in the supplementary material, and for more details the reader is also referred to Kaiser et al. (2017).

### 2.2.2. Experimental set up and sensitivity experiments

The model was forced with the daily reanalysis atmospheric data CRUNCEPv7 (The Climate Research Unit from University of East Anglia, analysis of the National Centers for

Environmental Prediction reanalysis atmospheric forcing version 7.0) from 1901-2015 with a spatial resolution of 0.5° (Viovy and Ciais, 2016). Prescribed annual means of atmospheric $CO_2$ values (https://www.esrl.noaa.gov/gmd/ccgg/trends/global.html) were also used to drive the model. The model was spun-up for 10,000 years of simulation by repeating cycles of atmospheric data from 1901-1930 (~330 cycles) to equilibrate the soil carbon pools and

ensure pre-industrial steady state (Chadburn et al., 2017; McGuire et al., 2016). The total carbon (woody, green and reserve) after spin up in the entire model domain showed little change over the last 500 years of the spin up period. The methane module was de-activated during this procedure. After that, simulations were initialized with reanalysis data from 1931 until 2015 (85 years). To allow equilibration of the soil carbon pools to the hydrology as well

as equilibration of $CH_4$, a model adjustment period of 850 years (10 cycles using the 85 years of reanalysis data) was added. After this period, the subsequent output of the model was stored and used for data analysis. In this simulation, we used prescribed reference values for parameters in the TOPMODEL and methane modules that represent the control simulation. A description of the most relevant prescribed parameters and variables in the control simulation

is outlined in Table 1.

    To evaluate the robustness of the model and identify the parameters to which the model is most sensitive, a set of sensitivity experiments was done following a cost efficient parameter-permutation approach (Saltelli et al., 2000). Six model parameters that are prescribed in the model and are involved in the newly modified code for this model version were selected.

These parameters are either not provided in published literature, the published values are largely uncertain due to the nature of method used to obtain these values, or the measured values cover a wide range of options characterizing different conditions in nature. The selected parameters are: $\chi_{min\_cti}$ for the evaluation of TOPMODEL, $d_r$ and $R_{ff}$ for the evaluation of plant-mediated transport, $h_{snow}$ and $\phi$ for evaluation of the transport via

diffusion through the snow, and the fraction of anoxic decomposed carbon that becomes $CH_4$ ($f_{CH4anox}$) for the evaluation of the methane production. For each parameter, reference values from the control simulation were decreased or increased for one parameter at a time, by a fixed value (shown together with the results in Table 2), resulting in a total of 12 independent sensitivity simulations.

The values for the parameters $\chi_{min\_cti}$, $d_r$ and $R_{ff}$ and $h_{snow}$ are highly uncertain. The first one is a parameter that is part of the TOPMODEL parameterization, whereas the rest are highly uncertain or absent in published literature, therefore we decided to choose extreme values

with respect to their values in the control simulation. The selected values for $\phi$ and $f_{CH4anox}$ were kept within ranges reported in the literature. The snow porosity is derived from measurements of snow and ice, and ultimately controls the amount of gas that can diffuse through the snow layer. Different snow densities lead to different snow porosities: 330 kg/m$^3$ ($\phi$=0.64) for wind packed snow), 263 kg/m$^3$ ($\phi$=0.71) for settled snow and 128 kg/m$^3$ ($\phi$=0.86) for fresh damp new snow. These values were tested to reflect the effect of gas diffusion through less to more porous snow layers. All $\phi$ values were calculated with $\rho_{ice}$ = 910 kg/m$^3$.

The parameter $f_{CH4anox}$ is highly uncertain in literature. In our model, a setting of $f_{CH4anox}$ = 1.0 would imply that all of the decomposed soil carbon would become CH$_4$ under anaerobic conditions. The value used in the reference simulation is 0.5. In the context of the sensitivity experiments, we decrease $f_{CH4anox}$ to 0.1 (i.e. 10 % of the decomposed carbon will become CH$_4$ and 90 % will be oxidized), and to 0.3 (i.e. 30 % of the decomposed carbon will become CH$_4$ and 70 % will be oxidized).

Each sensitivity simulation consisted of a re-initialization from the conditions in the control simulation from the last time step on 31 December 1999. This was to allow the model to adjust to the parameter change for 13 years before the year of result analysis (i.e. 2014). In order to keep consistency in the treatment of our simulations, the same re-initialization procedure was done for a reference simulation by re-initializing the control simulation from the restart conditions on 31 Dec 1999, as in the sensitivity experiments, but without changing any parameter (i.e. maintaining the same parameters as in the control simulation). The results from the sensitivity experiments were compared to the results from the reference simulation. The temporal resolution of all the model simulations is 30 min, with hourly output averaged for analysis into daily and monthly values.

### 2.3. Observational data

### 2.3.1. Wetland product

Methane emissions to the atmosphere in the model occur largely from areas with a water table at or above the surface. These fractions of "inundated" areas in each model grid cell represent the horizontal extent of wetlands (including lakes, peatlands, or temporally inundated areas). As described in Section 2.2, in our study the inundated fraction for each grid cell is estimated through the TOPMODEL approach. For evaluation, we compared the spatial distribution of the inundated areas per grid cell to the wetland extent remote sensing product from ENVISAT ASAR (European Space Agency's ENVISAT with an Advanced

Synthetic Aperture Radar operating in Wide Swath mode C-band). The ENVISAT ASAR WS-wetland product (EAWS) was tested for operational monitoring in northern Russia, where small-scale ponds and an overall high soil moisture level are common surface features (Reschke et al., 2012). The backscatter of the EAWS product for high latitudes has a higher spatial and temporal resolution (150 m and 2 to 3 days, respectively) than other commonly used wetland products (e.g. Prigent et al., 2007), which have spatial resolutions of the order of kilometers. Thus, the EAWS product is able to capture small water bodies like tundra ponds and wetland patches that remain almost unchanged throughout the year and are associated with permafrost areas. The spatial coverage of the EAWS product includes most of northern Russia and is subdivided into 10 mosaics, each with different coverage areas. It is freely available as GeoTIFF images, each representing a 10-days-mean in a wetland map during July and August in 2007 (i.e. 01-10 July, 11-20 July, 21-31 July, 01-10-August, 11-20 August, 21-31 August, all in 2007; Reschke et al., 2012).

For the evaluation of the modeled wetland extent, each 10-days-mean image of the EAWS product was mapped to the same grid of georeferenced rectangular cells of the JSBACH domain. From the total 84 model grid cells, 35 model grid cells fall into the area coverage of the EAWS images (Fig. 1b). The wetland fraction from the EAWS remote sensing product ($w_{rs}$) in percentage was calculated as the ratio of pixels flagged as wetland (ID = 1) to the total number of pixels contained in the grid cell. In the model, the spatial wetland fraction ($w_{mod}$) is represented as the fraction of the total grid cell area that is inundated (i.e. with a water table at or above the soil surface). To facilitate a direct comparison against $w_{rs}$, the $w_{mod}$ values from the control simulation were averaged to the same 10-days-mean in 2007 as the remote sensing data.

### 2.3.2. Chamber measurements

To evaluate the performance of the methane model, we compared the total modeled methane fluxes ($F_{mod}$) to the total methane fluxes measured with flux chambers ($F_{ch}$) in the Kolyma River floodplain (Fig. 1b, see also Kwon et al., 2016). In this study, chamber fluxes from an undisturbed control area were considered for model evaluation purposes. The chamber flux measurements were done during the early to mid-growing season (15 June to 20 August) in 2014. As additional ancillary variables, water table depth, vegetation cover, and soil temperature were also measured. For further details on the gas measurements, calculations, and discussion of the chamber flux results the reader is referred to Kwon et al. (2016).

The surface area of each chamber along the control transect is 0.36 m², therefore even all 10 chambers combined can only represent a very small fraction of the surface area of a single

model grid cell ($2.5 \times 10^9$ m$^2$). However, since both $F_{mod}$ and $F_{ch}$ are normalized to a unit area (CH$_4$/m$^2$/day), it is possible to directly compare $F_{ch}$ to $F_{mod}$. For the model evaluation exercise, we extracted the daily $F_{mod}$ corresponding to the same dates of the chambers flux measurements, and only the emissions from that model grid cell where the chamber plots were geographically positioned (grid cell A, Fig. 1b). We also show the results from an

adjacent grid cell (grid cell B, Fig. 1b) to demonstrate the spatial heterogeneity between the model grid cells for a region close to the chamber flux measurements. This specific 2$^{nd}$ grid cell was chosen to highlight the fact that even areas that appear similar in overall ecosystem structure can produce deviating CH$_4$ flux rates, for example influenced by environmental factors such as soil depth, inundation fractions or C3 grass coverage.

Due to the heterogeneous topographic characteristics in the study site, the microsites of the chamber plots within the control area include water-saturated (average water table during the growing season > 10 cm below the surface, observed in 8 chamber plots) and unsaturated characteristics (dry soil conditions, i.e. water table < 10 cm below the surface, observed in 2 chamber plots, Kwon et al., 2016).

Thus, the total $F_{ch}$ from the chamber plots was averaged separately for the wet plots ($F_{ch\_wet}$) and for the dry plots ($F_{ch\_dry}$). This heterogeneity in the data finds its equivalent in the model grid cell heterogeneity as estimated by TOPMODEL, where on average only a portion of the grid cell area is inundated and the rest remains dry during a specific period of time. The modeled methane emissions correspond exclusively to the portion of the grid cell with near-

water saturated soils. Similarly, the chamber flux measurements evidenced predominant emissions of this gas in plots with wet soils, whereas the emissions in dry plots were negligible. Thus, to obtain the total $F_{ch}$ the chamber flux measurements, and to account for emissions predominantly from wet soils, $F_{ch\_wet}$ and $F_{ch\_dry}$ were scaled to the daily-inundated fractions $w_{mod}$ for the corresponding model grid cell A. More details are presented in

Appendix A:

$$F_{ch} = F_{ch\_wet} \cdot w_{mod} + F_{ch\_dry} \cdot (1 - w_{mod}) \qquad (1)$$

At two of the chamber sites, temperature sensors (hereinafter referred to as redox systems) continuously recorded the soil temperature profile at three soil depths (4, 16, and 64 cm). The redox systems are located in a site dominated by dry soils and a site dominated by wet soils,

and thus these temperature measurements reflect the important influence of soil water levels on the soil thermal regime across the seasons.

### 2.3.3. Eddy covariance measurements

The model results were also compared to ecosystem-scale methane fluxes measured by an EC
tower situated in the Chersky floodplain near the north end of the chamber plot transect in a
control area (Tower 2 at 68.62° N and 161.35° E in Fig. 1 of Kittler et al., 2016). The
observation height is at 5.11 m.a.g.l., and fluxes are available at 30 min intervals. For details
on the instrumental setup, raw data collection, and EC data post-processing, the reader is
referred to Kittler et al. (2016; 2017). The analysis of uncertainties in the EC data is
presented in Appendix A. The field of view ("footprint area") of an EC system with the given
sensor height above the ground normally extends up to several hundred meters in the main
wind direction at any given time, changing with atmospheric turbulence conditions (Fig. 1 of
Kittler et al., 2016). The position of the EC tower falls within the area of model grid cell A
(shown in Fig. 1b) and far away from the grid cell borders; thus, it is assumed that all the $CH_4$
fluxes measured with the EC system fall within the area of grid cell A. To improve this
comparison due to the difference of spatial scale between the EC footprint and model grid
cell areas, for the former we analyzed the vegetation composition within the footprint using
highest resolution land cover maps based on WorldView-2 remote sensing imagery. For this
analysis, we aggregated vegetation classes to differentiate between areas of predominant wet
soils or wetlands (dominated by the cotton grass *Eriophorum angustifolium*) and dry soils
(dominated by shrubs and the tussock *Carex appendiculata*). We then compared the extent of
the wetlands to the inundated fractional area of the model grid cell considered as the
corresponding model wet area. It has been recently shown in the literature that the type of
vegetation in tundra landscapes is a good indicator of the spatial distribution and variation of
$CH_4$ fluxes (Davidson et al., 2017), and it is also expected that the majority of the $CH_4$ fluxes
are emitted from wetlands in tundra ecosystems (Helbig et al., 2017a). About 26 % of the
fluxes measured by the EC tower were emitted from wetland areas within the footprint, i.e.
from wet soils with cotton grasses. Within the entire model grid cell A, the inundated fraction
is between 17.7 % and 19.9 % (10-day mean values in summer months) during the summer
of 2014, while C3 grasses cover 33.3 % of the area (with no explicit separation between
cotton grasses and tussocks). To investigate the EC methane fluxes for a smaller wetland area
similar to that one in the model grid cell, it is possible to linearly scale the 10-day mean EC
methane fluxes to the inundated fraction from the model. Results of this scaling approach for
fluxes in summer 2014 are shown in Appendix A.

## 3. Results

### 3.1. Evaluation of inundated areas

Within the context of this analysis, fractions of inundation are given as the percentage of the total grid cell area that holds water at or above the surface. The first comparison between remote sensing ($w_{rs}$) and simulated ($w_{mod}$) wetland extents, using an initial TOPMODEL configuration, showed that the model mostly overestimated the extent of inundated fractions. For example, in the predominantly wet sections north of the model domain (> 68.5° N), the averaged $w_{rs}$ is 9 % whereas $w_{mod}$ was simulated at 15 %. However, in drier areas (< 68.5° N) $w_{rs}$ is on average 1.2 % whereas the model did not predict inundation in those grid cells. Since modeled methane emissions only take place in the inundated areas of a grid cell, it was necessary to modify the prescribed TOPMODEL parameters to improve $w_{mod}$ towards $w_{rs}$. To achieve this, the initially prescribed maximum threshold for inundation ($\chi_{min\_cti}$) was modified in a similar fashion than for the sensitivity experiments through a step change of the parameter value and subsequent analysis of results, until the horizontal extent of inundated areas in the model decreased compared to the results of the initial configuration. Changes in this value have an effect only on wet areas. In Fig. 2 the latitudinal distribution of the percent difference between $w_{mod}$ and $w_{rs}$ for 01-10 August 2007 after parameter adjustment (i.e. $\chi_{min\_cti}$=12) is depicted. We show only the results corresponding to one EAWS image because the results are similar for the other five available GeoTIFF images. The distribution of modeled grid cell inundated areas during the same period of time is shown in the inset of Fig. 2: model grid cells with more than 1 % of inundated area are found from the northwest to the southeast part of the model domain, and also include some grid cells in the western and northern parts. The spatial distribution of the modeled inundated areas throughout the year does not vary considerably because the inundated fraction in the model takes into account the accumulation of liquid and frozen water. However, the fraction of inundation within each grid cell varies in relation to drier or wetter conditions. After parameter adjustment through step change tests, the comparison between $w_{mod}$ and $w_{rs}$ resulted in a mean difference of −1±8 % and a median of 2 % integrated over all the six 10-days-mean periods. However, some outlier values result in considerable single-pixel differences between $w_{mod}$ and $w_{rs}$, ranging from +19 % in the southernmost areas (< 68.5° N) to −23 % in the northernmost areas (> 69° N). The best agreement between model and EAWS product is observed between latitudes 68.5° and 69° N (Fig. 2).

During the process of optimization between $w_{mod}$ and $w_{rs}$, the parameter $f$ was not modified because this would influence both the inundated and dry areas of a grid cell. The best value for $\chi_{min\_cti}$ that resulted in a closer agreement between $w_{mod}$ and $w_{rs}$ is applied in the

configuration of the control and reference simulations of this work. We exemplify the effect on the modeled methane emissions due to changes in the $\chi_{min\_cti}$ value, and include this parameter in the sensitivity experiments shown in the following section.

### 3.2. Sensitivity experiments

We investigated the impact of different values for selected model parameters (shown in column two of Table 2) on the individual transport processes and total $CH_4$ emissions. We compared the results from the reference simulation at daily resolution in 2014 to six pairs of sensitivity experiments (Fig. 3, with pairs of sensitivity experiments shown as panels within rows a through f). The annual mean model domain $CH_4$ emissions for each experiment are also summarized in Table 2. From all the sensitivity experiments, a significant difference in model output between the reference simulation and the simulations with modified settings was found only for parameters $\chi_{min\_cti}$ and $f_{CH4anox}$ (for both variables n=365 and $p<0.01$ after a nonparametric Wilcoxon rank-sum test; see also Fig. 3, all panels in row a and f).

The prescribed threshold parameter $\chi_{min\_cti}$ in the TOPMODEL module sets the maximum possible area in the grid cell that can be flooded. A higher $\chi_{min\_cti}$ value leads to a larger wetland extent in already inundated areas within the model grid cell. Consequently, our results show that a change in $\chi_{min\_cti}$ has a large effect on the $CH_4$ emissions: describing $\chi_{min\_cti} = 13$ leads to nearly 1.5 times higher $CH_4$ emissions during summer and autumn compared to the results using the reference value of 12, and about two times higher than the results with the lower $\chi_{min\_cti}$ test value of 11 (Fig. 3, row a). The effect of varying $\chi_{min\_cti}$ in the resulting model mean inundated fraction is shown in Fig. S3 of the supplementary material. With the higher $\chi_{min\_cti}$ value (i.e. 13), the annual average of the inundated fraction in the model domain (0.054) increases by 54 %, whereas with the lower $\chi_{min\_cti}$ value (i.e. 11) the annual average inundated fraction in the model domain (0.024) decreases by 35 %, both with respect to the annual average of the inundated fraction in the model domain from the reference simulation (0.0367).

The $f_{CH4anox}$ parameter is a prescribed fixed value used to define the fraction from the total decomposed soil organic matter that will be allocated for $CH_4$ production (i.e. anoxic carbon mineralization), with the rest becoming $CO_2$. As in many other land surface models, only mineral soils are considered in our model configuration (limitation further discussed below in Sect. 4.3). For a $f_{CH4anox}$ value of 0.5 (control) the resulting mean summertime $CH_4$ emissions in the model domain were five times higher than the emissions with lower $f_{CH4anox}$ values (0.1 or 0.3) (Fig. 3, row f).

The remaining parameters tested in this sensitivity test (Table 2) show that with the chosen values, the simulated $CH_4$ emissions are not significantly different from each other. The $d_r$ parameter associated with the plant-mediated transport pathway shows that the difference between the simulated $CH_4$ emissions is not statistically significant through the year between fine plant roots of 2 mm (as defined in the reference simulation) and thicker roots of 8 mm (Fig. 3, row b). A small variation can be noted for ebullition and diffusion mostly during July (Fig. 3, column 1 and 2 of row b). The difference in emissions due to an increase in the soil root volume from 20 % to 60 % is also not statistically significant (Fig. 3, row c).

For the selected parameters associated with the emissions of $CH_4$ through the snow, the porosity of snow $\phi = 0.64$ used in the reference simulation ($\rho_{snow}$ of 330 kg/m$^3$ for wind packed snow), has a mean tortuosity $\tau = 0.77$, calculated with Eq. S6 (suppl. material). The tortuosity value decreases with denser snow, thus for $\phi = 0.71$ corresponds a $\tau = 0.79$ ($\rho_{snow} =$ 263 kg/m$^3$ for aged settled snow), whereas for $\phi = 0.86$ means a $\tau = 0.85$ ($\rho_{snow}$ of 128 kg/m$^3$ for fresh damp new snow). Our sensitivity results from these experiments show that the differences between the winter $CH_4$ emissions through a layer of fresh damp snow, or through a wind packed snow layer, are not statistically significant (Fig. 3, row d).

Finally, the fixed limiting snow depth, which discriminates between ordinary $CH_4$ transport via diffusion and diffusion through the snow, was also tested. In the reference simulation, this switch happens at a fixed $h_{snow} \geq 5$ cm. In the sensitivity experiments, we decreased $h_{snow}$ to 3 and 1 cm. The results show that differences between the individual and total $CH_4$ emissions through various $h_{snow}$ values are not statistically significant (Fig. 3, row e). A time shift is seen however, in the $CH_4$ emissions from mid-October until mid-November (Fig. 3, column 4 of row e), with larger emissions through snow taking place earlier if $h_{snow}$ is thinner. Nevertheless, this temporal shift in the $CH_4$ emissions through the snow does not influence the total $CH_4$ emissions. The regionally aggregated $CH_4$ transport via ebullition, diffusion and plants during the same months is reduced as $h_{snow}$ becomes thinner, thus compensating for the shift in the emissions in the presence of snow and maintaining a mass balance in the annual total emissions.

### 3.3. Evaluation of modeled emissions with eddy covariance and chamber measurements

### 3.3.1. Evaluation of year-round modeled total $CH_4$ emissions

The methane emissions from EC measurements used here to evaluate the modeled $CH_4$ emissions at grid cell scale spans from April 2014 until September 2015, while data from

chamber measurements are restricted to the period of June to August 2014. The modeled total $CH_4$ emissions used for this comparison correspond to the grid cell where both the EC tower and the chambers are geographically located (grid cell A, Fig. 1b). Also, we show the modeled total $CH_4$ emissions from the neighboring grid cell to the west (grid cell B, Fig. 1b)

and further discuss their association with environmental variables and prescribed parameters in the model. The EC data presented here has been subject to a thorough quality check, and gap-filling was subsequently applied to produce a continuous time series (see Appendix A for details also on uncertainties analysis). For the year-round data evaluation it is not possible to apply the suggested linear scaling approach between the EC flux and model flux data, based

on the vegetation type as indicator of wetland areas in the EC footprint and the inundated fraction predicted in the model (section 2.3.3 and Appendix A). This is due to the lack of year-round vegetation coverage from remote sensing data that would otherwise allow obtaining a temporal varying wetness area for the EC footprint. However, the results shown in Appendix A from the suggested scaling approach for data in summer 2014 serve as a

demonstration that: 1) the areas with wet soils within the EC footprint and the model grid cell, translated into the areas where the majority of the $CH_4$ emissions take place, show only minor differences, and 2) the offset between methane fluxes from EC and from the model can be largely attributed to these differences in the extent of wetland areas. In the course of this manuscript we will consider the EC fluxes as representative for the processes within the

entire model grid cell, therefore allowing a direct comparison to the modeled $CH_4$ fluxes.

Despite the large spatial scale of the modeled emissions, the monthly mean of the $CH_4$ emissions from the EC and chamber measurements agree well with the monthly model results for the grid cells A and B and for 2014 and 2015. Positive correlations between the measured and modeled $CH_4$ fluxes, with correlation coefficients ($R^2$) higher than 0.95, are observed in

all comparisons except for the correlation between the chamber measurements and the results from grid cell B ($R^2$=0.85; Fig. 4). Fig.5a and b display box plots of the monthly mean $CH_4$ emissions for summer months (June, July, and August) from each data set: both grid cells (A and B), EC data for 2014 and 2015, and chamber flux measurements for 2014 only. In 2014, the median of the $CH_4$ emissions from grid cell B is consistently higher than the rest of the

other compared datasets, and this is followed by the EC fluxes (Fig. 5a). The same is observed in 2015 (except for the lack of chamber flux measurements during that year; Fig. 5b). During both years, the median of the modeled $CH_4$ emissions from grid cell A is generally lower than the rest of the compared data sets (Fig. 5a and b).

The time series of the monthly mean $CH_4$ emissions from the model grid cell A and B is

compared with the observational datasets in Fig. 5c. The shaded area around the mean values is one standard deviation calculated from the daily values; thus, it represents the range of variability in the emissions within each month. To analyze the contribution of uncertainties in the daily variability of the EC data, random (due to e.g. turbulent sampling or instrument error) and systematic (e.g. instrument calibration or drift) errors in this data set were assessed.

The uncertainties in the EC data are given as error bars in the monthly averages in Fig. 5c and account on average for $0.35 \pm 0.22$ mg $CH_4$ m$^{-2}$ d$^{-1}$ of the monthly emissions. These uncertainties are smaller than the spread of the daily variability. A summary of the methods followed to account for these errors is presented in Appendix A.

For the two years of analysis, modeled $CH_4$ emissions of grid cell A underestimate the EC

monthly values by $4.7 \pm 8.1$ mg $CH_4$ m$^{-2}$ d$^{-1}$. However, the modeled values from grid cell B are higher for most of 2014 and 2015 by as much as $6.1 \pm 10.5$ mg $CH_4$ m$^{-2}$ d$^{-1}$. During winter, the model $CH_4$ emissions from both grid cells are on average 3.7 mg $CH_4$ m$^{-2}$ d$^{-1}$ lower than the EC measurements (Fig. 5c). The modeled $CH_4$ emissions from both grid cells show large interannual variability. This is evidenced in Fig. 6, which compares the standard deviation of

the monthly fluxes between the two years of analysis. In the model results, particularly grid cell B shows large interannual variability in summer months (10.9 mg $CH_4$ m$^{-2}$ d$^{-1}$ in June and 5.6 mg $CH_4$ m$^{-2}$ d$^{-1}$ in July).

Methane emissions from the chamber flux measurements are lower than the model results of both grid cells for June and July 2014 (on average by 16.6 mg $CH_4$ m$^{-2}$ d$^{-1}$ in June and 24.3

mg $CH_4$ m$^{-2}$ d$^{-1}$ in July), but also than the EC flux data (on average 15.3 mg $CH_4$ m$^{-2}$ d$^{-1}$ and 25.1 mg $CH_4$ m$^{-2}$ d$^{-1}$ in June and July, respectively). However, the results from grid cell A are in closer agreement with the chamber flux measurements, with the chamber data showing larger emissions by 8.2 mg $CH_4$ m$^{-2}$ d$^{-1}$ compared to results from grid cell A during August 2014. The shaded areas of the EC data evidence the largest spread in daily variability of all of

the presented data sets, particularly in summer months. Thus, despite the disagreement between monthly mean values, there is an obvious overlap in the shaded areas between all data sets during 2014, while a larger disagreement is observed only during the summer of 2015 between results from grid cell B and EC data (Fig. 5c).

The root mean square error between the daily $CH_4$ fluxes from grid cell A and the

observations, normalized to the mean of the measurements (NRMSE = RMSE / mean $(CH_4)_{obs}$ x 100) is on average < 30 % from June to October for both years, while for spring and winter is on average 80 % with the maximum NRMSE during May in 2014 (107 %) and

2015 (104 %). Thus, the large variation of the measured daily fluxes in summer leads to a lower error when compared to the summertime modeled fluxes, whereas the lower magnitude and variation in wintertime fluxes leads to capture a larger error between modeled values and the observations.

### 3.3.2. Relationship between soil temperatures and $CH_4$ emissions

To examine the relationship between soil temperatures and $CH_4$ emissions, we first compared the modeled and measured soil temperature profiles. The temporal evolution of the vertical profiles of daily soil temperatures, measured with the redox systems, is shown in Fig. 7a for the wet plot and in Fig. 7b for the dry plot in 2015. The measured soil temperatures were only available from August to December in 2014, and behaved similarly in 2015 for the same months. The temperature values measured with the sensors at 4, 16, and 64 cm depth were linearly interpolated every 2 cm through the vertical soil column to construct the soil temperature profiles shown in Fig. 7a and b. For comparison, the modeled vertical profiles of the daily soil temperatures in 2015 for the top four soil layers (bottom depth of 3, 12, 29, and 58 cm) in grid cells A and B were also linearly interpolated every 2 cm (Fig. 7c and d). During winter and spring, the measured soil temperatures are not lower than −16 °C, while the modeled temperature values are as low as −26 °C within extended sections of the period from December to May. The measured values in the dry plot show abrupt temperature changes during the transition between freezing conditions (< 0 °C) and warmer conditions (> 0 °C) during mid-December and mid-May. This abrupt change is also seen in the wet plots with freezing conditions remaining for a shorter period of time; i.e. the change to and from warmer temperatures takes place only from the end of January until mid-May. Also, generally colder temperatures are observed in the top part of the soil column and gradually extend to deeper soil layers as the season progresses. In contrast, although the modeled soil temperatures reach lower values during winter, a smoother transition of temperature is evidenced from freezing to warmer conditions in spring, and to freezing conditions again in autumn. In the model results, the soil temperature remains homogeneous along the vertical profile.

Measured temperatures above freezing conditions occur from mid-June until the end of September. As summer progresses, warmer soil temperatures extend from the surface to deeper soil layers (Fig. 7a and b). In the dry plot, however, the warmer conditions remain only in the top 16 cm of the soil column (Fig. 7b) due to lower soil moisture content and lower thermal conductivity compared to the wet plot. The model is able to capture the timing

of the seasonal transition from spring to summer at the end of May, the duration of the summer conditions, and the magnitude of the temperature values. For grid cells A and B, the summer temperature profiles are more similar to the wet than to the dry plot. The average measured soil temperature in the range of the sensor's depths (top 64 cm) during summer (June, July, and August, 2015) in the dry plot was 2.1 °C, while in the wet plot for the same period the average measured soil temperature was 4.7 °C; in the model, the average soil temperature in the top 58 cm is about 4.9 °C during summer of 2015. The modeled warm soil temperatures (> 5 °C) reach deeper soil layers in summer; however, this is not observed in the measured data. This could simply be due to the coarse vertical resolution of the data because of the large gap between sensors (from 16 to 64 cm depth). Thus, to evaluate the extent of the warm soil temperatures depicted in the model, this portion of the soil column needs to be better resolved vertically by the measurements.

A larger disagreement between measured and modeled soil temperatures, however, occurs during the transition from autumn to winter. The measured temperatures remain around −3 °C in the top 64 cm from October to mid-December, until they change abruptly to around < −10 °C in the dry plot during mid December (Fig. 7b), and in the wet plot towards the end of January (Fig. 7a). In contrast, the model results show a gradual transition between the seasons, with decreasing soil temperatures to values < 0 °C starting in mid-October (Fig. 7c and d).

To investigate the effect on the $CH_4$ emissions due to the abrupt changes in the measured soil temperatures, we plotted the soil temperature at 12 cm (for model data) and at 16 cm (measured values for the wet plot) against the total $CH_4$ emissions for grid cells A and B and from EC measurements in 2014 (Fig. 8). The modeled soil temperatures represent the entire grid cell conditions whereas the $CH_4$ emissions are only from the saturated and inundated portion of the grid cell (Fig. S2). Despite this disagreement, $CH_4$ processes in the model follow the seasonal variation in soil temperature. This relationship, however, is only possible to analyze on a qualitative basis. A positive non-linear correlation between soil temperatures and $CH_4$ emissions is observed in all comparisons. Fitted polynomial curves are plotted on top of each data set. During 2015 (and 2014, data not shown), the $CH_4$ emissions measured with EC drop faster with the changes of temperature until freezing conditions. Between −3 and −14 °C, little variation in the lowest $CH_4$ emissions is observed, whereas the change of modeled $CH_4$ emissions with respect to changes in soil temperature is more gradual within that range of sub zero temperatures. Lower methane emissions in the model compared to

those in the EC data, take place in winter and are associated with even lower soil temperatures than the ones registered by the sensors in the redox systems.

### 3.4. Year-round modeled methane emissions

Domain means of the seasonal courses of $CH_4$ emissions from the different $CH_4$ transport pathways in 2014 and 2015 from the reference simulation, as well as the daily mean snow depth, are shown in Fig. 10. The results show a distinct seasonality for each of the individual methane emission pathways. Overall, the lowest $CH_4$ emissions occur between November and May. During these months, the timing of the $CH_4$ emissions through the snow is largely modulated by the changes of the snow depth, and accordingly, takes place predominantly in spring and autumn. The methane emissions via plants, ordinary molecular diffusion, and ebullition are mostly restricted to the period May through mid-November in areas when and where $h_{snow}$ does not exceed 5 cm (or it is absent). The magnitude of the $CH_4$ emissions through molecular diffusion is the least relevant among the four modeled transport pathways.

### 3.4.1. Summertime $CH_4$ transport pathways

From May to mid-November, $CH_4$ emissions take place only in the grid cells with inundated areas, with the highest flux rates simulated for the center west of the domain (Fig. S5). During this period $h_{snow}$ is either absent or does not exceed 5 cm. Ebullition precedes the emissions through plants during late March 2014 and during early April 2015. In both years, the mean of the $CH_4$ emissions in the model domain through ebullition rise steadily, followed by a short but pronounced decrease to 3.5 mg $CH_4$ m$^{-2}$ d$^{-1}$, the ebullition of $CH_4$ rises again to reach its maximum during mid summer with similar magnitude in both years (7.2 mg $CH_4$ m$^{-2}$ d$^{-1}$). This maximum is achieved later in 2014 (9$^{th}$ of August) than in 2015 (16$^{th}$ of July) and during the same days than the peak of maximum gas transported through plants. The domain means of the $CH_4$ emissions through plants reached their maximum value of 15 mg $CH_4$ m$^{-2}$ d$^{-1}$ in both years. Similar to ebullition, right after the summer maximum of emissions, the emissions through plants start to decrease until mid-November. In contrast to 2014, a shoulder is shaped in the emissions through ebullition and plants, and to less extent in diffusion, during the end of August to mid September in 2015, indicating the continuation of $CH_4$ emissions even as the soil starts to freeze (Fig. 9). Annually, $CH_4$ transported by plants is the dominant pathway, contributing 61 % to the total domain mean annual $CH_4$ emissions. These emissions take place from May to mid-November in areas where $h_{snow}$ < 5 cm, and are restricted to areas where C3 grasses are present (Fig. S1 and the panels of the first column in Fig. S5). The gas transported through ebullition is 33.9 % in 2014 and 35.7 % in 2015 of the total annual $CH_4$ emissions.

Methane emissions through ordinary molecular diffusion also take place if $h_{snow}$ < 5 cm in the inundated portion of the grid cells (panels in the third column of Fig. S5). In the absence of snow during summer and early autumn, $CH_4$ emissions via diffusion in the model domain average about $2.9 \times 10^{-3}$ mg $CH_4$ m$^{-2}$ d$^{-1}$ (similar in 2014 and 2015), while during late autumn, winter and early spring the emissions via this pathway are only possible if $h_{snow} \geq 5$ cm. For those few grid cells with $h_{snow}$ < 5 cm during the non-growing season (November to May), the $CH_4$ emitted via molecular diffusion is two to three orders of magnitude lower (mean of $3.4 \times 10^{-5}$ mg $CH_4$ m$^{-2}$ d$^{-1}$ for both years) than during the growing season (June to September) (Fig. 9). Methane transported via molecular diffusion during the growing season contributes only 0.02 % to the total $CH_4$ annual budget.

### 3.4.2. Impact of snow on the winter and seasonal variation of $CH_4$ emissions

During early spring, late autumn and winter, methane emissions take place through a layer of snow ≥5 cm deep. The mean maximum accumulation of snow in the model domain takes place in spring: earlier in 2014 (0.23 m on 21$^{st}$ March) than in 2015 (0.17 m on 8$^{th}$ March). The spatial distribution of the spring snow depths in 2014 and 2015 (Fig. S4a and b) show deeper snow layers in the dryer southwestern part of the model domain. On average, the layer of snow starts to melt rapidly at the beginning of May in 2014 and at the end of April in 2015, reaching total snowmelt by 2$^{nd}$ June 2014 and 27$^{th}$ May 2015 (Fig. 9). The average $CH_4$ emissions through the snow in the entire model domain during January and February are 0.17 mg $CH_4$ m$^{-2}$ d$^{-1}$ in 2014 and 0.12 mg $CH_4$ m$^{-2}$ d$^{-1}$ in 2015. The $CH_4$ emissions fluctuate through the winter, and these changes are related to changes in the thickness of the snow cover. During the rapid snowmelt period in spring (March, April and May), the daily domain average $CH_4$ emissions to the atmosphere through the snow increase (Fig. 9) with domain mean average spring $CH_4$ emissions of 0.65 $CH_4$ mg m$^{-2}$ d$^{-1}$ and 0.43 mg $CH_4$ m$^{-2}$ d$^{-1}$ in 2014 and 2015, respectively. The maximum domain mean daily emissions of $CH_4$ outside the growing season are modeled during May, with 1.66 mg $CH_4$ m$^{-2}$ d$^{-1}$ in 2014 and 0.96 mg $CH_4$ m$^{-2}$ d$^{-1}$ in 2015, and these take place predominantly in the central part of the model domain (panels in the fourth column of Fig. S5). In the entire model domain, the emissions of $CH_4$ through snow contribute 4.7 % and 2.7 % to the total mean annual $CH_4$ emissions for 2014 and 2015, respectively. Although deeper spring snow layers are modeled in 2014 than in spring 2015 (Fig. 9) in the areas where $CH_4$ is emitted to the atmosphere (Fig. S4a and b), the total methane emissions through snow from January to mid-May amount to ~70 mg $CH_4$ m$^{-2}$ in 2014, and only 66 % of that value in 2015 (~46 mg $CH_4$ m$^{-2}$; Fig. 9).

Integrated over the model domain during autumn, the snow starts to accumulate later in 2014 (9[th] October 2014) than in 2015 (30[th] September 2015), and the snow layer becomes rapidly deeper until December at a similar accumulation rate for both years (Fig. 9). As the snow accumulates, the emissions via ebullition and plants decline, but diffusion through snow rises as soon as the snow depth reaches 5 cm in some grid cells. From November to December, the mean $CH_4$ emissions through the snow in the domain amount to 37.3 mg $CH_4$ m$^{-2}$ in 2014, and 33 % less in 2015 (12.4 mg $CH_4$ m$^{-2}$). The modeled $CH_4$ emissions through the snow only consider the ordinary molecular diffusion of $CH_4$ between the soil and the atmosphere, and the pressure pumping effects due to advection of gas by wind is not taken into account.

At the grid cell level, in Fig. 11 we show the $CH_4$ emissions through the snow from the EC measurements and those from grid cell A and B simulated by the JSBACH model, all at daily resolution. The time series of daily emissions are shown from the beginning of October 2014 to the end of April 2015 (Fig. 11a) and in October 2015 (Fig. 11c). The difference between the model methane emissions for grid cells A and B, and EC data is shown in Fig. 9b and d for the same cold season periods. Comparable to the EC measurements, the winter emissions in the model drop abruptly at the end of October 2014, remaining low until March 2015. During October 2014, the model $CH_4$ emissions in grid cell B are higher, while the emissions from grid cell A are more similar to the EC measurements (Fig. 11a). This is also found in the first half of October 2015 (Fig. 11c). However, during this month the EC measurements show no clear trend, while the model $CH_4$ emissions show a decreasing trend over time. During most of the winter in 2014/2015 (i.e. from November 2014 until April 2015), the modeled $CH_4$ emissions from grid cells A and B remain lower than the EC measurements by on average 2.8 mg $CH_4$ m$^{-2}$ d$^{-1}$. During January, February and March in 2015 the mean model $CH_4$ emissions for grid cells A and B are 0.4 mg $CH_4$ m$^{-2}$ d$^{-1}$, while the EC data show persistently higher values averaging 3.8 mg $CH_4$ m$^{-2}$ d$^{-1}$ for the same months (Fig. 11a). Model emissions start rising (2.0 mg $CH_4$ m$^{-2}$ d$^{-1}$) to values similar to those in the EC data (2.8 mg $CH_4$ m$^{-2}$ d$^{-1}$) only in mid-April.

To investigate if the $CH_4$ emissions from the model during the entire wintertime are equivalent to the total winter emissions measured by EC, we calculated the cumulative sum of the modeled $CH_4$ emissions and EC from October 2014 to March 2015. The uncertainty as the standard deviation of the monthly cumulative fluxes is shown in error bars for each data set (Fig. 11e). Our results show that, despite a higher earlier release of methane in grid cell A, the modeled total emissions released during that winter are not equivalent to those from the EC measurements, with the latter providing evidence for larger total $CH_4$ emissions in winter

than predicted by the model. The cumulative uncertainties are also larger in the eddy covariance data and this is due to the large daily variability compared to the model results. In our model, the emissions through the snowpack take only into account the molecular diffusion of gas, whereas the advection of gas due to wind as an additional transport pathway 815 is not included.

### 3.4.3. Impact of environmental controls on $CH_4$ flux seasonality

Several systematic interannual differences between the timing and magnitude of the individual $CH_4$ transport pathways in 2014 and 2015 were found in the model results. These include e.g. the maxima of the individual emissions, which occur a few days later in 2014 820 than in 2015. To improve the interpretation of the temporal variability of $CH_4$ emissions through the different pathways, we analyze the temporal changes in soil temperatures within the root zone (top five soil layers) as simulated by the model. It is important to note that because of the current structure of the model, the depicted soil temperature in Fig. 10b reflects the average conditions of the entire grid cell, and not only the inundated portion with 825 saturated soils where $CH_4$ emissions take place. Still, the analysis of the temporal changes in the mean grid cell soil temperatures gives an indication of the nature and magnitude of the seasonal changes that indirectly control the $CH_4$ emissions. The gradient of temperatures in the root zone for the entire domain between spring and summer is steeper in 2015 than in 2014 (Fig. 10b). The maximum soil temperatures are similar in both years (8.7 °C); however, 830 this maximum was reached at the beginning of August in 2014 while in 2015 the maximum was reached at the beginning of July and remained high throughout August. During the rest of the year, the mean soil temperatures were 2 °C higher in 2014 compared to 2015 (−4.5 °C and −6.5 °C, respectively). The mean changes in temperature in the top five soil layers reflect the changes in the air temperature as given in the atmospheric forcing data. According to the 835 mean air temperature in the model domain, the summer of 2014 was colder than the summer of 2015 (by up to 10 °C for individual days during June, Fig. S6a). This leads to delayed warming of the soil, later high $CH_4$ production, and thus a later release of $CH_4$ into the atmosphere during summer in 2014 than in 2015, as shown in Fig. 10a. These findings are in good agreement with those recently presented in Helbig et al., (2017b). In a comparison of 840 meteorological records of air temperature between 2013 and 2016 in northwestern Canada, the authors found that the coldest May of those years took place in 2014. As a result, during that year a shift in air temperature influenced the soil temperature, and with it the year-to-year methane fluxes, especially during spring.

Figure 10c depicts the model domain mean relative soil moisture content in the top five soil layers for 2014 and 2015. As with soil temperature, the soil moisture reflects the average conditions of the entire grid cell and not just those in the inundated portion where the soil moisture is set to nearly saturation levels. Although these values are not linked to the area of the grid cell where $CH_4$ is transported and emitted, we can still show the temporal changes of soil moisture content in the non-saturated portion of the grid cell between years and seasons. These changes can be linked to changes in precipitation patterns (Fig. S6b) and soil temperatures. According to the mean precipitation from the CRU-NCEP reanalysis data, more precipitation fell in the model domain during early July in 2014 compared to the same period in 2015 (Fig. S6b). This led to the top five soil layers becoming wetter on average in 2014 (Fig. 10c) and potentially allowed higher thermal capacity in the soil during that period. In contrast, more precipitation fell during most of August and September 2015 than for the same periods in 2014 (Fig. S6b) leading to an increase in the relative moisture content towards end of summer and early autumn (Fig. 10c). These changes in soil moisture influence the soil temperature at the grid cell scale, and thus the soil temperature feedbacks to the $CH_4$ processes. Therefore, it is possible to indirectly relate the effects of changes in grid cell scale soil moisture to the changes in the modeled $CH_4$ emissions.

The mean relative soil ice content in the top five layers of the model domain (Fig. 10d) was higher in winter and spring of 2014 than in 2015, and this is a general observation for the entire domain. However, the air temperatures from the reanalysis data during that period were on average higher in 2014 than in 2015 (Fig. S6a). The ice content decreases at a fast rate during June in both years, however the complete loss of ice in the soil is reached earlier in June of 2015 than in 2014, and this is a reflection of colder temperatures in June 2014, delaying the complete melt of the more abundant ice in the soil during that year relative to the same month in 2015 (Fig. S6a). The soil ice content feeds back to the modeled available pore space for $CH_4$ production, thus the ice content changes in the soil can be indirectly linked to the $CH_4$ emissions. The earlier reduction of ice content in the soil during June 2015 might have contributed to the earlier release of methane during that month, via ebullition, compared to 2014 (Fig. 9). The lower air and soil temperatures at the beginning of autumn in 2015 (Fig. S6a) led to higher ice content in the soil during October 2015 compared to 2014 (Fig. 10d). The soil temperatures remain warmer in autumn of 2014, enabling more $CH_4$ to be emitted during November 2014 when the snow starts to accumulate in contrast to 2015 (Fig. 9 and 10a).

## 4. Discussion

### 4.1. Sensitivity experiments

Through the model sensitivity experiments we identified that changes to the values of the parameters $\chi_{min\_cti}$ and $f_{CH4anox}$ caused statistically significant differences in the total $CH_4$ emissions ($p<0.001$). A significant increase in $CH_4$ emissions with increasing inundated surface area (TOPMODEL parameter $\chi_{min\_cti}$) highlights the importance of this approach to regulate the extent of the grid cell inundated areas. However, further investigations and improvements in the TOPMODEL approach, as well as a better integration into the hydrology scheme of JSBACH, are needed in order to better constrain the modeled $CH_4$ emissions with JSBACH. The results of our sensitivity experiments also provided evidence that the magnitude of the simulated $CH_4$ emissions responds strongly to changes in the parameter values of the fraction of anaerobic decomposed soil organic matter that becomes methane, $f_{CH4anox}$. In soil systems where fermentation and methanogenesis are exclusive processes, i.e. without the presence of alternative pathways for respiration via terminal electron acceptors by other microbial groups that ultimately can suppress the production of $CH_4$, the $CO_2$:$CH_4$ ratio after anaerobic carbon mineralization is normally 1:1 (Conrad, 1999), i.e. $f_{CH4anox} = 0.5$. We used this value in the reference simulation because it was previously reported in the literature as characteristic of water-saturated polygon centers (Preuss et al., 2013), and it is similar to the value reported for unsaturated zones in boreal bogs (Whalen and Reeburgh, 2000). However, in wetland areas, $CH_4$ is still subject to oxidation after its production and the $CO_2$:$CH_4$ ratio is expected to increase and to vary among types of wetlands (Bridgham et al., 2013). Thus, although the value of $f_{CH4anox}$ determines the fraction of $CH_4$ produced under anoxic conditions, this $CH_4$ still can undergo oxidation before it is emitted to the atmosphere. Furthermore, $f_{CH4anox}$ can be theoretically related to the fraction of $CH_4$ that is left after oxidation and before it is emitted to the atmosphere ($f_{ox} = 1 - f_{CH4anox\_left}$). Values of $f_{ox}$ have been previously reported as ranging between 0.6-0.7 for sites with vascular plants. On the other hand, it can be nearly equal to 1 in sites with, for example, a layer of *Sphagnum* moss, where the majority of the produced $CH_4$ is oxidized, or in bottom soils in pond centers where slow molecular diffusion of $CH_4$ takes place through the water (Knoblauch et al., 2016). Under the latter conditions, $f_{ox}$ can be approximated to $> 0.9$ (i.e. $> 90$ % of the produced $CH_4$ is oxidized before it is emitted to the atmosphere). This value has been estimated in polygonal ponds without vascular plants, empirically supporting the relevance of $CH_4$ oxidation below the water table in these types of

environments (Knoblauch et al., 2016). A lower $CH_4$ oxidation fraction occurs in the presence of vascular plants that are effective at bypassing the aerobic areas in the soil. Under these conditions, $f_{CH4anox\_left}$ can increase moderately from 0.2 to 0.4 (i.e. $f_{ox}$ is from 0.6 to 0.8, meaning that 60 to 80 % of the produced $CH_4$ is oxidized in the soil column). Although current estimates for $f_{CH4anox\_left}$ from laboratory and on-site experiments are still scarce, they mostly agree that those are lower than our reference value of 0.5. This is expected because $f_{CH4anox\_left}$ excludes the portion of $CH_4$ that is oxidized directly after production, whereas $f_{CH4anox}$ is only the initially produced $CH_4$. Still, our modeled $CH_4$ emissions might benefit from prescribing a spatially variable $f_{CH4anox}$ value linked to the distribution of vascular plants and soil wetness in the model domain.

As for the rest of the selected parameters for the sensitivity exercise, no significant differences were observed in the modeled $CH_4$ emissions for the individual pathways or the total flux. Specifically, varying the diameter of roots from finer to thicker, and varying the amount of available soil volume occupied by roots, did not cause significant differences in modeled $CH_4$ emissions with the new formulation of the plant transport in the JSBACH-methane model. These results suggest that the revisited and simplified formulation for plant-mediated transport of gas allows a reduction in the uncertainties of methane transported through this pathway, which previously relied on predefined plant root characteristics that are often not available from observational studies. Instead, we define the volume in the soil that is occupied by roots.

The lack of sensitivity in the $CH_4$ emissions to most of the selected parameters might ultimately be due to the explicit restriction of gas transport via diffusive processes modeled by Fick's first law (plant transport and molecular diffusion through snow) that was set in the model. The role of this restriction is to limit the diffusion of gas once the concentration gradient between two interfaces equals zero i.e. it reaches equilibrium. Thus, this restriction takes place when the concentration gradient between e.g. the gas in the soil pore spaces and within the plant's roots (for plant-mediated transport) or between the gas in the soil pore spaces and the atmosphere above the snow layer (for diffusion of gas trough the snow), equals zero. Because the transport pathways can occur in parallel (except for diffusion with and without a snow layer), or emissions can be shifted in time, the modeled total $CH_4$ emissions may not be influenced by the set of parameters tested here. Finally, changes in the threshold depth of snow that limit the diffusion of gas through this layer revealed some differences in the partitioning of the methane flux into the four transport pathways. These differences indicate that a thinner threshold depth favors the other three transport pathways.

However, the resulting total $CH_4$ emissions with the three tested snow threshold depths were not statistically different.

## 4.2. Year-round model methane emissions

We simulated for the first time year-round methane emissions in a Northeast Siberian region centered on the city of Chersky. Our results showcase the ability of the improved JSBACH-
methane model to reproduce seasonality in the $CH_4$ emissions when compared to fluxes measured by EC and chambers in a study site near Chersky. The different transport pathways in this process-based model play an important role to define the timing of the year-round emissions since they are closely linked to the soil physical state and speed of transport processes by their definition. During the growing season, plant-mediated transport dominated
the emissions, contributing about 61.4 % in 2014 and 61.7 % in 2015 of the total annual $CH_4$ emissions, followed by ebullition (33.9 % and 35.7 %) and molecular diffusion during summer when snow is not hindering the emissions (0.02 % for both years). These patterns agree well with the findings presented in Kwon et al. (2016) for the $CH_4$ emissions measured with chambers at the Chersky floodplain, and by Kutzbach et al. (2004) and Knoblauch et al.
(2016) in the Lena River Delta. In these works is shown that the dominant $CH_4$ transport pathway in tundra wetland ecosystems (about 70-90 % of the total annual emissions) is diffusion through the aerenchyma structures of the plants when they are present. Methane emissions during the non-growing season contributed 4.7 % and 2.7 % of the annual methane emissions in 2014 and 2015, respectively.

As for the methane oxidation, the bulk soil oxidation accounts for about 1 % of the total methane production during the growing season at grid cell scale, and only about 0.6 % for rhizospheric $CH_4$ oxidation (results not shown). This leads to most of the methane that is produced in the soil to be emitted to the atmosphere through the different transport pathways. Past observational and laboratory studies have estimated the methane oxidation in boreal and
tundra soils. Whalen and Reeburgh (2000) showed that about 55 % of the $CH_4$ diffusing from saturated boreal soils, were oxidized while reaching the surface. Through bottle incubations, Knoblauch et al. (2016) measured the volumetric $CH_4$ oxidation potential of soil and moss samples collected from ponds of the Lena Delta. The authors found that the fraction of produced $CH_4$ that is oxidized before it is emitted was between 61 and 78 % using a stable
isotope approach. In samples from pond areas without vascular plants, the fraction increased up to 90 % of the total produced $CH_4$ following a potential methanogenesis approach, and from diffusive $CH_4$ fluxes into the bottom water this was between 63 % and 94 %. Berestovskaya et al. (2005) measured $CH_4$ oxidation rates of different soil samples from the

Russian Arctic tundra and found that generally the rates of methane oxidation exceeded those to the rates of methane production especially at temperatures of 5 °C. For this to happen, methane-oxidizing bacteria rapidly consumes the methane released from the freshly thawed tundra soils and the methane already deposited in the unfrozen soil, and this takes place even before methanogens produce new methane. Based on these scarce observations in boreal soils, the oxidation processes in our model are still far off and need to be revisited in order to improve the contribution of the methane oxidation processes into the total methane emissions.

The JSBACH-methane model does not explicitly consider specific mechanisms related to the carbon decomposition and thaw in Arctic permafrost and wetland ecosystems, such as: $CH_4$ production in the soil from root exudates (Knoblauch et al., 2016; Ström et al., 2012), vertical transport of soil organic matter and its vertically resolved decomposition (Braakhekke et al., 2011, 2013; Koven et al., 2015), and microbial community dynamics (McCalley et al., 2014) involved in anoxic $CH_4$ oxidation or the production of $CH_4$ in anaerobic microsites confined in oxic soils. Although these processes might contribute substantially to the dynamics of $CH_4$, research on these processes in soil-permafrost and wetland environments is still lacking or poorly understood with controversial results so far (Bridgham et al., 2013).

At the grid cell scale, the characteristics defined in the model input parameters exert an important influence on the spatial heterogeneity and temporal variability of the modeled environmental controls and $CH_4$ emissions. For example, in the model domain the soil depths range between 0.1 to 10.6 m (Fig. S4c; grid cell A is 0.89 m and grid cell B is 10.6 m), whereas the depth of the root zone is from 0.1 to 0.89 m (Fig. S4d; grid cell A is 0.72 m and grid cell B is 0.67 m). Also, the cover fraction of vegetation differs among grid cells, and in this model, the coverage of C3 grasses is particularly relevant for $CH_4$ emissions through the plants roots, e.g. 33.3 % in the area of grid cell A and 91.6 % in grid cell B (Fig. S1). Finally, grid cell A has lower soil moisture and soil ice content relative to the pore volume in the top five soil layers, and larger inundated area, compared to grid cell B (Fig. S8a, b, and d). These differences predominantly explain the shift in the dominant growing season $CH_4$ transport pathways and seasonal changes between grid cells (Fig. S8e and f).

To further demonstrate the heterogeneity in the modeled total $CH_4$ emissions, we show in Fig. S5 the time series of the daily $CH_4$ fluxes in 2014 and 2015, for nine model grid cells (grid cell A and the eight grid cells surrounding it which includes grid cell B). In this area of the domain, the range of the mean emissions is between 24 to 75 mg $CH_4$ $m^{-2}$ $d^{-1}$, with similar values between years 2014 and 2015. To further analyze the spatial heterogeneity, more is

discussed below in the context of spatial distribution of $CH_4$ fluxes in the entire model domain (Fig. S9).

The modeled $CH_4$ emissions represent fluxes from exclusively inundated areas (water table at or above the surface), thus emissions from areas with a water table below the surface are neglected. In Figure A3b (Appendix A) are shown summertime $CH_4$ fluxes (June to August, 2014) measured with chambers in the Chersky floodplain (Kwon et al., 2016), plotted against the water table in the chamber microsite at the time of the flux measurements. $CH_4$ oxidation

predominantly exceeded production in dry microsites (water tables were below the surface up to about 10 cm), and this was evidenced by a small uptake of $CH_4$ (on average 3 mg $CH_4$ m$^{-2}$ d$^{-1}$). The fluxes in these dry plots were almost negligible during the growing season. Thus, the modeled fluxes of $CH_4$ represent the majority of the emissions in this tundra ecosystem.

    Our results show a good agreement between the modeled $CH_4$ emissions (at the grid cell

scale) and measured $CH_4$ emissions with EC and chambers. Overall, the modeled year-round and measured methane emissions at daily temporal resolution are in the same order of magnitude, and both fall within their monthly range of variability. In both the EC footprint area and model grid cell area, the methane emissions are not spatially homogeneous but bound to the distribution of wetland (inundated) areas, which are also linked to the type of

vegetation. This was demonstrated for summer of 2014, where EC $CH_4$ fluxes are in closer agreement to the model methane emissions after a linear scaling approach of the wet soil areas in the EC footprint.

    In the model, $CH_4$ emissions integrated in our study region were on average 22.5 mg $CH_4$ m$^{-2}$ d$^{-1}$ during the growing season of 2014 and 2015. These modeled values are also in good

agreement with measurements in other Arctic wetland areas influenced by permafrost using eddy towers, chambers, and more recently with airborne techniques. Kutzbach et al., (2004) reported $CH_4$ emissions of 28 mg $CH_4$ m$^{-2}$ d$^{-1}$ measured with chambers during the onset of the growing season from a polygon center of the wet tundra in the Lena Delta. For a variety of locations in polygons of the same region, Sachs et al. (2010) reported mean summer

methane emissions of about 55 mg $CH_4$ m$^{-2}$ d$^{-1}$. Knoblauch et al. (2016) presented mean summer fluxes of 46 mg $CH_4$ m$^{-2}$ d$^{-1}$ also measured with chambers at the margins of ponds also in Lena Delta. Larger summer methane emission values have been reported elsewhere, e.g. from automatic chambers at the Zackenberg research station, with maximum emissions of about 168 mg $CH_4$ m$^{-2}$ d$^{-1}$ at the onset of the growing season (Mastepanov et al., 2008).

Merbold et al. (2009) reported $CH_4$ emissions of ~600 mg $CH_4$ m$^{-2}$ d$^{-1}$ measured by chambers at the peak of the growing season (August) in 2005 in the Chersky floodplain.

At the lower end of the observational data, Wille et al. (2008) measured $CH_4$ emissions of about 30 mg $CH_4$ $m^{-2}$ $d^{-1}$ during mid-summer in the Lena Delta. The authors argued that the measured values were generally lower than other estimates and that the main controlling factors of their measurements were low soil temperatures and the influence of atmospheric turbulence during their period of study. Rinne et al. (2007) reported $CH_4$ fluxes of about 84 mg $CH_4$ $m^{-2}$ $d^{-1}$ measured using EC at a boreal fen in southern Finland. Eddy covariance $CH_4$ fluxes measured in the Alaskan tundra showed a larger range of values, with an average of 32 mg $CH_4$ $m^{-2}$ $d^{-1}$ during the onset of the growing season (Zona et al., 2016). Finally, airborne measurements of $CH_4$ emissions from wetlands in Alaska were estimated to be about 56 mg $CH_4$ $m^{-2}$ $d^{-1}$ (Chang et al., 2014).

### 4.3. Representation of inundated fractions of the grid cell

In this model version, we incorporated the TOPMODEL approach to explicitly model the distribution of inundated areas according to the topography profile. Although this is still only a robust approximation, the implementation of this approach enabled the representation of wetlands in the highly heterogeneous landscape of northeastern Siberia, which is not possible with the traditional hydrology scheme of JSBACH because it does not allow standing water (Kaiser et al., 2017; Hagemann and Stacke, 2015). In contrast to standard remote sensing products, the highly resolved product of Reschke et al. (2012) is the best data available so far for our application. After TOPMODEL parameter adjustments, still large differences remain between $w_{mod}$ and $w_{rs}$, however, most of these can be attributed to uncertainties due to the model technique to simulate wetlands and its horizontal resolution.

In the model, the fraction of the grid cell that becomes inundated refers to the area where the water table lies at or above the soil surface and varies with changes in available solid (snow) and liquid water (precipitation) leading to a 15-day time step. However, in nature, most wetlands have periods of the year when no visible standing water is above the surface, and the water table is located few cm below the surface (Bridgham et al., 2013; Kwon et al., 2016). Previous studies have demonstrated the dependence of $CH_4$ emissions on the location of the water table in tundra ecosystems (e.g. Helbig et al., 2017a; Kwon et al., 2016; Merbold et al., 2009; Sturtevant et al., 2012; Zona et al., 2009) and as shown before, the work by Kwon et al. (2016) revealed that $CH_4$ fluxes measured by chambers in the Chersky floodplain are significantly influenced by the location of the water table at the plot scale. Larger $CH_4$ emissions were measured in sites where the water table was at or above the surface compared to drier sites. With the TOPMODEL approach it is not possible to characterize the location of inland water bodies (i.e. lakes), and the explicit location of peatlands is also not taken into

account because the model only considers mineral soils. This separation would help to identify the inundated portions of land with more or less relative input of organic carbon to better localize the methane emissions. The lack of organic layers representation in the model is mainly due to the difficulties of coupling sub-grid scale hydrology and carbon cycle in a holistic manner. The model however, considers the amount of carbon that is available in the soil, based on a soil carbon and litterbag approach, and that one for decomposition and production of methane.

Future important advancements in our model are necessary in the context of a process-based representation of peatland extent as well as the $CH_4$ balance in non-inundated areas; currently, these are not taken into account in our study, contrary to Kaiser et al., (2017) for a site level study. This is especially relevant for the applicability of this model to other regions where uptake of methane in dry areas might play a substantial role (e.g. Flessa et al., 2008; Jørgensen et al., 2015).

### 4.4. **Impact of the revised model structure**

The model reproduces well the observed temporal trends in the $CH_4$ emissions, and patterns can be linked to changes in the environmental controls. However, integrating the TOPMODEL approach into JSBACH led to a decoupling of some physical soil state variables. Soil moisture content, soil ice content and soil temperature influence the heat capacity of the soil and the ice content (i.e. soil freeze and thaw processes), and control the accumulation of gas, microbial activity, diffusion rates of gases, and the amount of oxygen in the soil (Sturtevant et al., 2012; Wickland et al., 1999; Pirk et al., 2016). In the JSBACH-methane model, soil moisture of the ice-free soil pores in the inundated part of the grid cell was set to 95 % saturation, for purposes of justifying inundation in the TOPMODEL approach. Although, the temperature and ice conditions in the soil are not influenced by this change, this leads to a missing link in terms of the distribution of soil water to soil ice or soil moisture. However, a direct connection between each of these physical variables and the $CH_4$ processes is definitively present.

Between data years, the soil temperature in summer of 2014 was lower from mid-June than at the same period during 2015, leading to a phase lag in the maximum summer $CH_4$ emissions of nearly a month earlier in 2015. However, during the autumn of 2015, the soil was colder than in 2014, with more soil ice content and less methane production during this period of the year. This translates into lower $CH_4$ emissions to the atmosphere from November 2015 until the end of the year than during the same period in 2014. Because seasonal changes in soil wetness must be taken into account for modeling year-round gas emissions in permafrost

Arctic tundra environments (Pirk et al., 2016), the JSBACH-methane version used for this work requires further improvements to better integrate the TOPMODEL approach using a fully mechanistic thermal and hydrology scheme at landscape scale able to interact with inundated area fractions at grid cell scale (Stacke and Hagemann, 2012).

## 4.5. Simulation of grid cell soil temperature

Large uncertainties in the simulation of $CH_4$ emissions from northern wetlands with models come from limitations in the representation of freezing and thawing soil processes, snow layer dynamics, and the robust mapping of the distribution of wetlands (Bridgham et al., 2013). Kaiser et al., (2017) reported that the process-based JSBACH-methane module considers the effects of permafrost thawing and freezing, thus also the seasonal changes of the physical state of the soil on $CH_4$ processes. However, our analysis of the soil temperature profiles showed that during the cold season the simulated soil temperatures are nearly 10 °C lower than the values measured on site. Moreover, they gradually increase through spring and summer to reach values similar to the measurements. In contrast, the soil temperature seasonal cycle observed in the Chersky floodplain shows strong links to thawing and freezing processes (Göckede et al., 2017). These differences could be related to a negative bias in soil moisture content at the grid cell scale – which is driven by non-inundated areas - used to calculate the soil thermal regime. This limits the validity of the soil thermal properties as well as changes in latent heat. In addition, the carbon decomposition scheme used in this model version is driven by precipitation and atmospheric temperature. Therefore, actual changes in the soil temperature regime and wetness are not fully linked to the carbon dynamics. Finally, in addition to snow, near-surface vegetation in tundra environments (e.g., mosses and lichens) are also effective thermal insulators of soil (Porada et al., 2016), regulating high surface temperatures in summer and cold temperatures during winter, and should be taken into account in a next version of the land surface model.

## 4.6. Role of non-growing season $CH_4$ emissions

In this work, we present, to our knowledge, the first simulated $CH_4$ emissions during the non-growing season with a land surface model at a regional scale. Our results show that changes in the snow layer depth control the temporal variation of the molecular diffusion of $CH_4$ through the snow. Our sensitivity studies corroborate that setting a thinner layer of snow as a threshold depth to switch to the $CH_4$ emission process during the cold season, only promotes some changes in the partitioning of the methane flux among the four transport pathways. For example, a thinner snow layer promotes an earlier release of $CH_4$ than was otherwise emitted during late summer with a thicker layer of snow. However, the magnitude of the emissions

through the snow is also determined by the amount of $CH_4$ that is produced, calculated from the decomposed carbon that is driven only by air temperature and precipitation in the carbon decomposition module. Changes in the physical properties of the snowpack (i.e. porosity and density) defined in the model have no clear effect on the timing of the emissions through the snow; this may lead to the conclusion that our choice of values for the capacity of snow to transport $CH_4$ was large enough. The physical restriction of gas transport via diffusive processes modeled by Fick's first law ensures that only physically possible rates of gas transport are being modeled.

In the recent work by Pirk et al. (2016), the authors demonstrated that the fluxes of $CH_4$ through the snowpack of permafrost Arctic wetlands during wintertime reflected a continuous emission of low amounts of gas still being produced in the soil, rather than solely the release of gas stored in the soil that was produced during the preceding growing season. These observations are in agreement with those from Mast et al. (1998), where the authors reported evidence of microbial activity throughout winter in subalpine soils permitted by the insulating effect of the snow layer. The results of Pirk et al. (2016) showed that there was no apparent sink or source of $CH_4$ within the snowpack, and their measurements captured a linear concentration gradient through the snow (Pirk et al., 2016). This observation validates the application of Fick's first law for diffusion of fluxes through the snow during winter, as applied in our model configuration. However, our formulation does not take into account the "pressure pumping" process reported by Massman et al. (1997) and Bowling and Massman (2011) that is related to the persistent advection of gas enhanced by wind through the snowpack. Based on isotopic analysis of $CO_2$ through the snowpack of a mountain forest, Bowling and Massman (2011) found that in the presence of wind, the pressure pumping effect contributed up to 11 % of the total emissions during winter.

Our comparison at a grid cell scale to wintertime fluxes measured from EC at the Chersky floodplain (from January until March 2015 on average 3.8 mg $CH_4$ m$^{-2}$ d$^{-1}$) shows that the modeled $CH_4$ emissions during this season (0.4 mg $CH_4$ m$^{-2}$ d$^{-1}$) are consistently lower by about one order of magnitude. The measured EC fluxes are similar to other measurements with other methods from earlier studies. The work by Panikov and Dedysh (2000) showed winter methane emissions measured by chambers of about 5.0 mg $CH_4$ m$^{-2}$ d$^{-1}$ from boreal peat bogs in western Siberia in mid-February. Pirk et al., (2016) measured $CH_4$ fluxes above the snowpack of about 2.4 mg $CH_4$ m$^{-2}$ d$^{-1}$. In subalpine soils covered with snow, Mast et al. (1998) reported average winter $CH_4$ emissions of 4.4 mg $CH_4$ m$^{-2}$ d$^{-1}$ in moist soils calculated from samples collected through the snowpack. However, our modeled winter $CH_4$ emissions

are comparable to those reported by Smagin and Shnyrev (2015) of about 0.6 mg $CH_4$ $m^{-2}$ $d^{-1}$ measured by chambers during the coldest months of the year (February and March) in environments with different soil wetness in a West-Siberian bog landscape.

Moreover, it is important to note that our results represent average values of a grid cell with a 0.5° x 0.5° horizontal resolution, whereas measurements represent a much smaller spatial scale. Integrating the latter to the grid cell level must lead to an overestimation of the emission values at the grid cell level.

Other works have reported large $CH_4$ emissions from dry areas during the non-growing season. Using EC measurements, Zona et al., (2016) showed large fluxes from dry areas of the Alaskan tundra during the zero curtain period. Also, the findings of Mastepanov et al. (2013) imply that a portion of the active layer still remains free of ice during late autumn, and moisture and temperature changes are limited by the low thermal conductivity and heat capacity of dry soils. In the model, the consistently lower $CH_4$ emissions during winter can be explained by a low bias in soil temperature, leading to a low bias also in methanogenesis and larger oxidation within the topsoil.

## 5.    Conclusion and outlook

The refined configuration of the JSBACH-methane model presented in this study has the ability to represent grid cell scale year-round $CH_4$ emissions at a comparable magnitude to those measured by chambers and EC in the same study area. The model was successfully applied to a regional domain in a floodplain of Northeastern Siberia underlain by permafrost. The seasonal transition of the four $CH_4$ transport pathways is mainly controlled by changes in the soil temperature, and only indirectly linked to soil moisture. The majority of the annual emissions take place through vascular plants. Given the relatively large scale of the model regional domain and uncertainties in the methane fluxes associated with forcing the model with reanalysis data and evaluating net emissions at the local level, it is difficult to quantify the emissions through the individual emission pathways. However, this explicit representation is necessary in process-based modeling, particularly for forecast regarding Arctic methane emissions under future climate scenarios.

The findings of this study demonstrate that to improve the understanding of the interannual variability of $CH_4$ fluxes form wetlands in boreal permafrost areas, to improve process model evaluation, and contribution of the individual emission pathways, more highly resolved temporal observational data is required, specially of year-round $CH_4$ EC fluxes and soil temperatures which are generally scarce and challenging for boreal and tundra areas. This is

particularly important to improve modeling $CH_4$ emissions through snow, which in our model show a low bias when compared to EC measurements.

Finally, our model will greatly benefit from further improvements for regional simulations, which will also contribute to advancing the application to a global scale. In summary, the following model improvements are suggested: 1) a descriptive scheme for snow layer dynamics may benefit the simulation of wintertime $CH_4$ emissions, including pressure pumping effects due to advection of gas enhanced by wind, 2) improvements to prescribed model parameters such as soil depth until bedrock and initial soil moisture saturation, which are normally obtained from global scale configurations of JSBACH, 3) an improved connection between the TOPMODEL approach for simulating the inundated fractions in a model grid cell, and soil state variables such as soil moisture, soil temperature, and ice content. This in turn might lead to improvements in the soil thermal properties for dry versus wet areas, and to the representation of non-inundated areas to understand the dynamics of sources and sinks of $CH_4$. This might be alleviated if sub-grid scale heterogeneity is included in future model developments.

4) finally, improving the temporal transitions and seasonality of the water table levels will help to better constrain the surface heterogeneity of hydrologic responses to permafrost thaw and the spatial distribution of carbon decomposition.

## 6.     Code and data availability

The land surface model JSBACH used in this study is intellectual property of the Max Planck Society for the Advancement of Science, Germany. The JSBACH source code is distributed under the Software License Agreement of the Max Planck Institute for Meteorology and it can be accessed on personal request. The steps to gain access are explained under the following link: http://www.mpimet.mpg.de/en/science/models/license/. The EC dataset is available through the European Fluxes Database Cluster (site code: RU-Ch2). The chamber flux data is available upon request to M. Göckede (mgoeck@bgc-jena.mpg.de).

**Appendix A: Details on in-situ flux observation program.**

*Uncertainty assessment in EC flux data.*

The uncertainty analysis for the EC flux data followed procedures well-established in literature (Aubinet et al., 2012), and was split into random and systematic errors. The largest sources of random errors are associated with the turbulent sampling and instrument issues. These errors were quantified for each 30 min flux value through the flux processing software

TK3 (Mauder and Foken, 2015). Errors related to footprint uncertainties were not quantified, since there are no major transitions in biome types within the core areas of the flux footprints. Systematic errors can be introduced by unmet theoretical assumptions and methodological challenges, as well as by instrument calibration and data processing issues. To minimize this error, the instruments in the Chersky area were maintained and calibrated on regular basis.

Data intercomparison with a second EC tower located 600 m away of the tower that is source of the data presented here yielded no systematic offset in the frequency distributions of wind speed, sonic temperature, and methane mixing ratios between the two towers. The TK3 software package contains all the required conversions and corrections for the flux data processing, and yielded good agreement in a comparison with EddyPro (Fratini and Mauder,

2014). To avoid methodological issues that may bias flux data results, we employed a rigid post-processing quality control and flagging system scheme based on well-established analyses for stationarity and well-developed turbulence (Foken and Wichura, 1996), followed by additional tests to flag implausible data points in the resulting flux time series. Further details on this analysis are presented in Kittler et al. (2017).

No u*-threshold was applied to the flux dataset, since we determined stationarity of the signal and integral turbulence characteristics are also for nighttime conditions. This information facilitates identifying datasets with regular turbulent exchange also during stable stratification, therefore producing fewer gaps compared to a bulk exclusion of data during stable nighttime stratification through the u*-filter method. After filtering out low-quality

fluxes, the data coverage of methane fluxes was 86 % during the growing season and 67 % during the winter from the original full 30 min flux data set (Kittler et al., 2017). To produce a continuous flux record for quantification of long-term $CH_4$ budgets, the remaining gaps in the data were filled by averaging the existing flux data within a moving window of 10-day length centered on the gap. Uncertainties for gap-filled values were quantified as standard

deviation within the corresponding window, similar to the definition of gapfilling uncertainties for the $CO_2$ flux via the marginal distribution sampling routine of Reichstein et al. (2005).

To produce aggregated uncertainty values for longer time periods, we applied the procedures suggested in Rannik et al. (2016). All random errors were combined by considering them as

independent variables that normally decrease with the length of the averaging period. Averaged over 2014 and 2015, the $CH_4$ flux uncertainty based on the 30 min data is 7.4±8.3 nmol $m^{-2}$ $s^{-1}$, a result comparable to 4.7±3.8 nmol $m^{-2}$ $s^{-1}$ reported for a fen ecosystem by Jammet et al. (2017).

*Source-weight function of the EC flux data and scaling for model flux evaluation*

We conducted a source-weight analysis (i.e., footprint analysis), to determine the fractional contribution of different land cover types within the field of view of the EC flux tower. Source-weight functions for each 30-min flux measurement were computed based on the Lagrangian stochastic footprint model of Rannik et al. (2003). Footprints were accumulated, analyzed and interpreted using an approach presented by Göckede et al. (2006, 2008). We

projected these footprints onto a land cover map from WorldView-2 with 2 m horizontal resolution (Fig. A1). In the context of the presented study, we aggregated the originally identified 22 land cover classes into 9 classes to concentrate on the dominant elements of the vegetation community structure.

Since the EC tower is situated on a slightly elevated patch of tundra, tussocks and shrubs

featuring various levels of wetness (red and orange colors in Fig. A1) dominate the immediate surroundings. Even though inundated parts of the study area, in this case identified by the prevalence of the cotton grass *Eriophorum angustifolium* (blue-ish colors in Fig. A1), are dominating the area encircled by the 10 % isoline that is used here to mark the boundary of the cumulative footprint area, they are mostly present in the outer reaches, therefore

combining just about 26 % of the total flux signal sampled by the eddy system. Another 31 % is contributed by wet to moist tussock tundra with some shrubs. Overall coverage fractions within the major wetness categories remain approximately constant between tower footprint and two larger regions covered within the same WorldView-2 dataset, indicating that this composition of wetness levels is typical for the Kolyma floodplain ecosystems analyzed

within the context of this study. Furthermore, C3 grasses cover 33.3 % of the model grid cell A, whereas the inundated fraction in that same grid cell ranges between 17.7 % and 19.9 %, calculated as a 10-day mean during June, July and August 2014. Thus, to improve the comparison between EC and model $CH_4$ fluxes, we corrected the 10-day EC mean fluxes (related to 26 % of the footprint wet cotton grass area), through a linear scaling to obtain the

fluxes from a smaller wet area i.e. corresponding to the 10-dat mean inundated fraction of the model grid cell. The results of this scaling exercise are shown in Fig. A2. The non-scaled 10-day EC fluxes for the period of analysis are on average 65 mg $CH_4$ $m^{-2}$ $d^{-1}$, and the scaled fluxes decreased on average by 24 % (mean 49.6 mg $CH_4$ $m^{-2}$ $d^{-1}$) after a correction by considering a smaller wet area within the footprint, reaching a magnitude similar to the 10-

day mean fluxes from the model (48.6 mg $CH_4$ $m^{-2}$ $d^{-1}$). This exercise emphasizes that wetness is the dominant control for total methane emissions in these ecosystems.

*Flux chamber observations.*

As shown in the study of Kwon et al. (2016), in the Chersky site were located two transects of 10 permanently installed PVC collars for flux chamber measurements. With distances of approximately 25 m between individual microsites, both transects cover a distance of ~225 m within a drained and a control section in this area. Site locations were selected quasi-randomly to reflect the dominant microsite characteristics (e.g., vegetation composition, wetness level) that were observed at each of the target locations. With a chamber footprint of 60 cm x 60 cm, this technique allowed studying microsites with rather homogeneous environmental conditions, as compared to the EC fluxes with often heterogeneous footprint areas. Details on the chamber program, overall methane flux rates observed, and functional relationships with e.g. soil temperature, vegetation and wetness levels are provided in Kwon et al. (2017).

Figure A3a displays average flux rates for wet and dry microsites observed within a drained and control transects during sampling campaigns in summer 2014 (Kwon et al., 2016). These results demonstrate that methane release rates were virtually zero in the absence of standing water. At some of the dry microsites, defined by having the water table below the surface (on average up to 10 cm), slightly negative $CH_4$ flux rates were predominantly observed (mean of 3 mg $CH_4$ $m^{-2}$ $d^{-1}$) and almost negligible emissions, indicating the oxidation of methane (uptake) under highly aerobic conditions. Thus, the methane emissions in this tussock tundra ecosystem of Northeastern Siberia take place predominantly in wet areas.

## 7.    Special issue statement

This manuscript is a contribution to the special issue dedicated to the project: "Changing Permafrost in the Arctic and its Global Effects in the 21st Century (PAGE21)".

## 8.    Acknowledgements

This work was supported through funding by the European Commission (PAGE21 project, FP7-ENV-2011, Grant Agreement No. 282700, and PerCCOM project, FP7-PEOPLE-2012-CIG, Grant Agreement No. PCIG12-GA-201-333796), the German Ministry of Education and Research (CarboPerm-Project, BMBF Grant No. 03G0836G), and the AXA Research Fund (PDOC_2012_W2 campaign, ARF fellowship M. Göckede). We thank Andrew Durso for the English proofread of this manuscript.

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

**Table 1.** Summary of the most relevant prescribed parameters in the JSBACH-methane control and reference simulations.

| Parameter | Description | Value | Unit |
|---|---|---|---|
| $\chi_{min\_cti}$ | Threshold to define maximum areas that can be flooded in a grid cell (TOPMODEL) | 12 | [-] |
| $f$ | Exponential decay of transmissivity with depth (TOPMODEL) | 2.0 | [-] |
| $d_r$ | Root diameter | 2 | mm |
| $r$ | Resistance factor of root exodermis | 0.8 | [-] |
| $h_{exo}$ | Thickness of exodermis | 0.06 | mm |
| $R_{fr}$ | Principal fraction of the pore-free soil volume occupied by roots | 40 | % |
| $\phi$ | Porosity of snow | 0.64 | [-] |
| $h_{snow}$ | Snow depth threshold | 5 | cm |
| $f_{CH4anox}$ | Fraction of anoxic decomposed carbon that becomes $CH_4$ | 0.5 | [-] |
| $D_{air}^{CH_4}$ | Diffusion coefficient of $CH_4$ in free air at 0 °C and 1 atm | $1.95 \times 10^{-5}$ | $m^2\ s^{-1}$ |
| $D_{air}^{O_2}$ | Diffusion coefficient of $O_2$ in free air at 0 °C and 1 atm | $1.82 \times 10^{-5}$ | $m^2\ s^{-1}$ |
| $\rho_{ice}$ | Ice density | 910 | kg m$^{-3}$ |
| $\rho_{snow}$ | Snow density (Together with $\rho_{ice}$ leads to: $\phi$ =0.64 and $\tau$ =0.77) | 330 | kg m$^{-3}$ |

**Table 2.** Results from sensitivity experiments (the specific descriptions of the parameters listed below are given in Table 1). Statistical $p$-values are given for the experiments whose results significantly differ from the results in the reference simulation.

| Variable | Value | Unit | Annual mean of total CH$_4$ / (mg CH$_4$ m$^{-2}$ d$^{-1}$) |
|---|---|---|---|
| $\chi_{min\_cti}$ | 11 | [-] | $4.2 \pm 5.0$* |
| | 12$^{§}$ | | $6.2 \pm 7.3$ |
| | 13 | | $9.2 \pm 10.7$* |
| $d_r$ | 2$^{§}$ | mm | $6.2 \pm 7.3$ |
| | 5 | | $6.2 \pm 7.3$ |
| | 8 | | $6.2 \pm 7.3$ |
| $R_{ff}$ | 0.2 | [-] | $6.2 \pm 7.3$ |
| | 0.4$^{§}$ | | $6.2 \pm 7.3$ |
| | 0.6 | | $6.2 \pm 7.3$ |
| $\phi$ | 0.64$^{§}$ | [-] | $6.2 \pm 7.3$ |
| | 0.71 | | $6.2 \pm 7.3$ |
| | 0.86 | | $6.2 \pm 7.3$ |
| $h_{snow}$ | 1 | cm | $6.2 \pm 7.3$ |
| | 3 | | $6.2 \pm 7.3$ |
| | 5$^{§}$ | | $6.2 \pm 7.3$ |
| $f_{CH4anox}$ | 0.1 | [-] | $1.2 \pm 1.4$* |
| | 0.3 | | $3.7 \pm 4.3$* |
| | 0.5$^{§}$ | | $6.2 \pm 7.3$ |

 $^{§}$parameter value in reference simulation; *significant at $p<0.001$

**Figures**

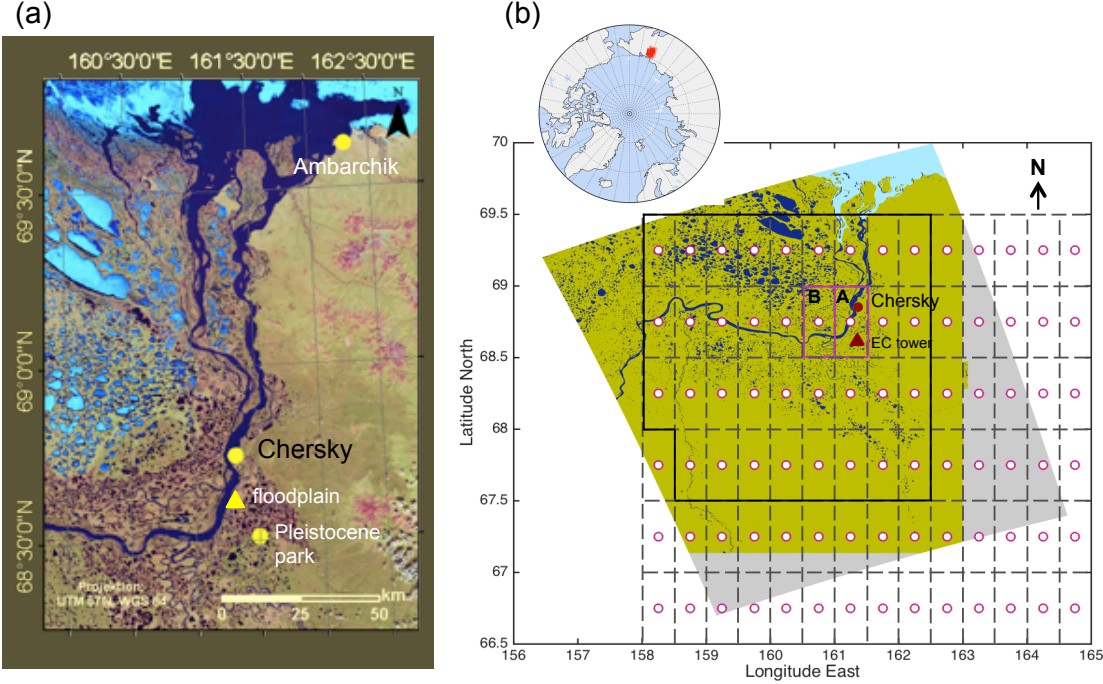

**Figure 1** – a) MODIS image showing the heterogeneous landscape in most of the model domain in Northeast Siberia, also showing the location of nearby cities and the floodplain, b) geographical location of the model domain used in this study also depicted with the midpoints of the model grid cells (pink circles) and boundaries (dashed lines) underlain by a geoTIFF image (data from 01-10 July 2007) from the EAWS product. The boundaries of the

grid cells A and B are delimited with pink lines. The continuous dark line delimits the 35 model grid cells used for the evaluation of modeled inundated areas against the EAWS product.

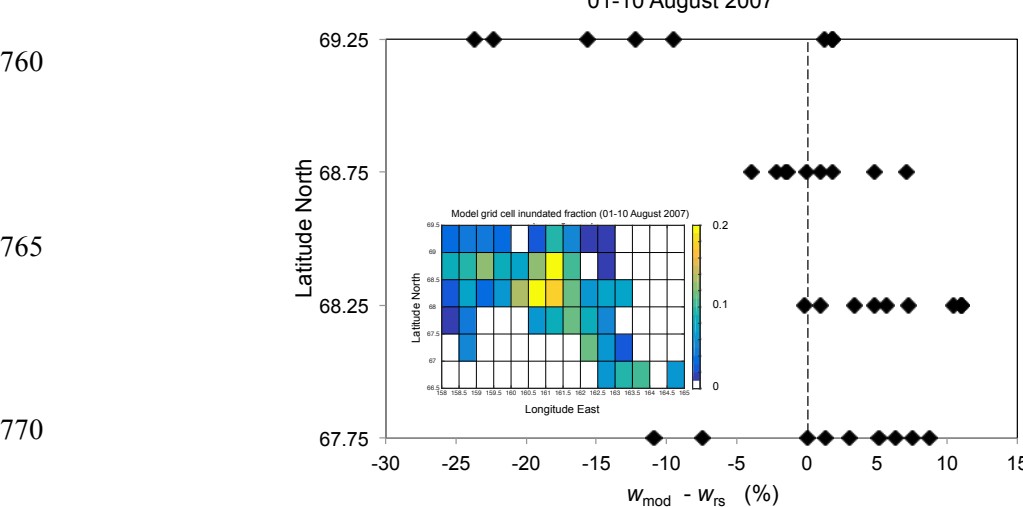




**Figure 2** – Latitudinal distribution of the difference between the fractions of the grid cell inundated areas simulated with TOPMODEL in JSBACH-methane ($w_{mod}$) and the inundated

areas estimated from the EAWS product ($w_{rs}$) for the same grid cells (01-10 August 2007). Inset figure is the mean spatial distribution of the fraction of inundated areas in the model domain for 01-10 August 2007. Grid cells with inundated areas < 1 % are not shown.

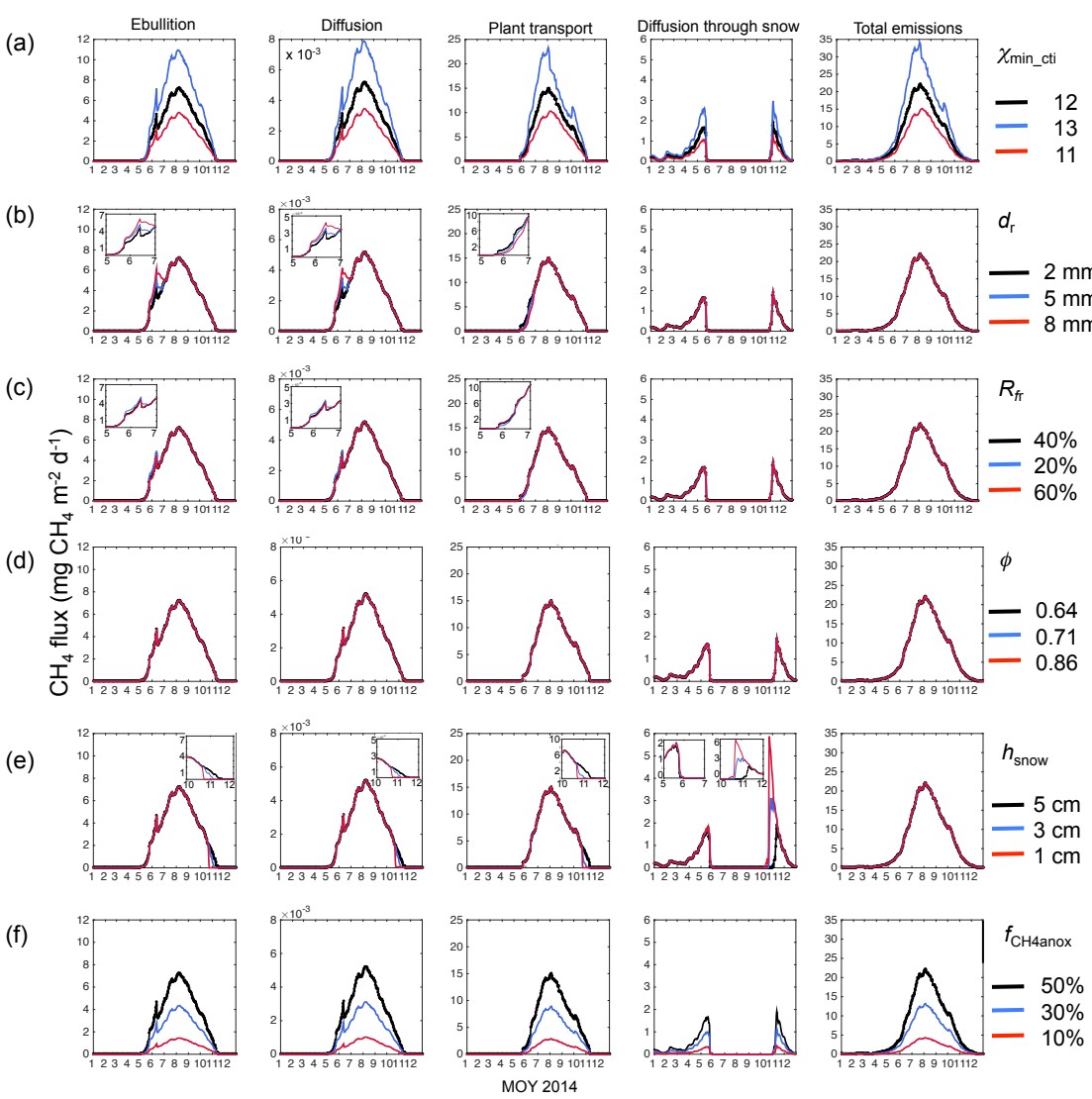


**Figure 3** – Results from the sensitivity experiments for the six selected parameters described in Table 2. Daily methane emissions for the individual transport pathways and total methane emissions are shown. The inset figures in some of the panels are zooms to periods of time where larger differences between signals are depicted.


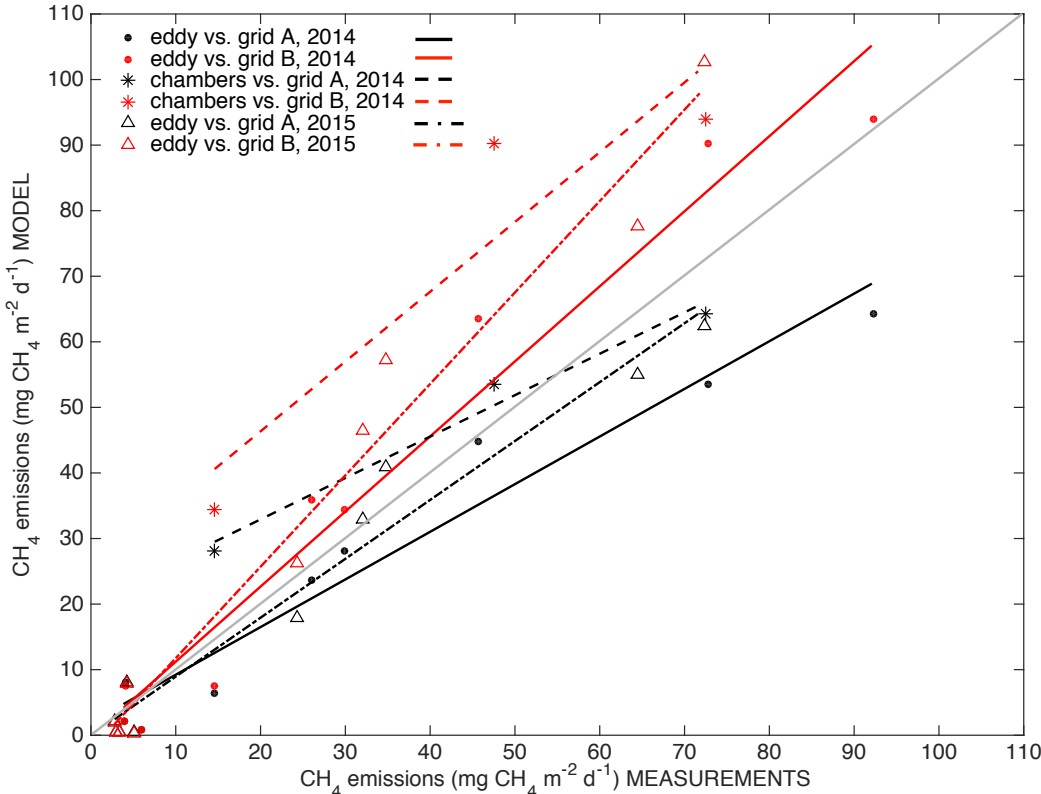

**Figure 4** – Comparison between modeled CH₄ emissions and flux measurements by chambers and EC in the Chersky floodplain: correlation between results for model grid cells A and B and measurements during 2014 and 2015 (the light grey line is the 1:1 line).

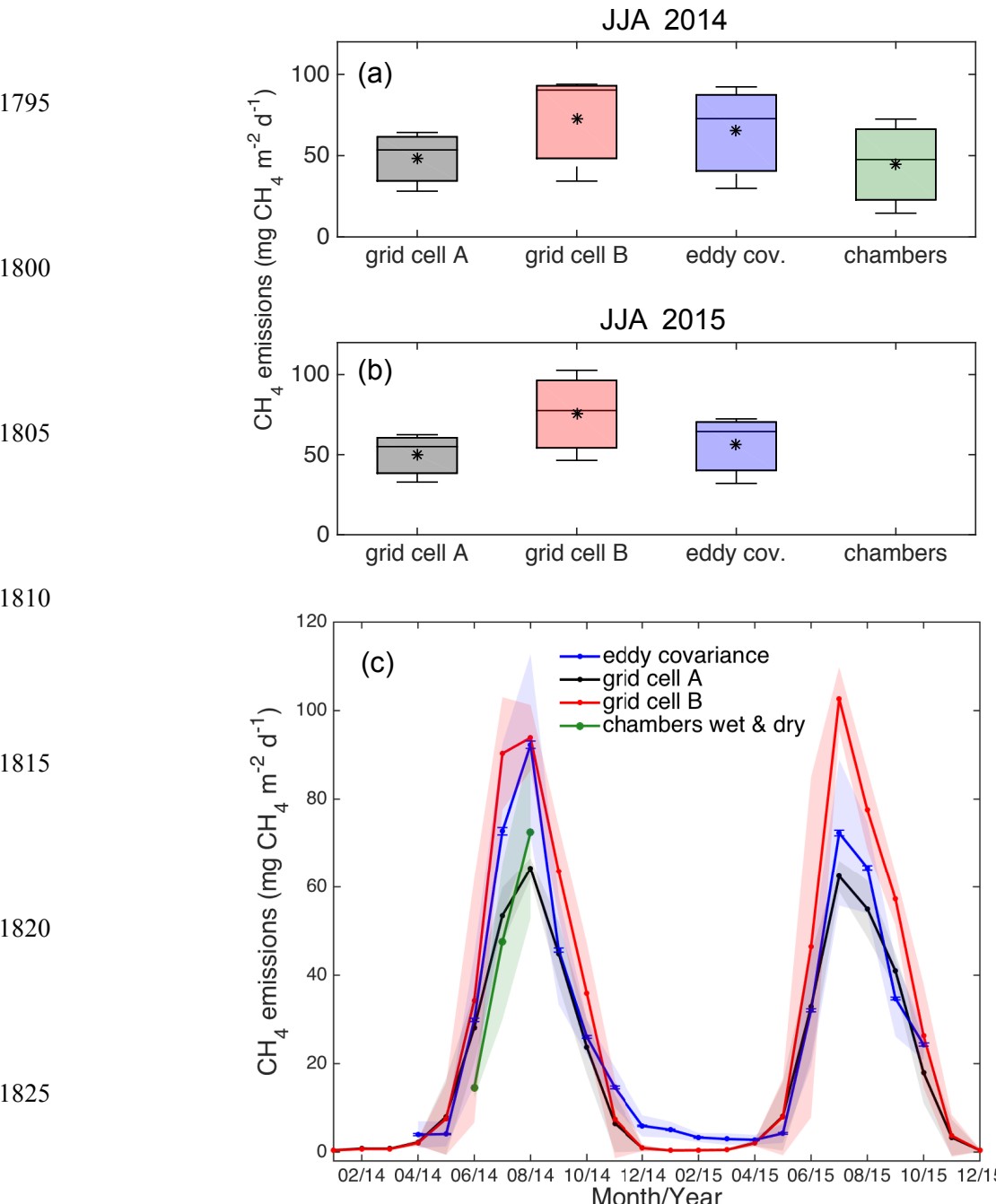

**Figure 5** – Box plot for summer (JJA) methane emissions from model grid cells A and B, eddy covariance and chamber flux measurements for a) 2014 and b) 2015 (without chamber flux measurements). The central horizontal line on each box is the median for each data set and whiskers are the minimum and maximum values; c) time series of monthly $CH_4$ emissions for 2014 and 2015 for grid cells A and B in the model, from eddy covariance as well as chamber flux measurements. Shaded areas depict one standard deviation of the monthly mean of each data set calculated from the daily resolution model output. Error bars in the EC fluxes are the uncertainty of the monthly averages of the gapfilled and quality checked signal.

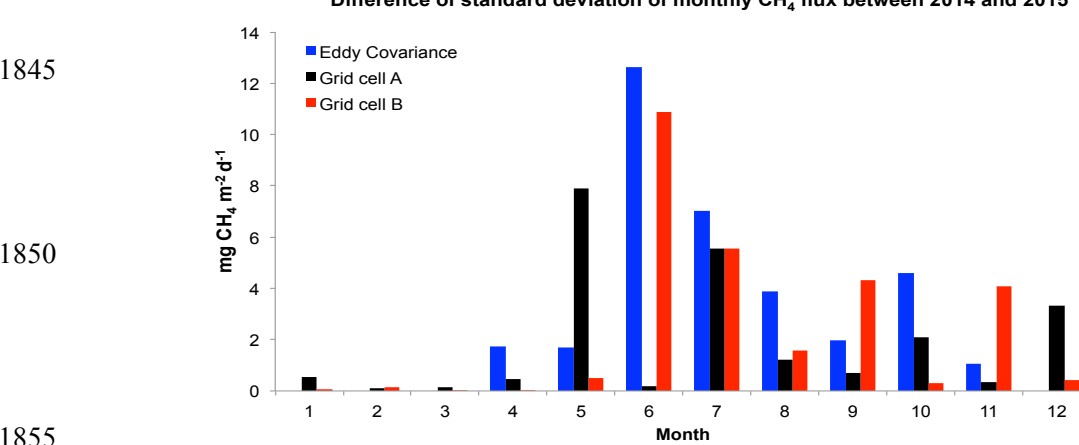

Figure 6 – Comparison of the standard deviation of the monthly fluxes between 2014 and 2015.

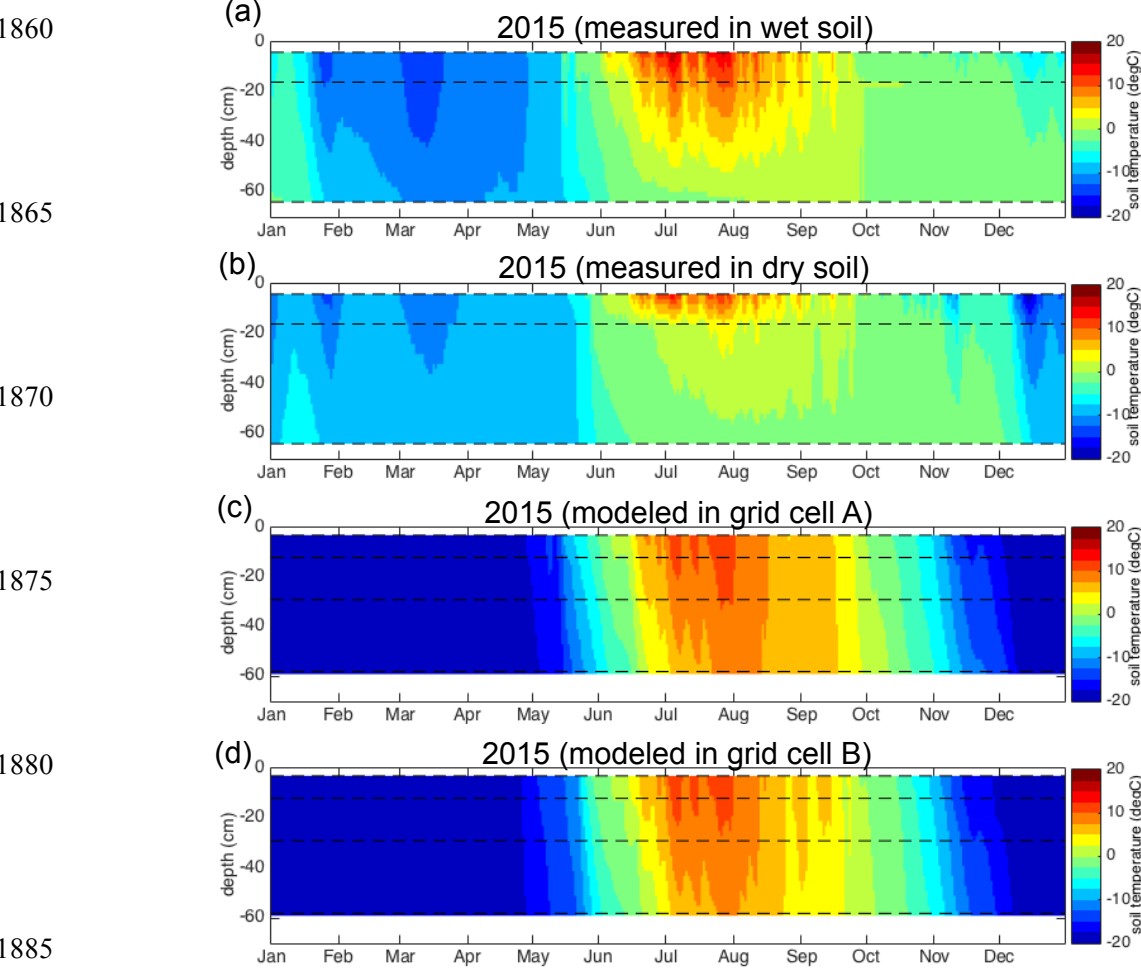

**Figure 7** – Hovmöller diagrams showing the time evolution of the vertical profiles of daily soil temperature during 2015 from eddy covariance fluxes measured a) at the wet plot and b) at the dry plot and from the model data c) grid cell A and d) grid cell B. The data were interpolated linearly from the depths where data is available (4, 16 and 64 cm in the sensors of redox systems and 3, 12, 29, and 58 cm in the model).

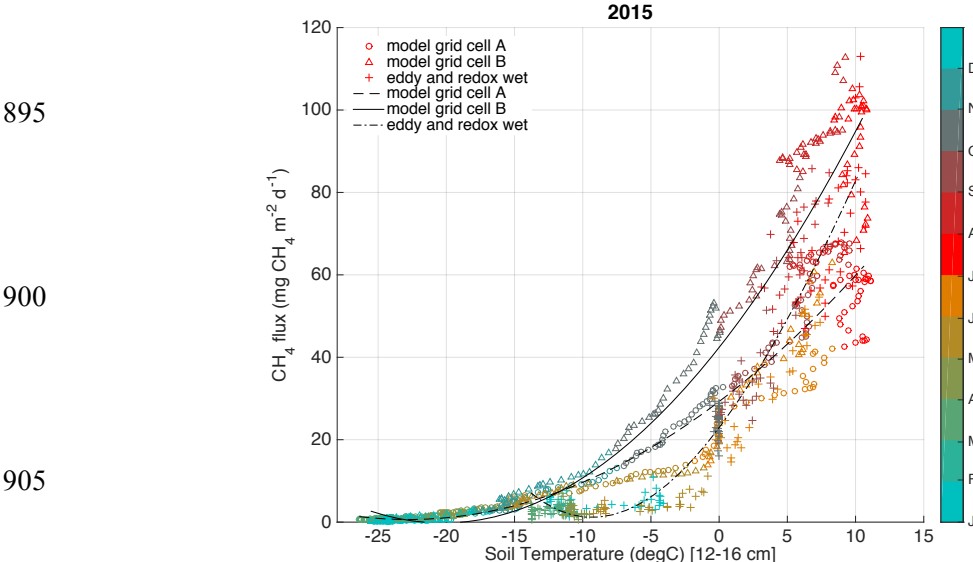

**Figure 8** – Model soil temperatures at 12 cm depth and measured values at 16 cm depth measured at a wet site, against the total methane emissions for grid cell A and B in 2015.

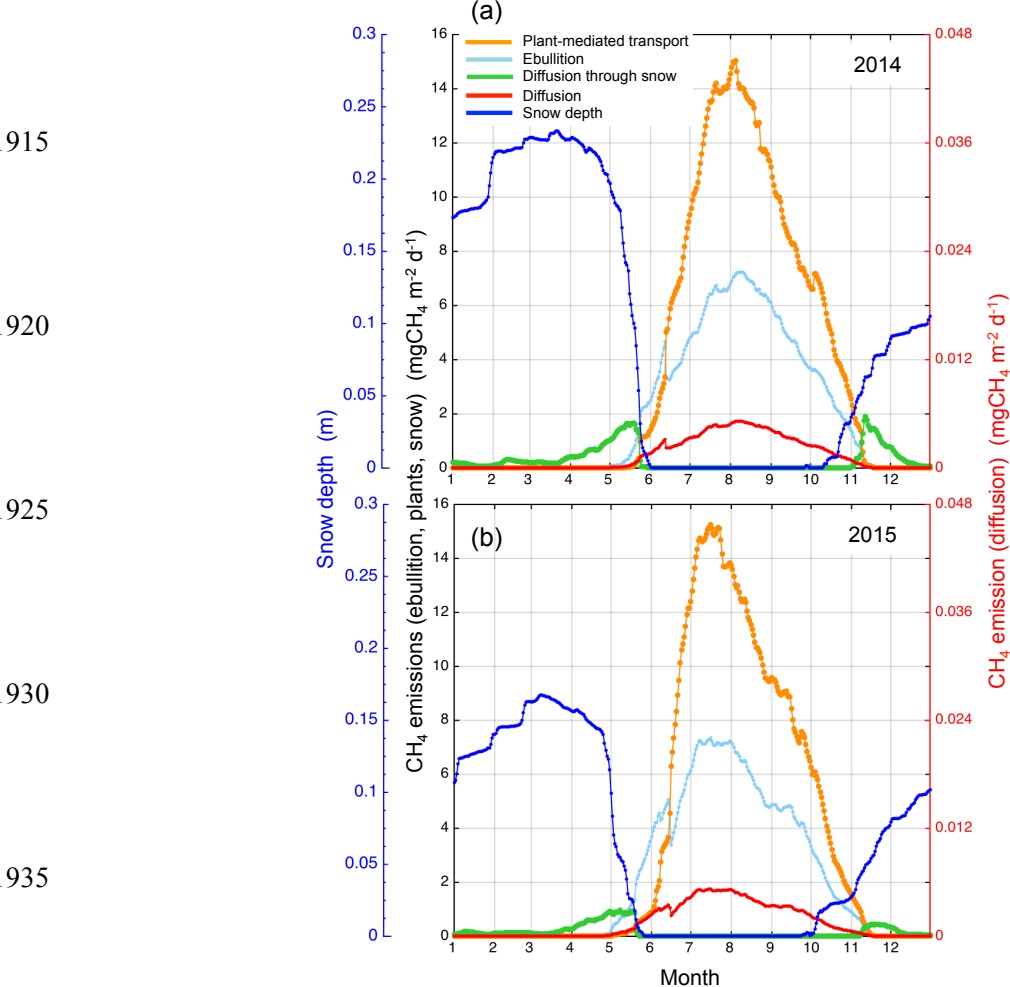

**Figure 9** – Year-round mean simulated CH₄ emissions of the model domain through different pathways and domain mean snow depth for a) 2014 and b) 2015.

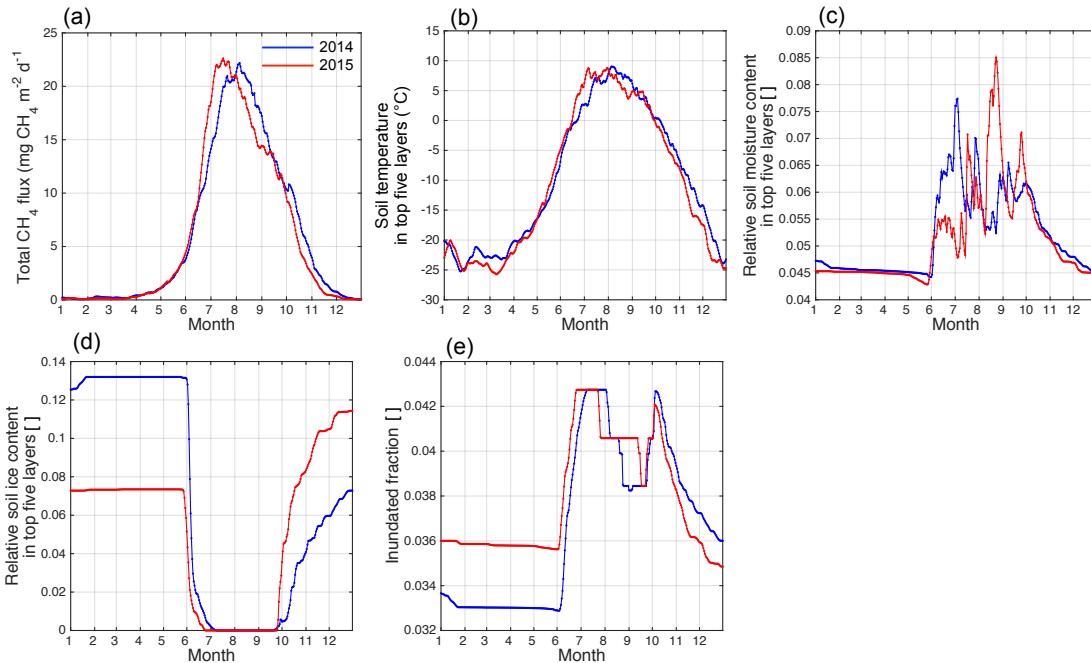

**Figure 10** – Mean daily ancillary variables and CH₄ emissions in the model domain in 2014 and 2015: a) total CH₄ emissions, b) mean soil temperature in the root zone (top five soil layers), c) domain mean relative soil moisture content in the top five soil layers, d) domain mean relative soil ice content in the top five soil layers, e) inundated fraction of the grid cell

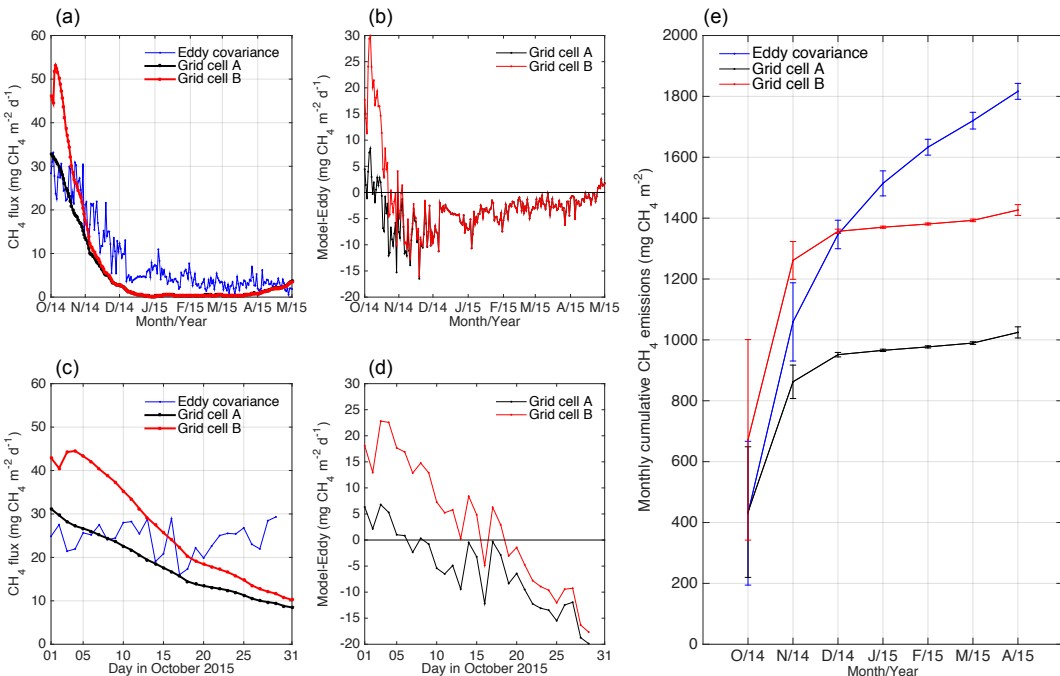

**Figure 11** – Time series of the daily mean of methane emissions through snow from eddy covariance measurements and model data for grid cell A and B during: a) October 2014 to March 2015 and c) October 2015; the difference between grid cell A and B, and the eddy covariance data are shown in panels b) and d) for the same period of time; e) cumulative CH₄ emissions for the period from the end of autumn in 2014 until the end of spring in 2015 for the same data sets. Error bars in each data set are the standard deviation of the monthly-accumulated fluxes.

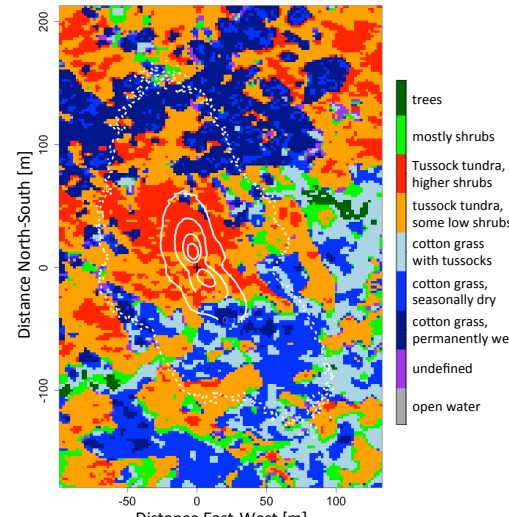

Figure A1: Accumulated source weight function for the EC tower in a control area within the Chersky study site, based on data from the growing season (mid June – mid September) in 2014. Solid white isolines indicate the 80, 60, 40, and 20 % levels, the dashed line is the 10 % level. Background colors indicate aggregated land cover classes based on WorldView-2 data.

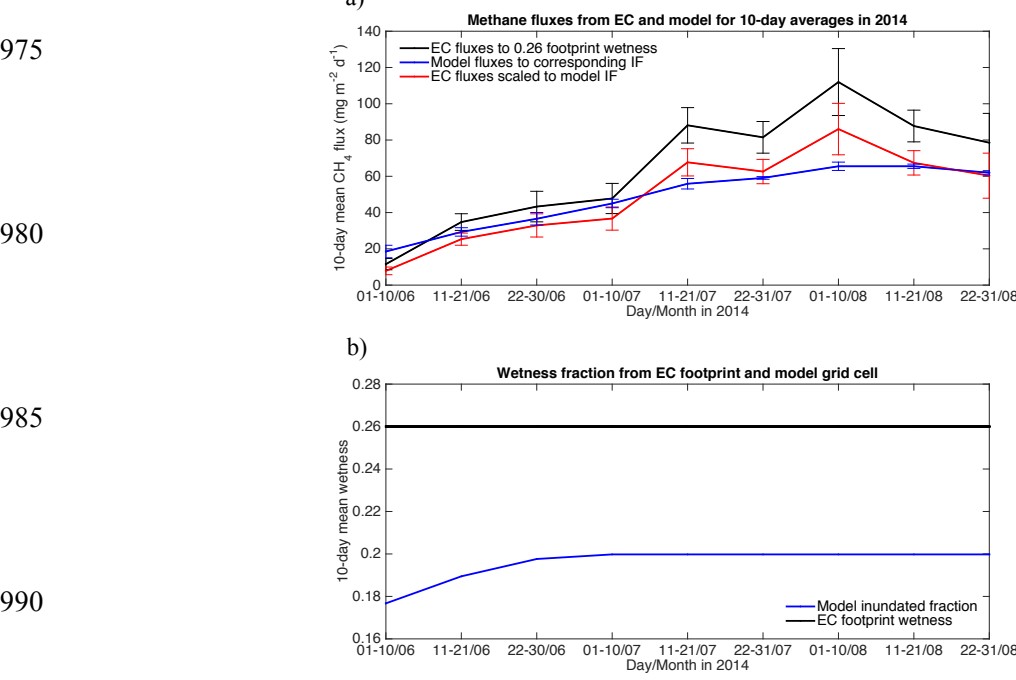

Figure A2 – a) Non-scaled 10-day mean EC methane fluxes representing emissions from a 26 % of wet area in the footprint between June and August 2014 (black line), the 10-day mean EC methane fluxes scaled to the 10-day mean inundated fraction (IF) from the model for the same period of time (red line) and 10-day mean model methane emissions for grid cell A, which imply emissions from the IF from the model (blue line). Error bars in all lines are one standard deviation of the 10-day mean flux values; b) shows the 10-day mean IF from the model used to scale the EC fluxes (blue line), and the constant wetness percentage of the footprint area calculated from the vegetation coverage remote sensing images (i.e. 26 %).

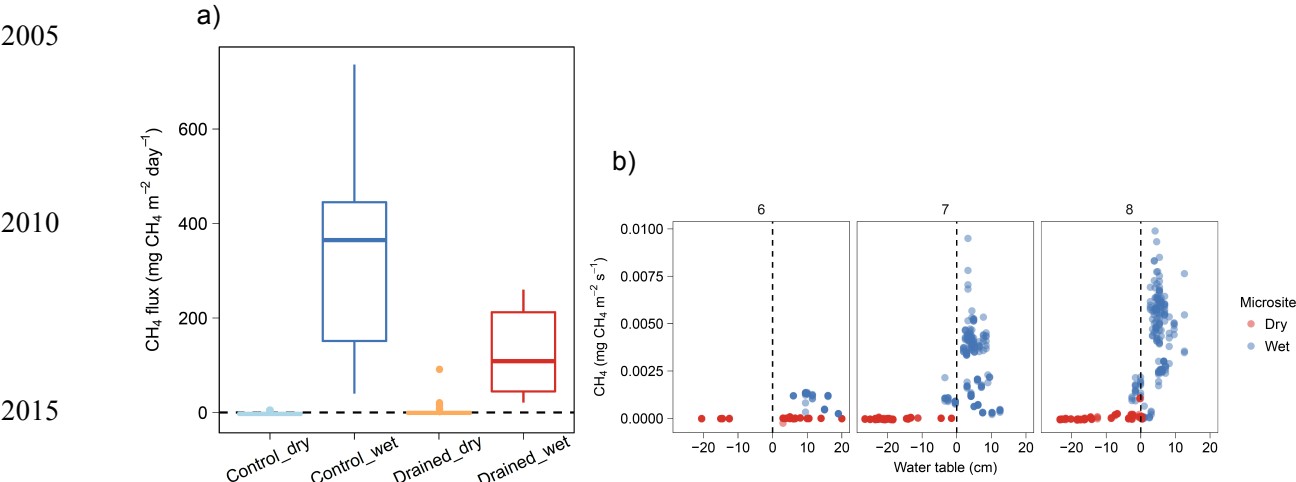

Figure A3 - Daily methane flux rates a) aggregated from flux chamber measurements within the growing season of 2014. Measurements are separated into drained (1 wet microsite, 9 dry microsites) and control (8 wet microsites, 2 dry microsites) transects; b) flux rates against the water table at each microsite. Dry plots had a water table at or below the surface (up to 10 cm), whereas wet plots had a water table at or above the surface.