# Peer review of "Year-round simulated methane emissions from a permafrost ecosystem in Northeast Siberia"

_Biogeosciences, 2017_

## Referee Comment (RC1) · Anonymous Referee #1 · 22 Sep 2017

General comments

The manuscript by Castro-Morales et al. reports simulated methane emissions for a permafrost region in Siberia using an updated version of the JSBACH-methane model. The revised model specifically accounts for (a) diffusion through snow and (b) varying fractions of wetland extent in model grid-cells (and an improved plant-mediated transport scheme). Castro-Morales et al. compare the modelling results to ground-based observations from an eddy covariance tower and from chamber measurements. The manuscript aims to improve current methane models for the permafrost region. Improved model performance for these regions is particularly important as methane

emissions are expected to become more important for the global methane budget in a warming climate. Thus, the manuscript could represent an important contribution to improved modelling of high-latitude methane emissions. The authors present a detailed analysis of the model results, but their results remain often of qualitative nature. The manuscript is mostly well written and discusses in detail many aspects modelling performance. However, a more focused presentation of key results and conclusions could make this manuscript more accessible to the reader. The authors did a thorough job to present and discuss improvements and shortcomings in the performance of the revised methane model. The topic of the manuscript is within the scope of the journal and could be considered for publication. In my opinion, the manuscript would substantially improve if the following issues would be addressed.

Specific comments

In my opinion, the comparison between observed and simulated methane emissions would however benefit from using an upscaling approach to avoid issues arising from the mismatch of scales. This was done for the chamber measurements, but it remains unclear how representative the flux tower footprint is of the entire grid cell. Comparing flux measurements from a single location to the entire grid cell is only meaningful if the grid cell is characterized by spatially homogeneous methane emissions. This is only rarely the case for such high-latitude landscapes (e.g., Sachs et al., 2010; Parmentier et al., 2011; Helbig et al., 2017). The authors should also address how representative the location of tower and chamber flux measurements is of the entire grid-cell. The authors estimate the fraction of inundated land for the grid-cell and demonstrate how this fraction is an important predictor for methane emissions. The same should apply for flux tower measurements where the fraction of wetlands is tightly coupled to the magnitude of methane emissions (see for example Helbig et al., 2017). How would the wetland fraction at the grid cell-scale compare to the same fraction at a smaller scale at the study sites?

The authors report "comparable" (line 30) methane emissions when comparing model

and measurements. The analysis could be much stronger if the authors give a quantitative measure for the performance (e.g, Root Mean Square Error or any other suitable metric).

The authors state that the aim of the work is to "improve our understanding". However, in my opinion, the manuscript mainly focuses on improvements in methane modelling and an evaluation of the performance of a revised methane model. The authors may consider reframing their research objectives and focus results and discussion on the specific research questions.

Large areas in northern Siberia are covered by polygonal tundra. The distinct microtopography of these landscapes has important implications for surface hydrology and thus also surface inundation (see Cresto-Aleina et al., 2013; Helbig et al., 2013; Liljedahl et al., 2016). I was wondering if such polygonal tundra covers a significant proportion of the study area? And if yes, what would be the consequences of distinct microtopography on the performance of the TOPMODEL and on the simulated methane emissions. Using a mean water table for methane modelling in such heterogeneous landscapes can lead to significant underestimation of methane emissions (Cresto-Aleina et al., 2016).

With the TOPMODEL approach, the authors can distinguish between inundated and non-inundated land. However, many peatlands are characterized by a water table just below the peat surface and are thus not inundated. Nevertheless, they can emit large amounts of methane, which would be neglected in the current modeling approach. At the same time, lakes (i.e., inundated land) may be characterized by lower methane emissions than these peatlands due to a lack of fresh organic carbon input. What are the implications of this for the modeling performance? The authors may consider discussing this shortcoming.

In the current manuscript, the authors "decreased or increased [the parameters] by a fixed value" (line 343). Could the authors use a Monte-Carlo approach instead to assess the parameter sensitivity? The authors mention "reported values in the literature". Could they specifically discuss/show the observational constraints on the individual parameters?

Line 406-408: Why do the authors only show one adjacent cell? What is the justification to compare a neighboring grid cell to the ground-based observations? To demonstrate the spatial heterogeneity the authors could consider using more than just two grid cells.

In line 464-465, the authors mention the "parameter adjustment", but do not elaborate how exactly the parameter for the TOPMODEL was adjusted. Did the authors use an objective (cost) function to optimize this parameter?

The authors demonstrate in their sensitivity analysis that the threshold TOPMODEL parameter and "allocation-of-decomposition-to-CH4" are the most important parameters determining the magnitude of simulated methane emissions. In my opinion, the authors should strengthen these results throughout the manuscript. It appears as if their results indicate that methane emissions mainly depend on methane production dynamics (i.e., fCH4anox) and on inundation as "on-off" switch of methane emissions. Transport pathways and methane oxidation appear to be less important (merely changing the timing of emissions). Are these modelling results supported by observations in the field? The authors may consider discussing this in more detail.

Line 61-62: Perhaps the authors could mention another important permafrost thaw effect on methane emissions here: increasing surface wetness due to surface subsidence of ice-rich soils (see for example Christensen et al., 2004; Johnston et al., 2014, Helbig et al., 2017).

Line 94-100: Wintertime methane emissions have also been reported by Helbig et al. (2017) for a boreal peat landscape in northwestern Canada, where they found winter emissions to contribute about 25% to the annual budget.

Line 121: Could the authors discuss here the most important "shortcomings in the

parameterization" of the state-of-the-art methane models?

Line 133: Perhaps the work by Cresto-Aleina et al. (2013, 2016) on microtopography effects on surface water and methane emission dynamics could be mentioned here too.

Line 500-501: Only mineral soils are considered for the methane modelling? How common are organic soil in the study area? I would assume that at least top-soils in the floodplain would be organic-rich. How would "considering" organic soils change the results?

Line 577-579: The authors may consider supporting this statement with information on the exact magnitude of interannual variability.

Line 589-592: What is the uncertainty in the eddy covariance flux measurements? Could the authors quantify uncertainties due to random errors, gap-filling, u*-threshold, and footprint heterogeneity? An uncertainty quantification of eddy covariance fluxes would further strengthen the model-observation comparison.

Line 691-711: I am not sure how this section contributes to the research questions of this manuscript? Perhaps the authors could mention differences in environmental characteristics of grid-cell A and B briefly in the manuscript and move figure 9 to the supplementary material?

Line 808-810: The impact of cooler early summer temperatures on soil warming and methane emissions has been demonstrated recently using multi-year methane observations in a boreal peat landscape (see Helbig et al., in press). The authors may consider discussing their modelling results in relation to these observations.

Line 847-851: The authors may consider starting the discussion mentioning the parameters that actually made a difference and not with the parameters that did not change the results. It should be highlighted what process/parameter matters in the model.

Line 991-992: Few studies have shown that non-inundated upland areas may take

up methane (e.g., Flessa et al., 2008). As far as I understand, such uptake is not considered in the current work. How could uptake in the drier areas of the model domain change simulation results? There are large areas in the model domain that appear to be characterized by upland landscapes and thus potential methane uptake (see Fig. 1).

Line 1134-1141: The authors may consider not to introduce a new concept (e.g., anaerobic microsites) at the very end of the conclusions. I would recommend to only refer here to what has been shown in the manuscript so far.

Line 1252-1255: What would happen if the model would run with the old order of processes? Shouldn't this be part of the uncertainty analysis?

Fig. 1: Why did the authors use such a large study area, if ground-based observations were only available for a very small fraction of the model domain? How can the model performance be evaluated for the other non-floodplain grid cells that appear to be characterized by different landscape characteristics?

Fig. 6: Why do the authors compare the mean grid-cell soil temperature profile to measured wet and dry soil temperature profiles? Physical soil properties differ drastically between wet and dry soils and consequently strongly determine soil temperature dynamics (see end of discussion). Wouldn't it be therefore necessary to at least model soil temperature dynamics of the inundated and non-inundated land surface separately?

Fig. 7: Methane emissions increase considerably in the model at sub-zero soil temperatures. In contrast, measured methane emissions appear to be quite insensitive to soil temperature below 0°C. The authors mention this mismatch in lines 655-659. Perhaps the authors can discuss this mismatch between temperature-emission responses in more detail. How is it possible that such cold simulated soil temperatures result in emissions of > 30 mg CH4 m-2 day-1?

Fig. 8: Here, an uncertainty estimate for the measured cumulative methane emissions would help interpreting the comparison between simulated and measured fluxes.

Fig. 11: I am not sure how this figure contributes to the research questions. The seasonality of different methane emission pathways is already shown in Fig. 10. How does a representation of the spatial distribution of the methane emissions add to the manuscript?

Technical comments

Line 149: Remove "done".

Line 150: Remove "are".

Line 196: Please define what "hospitable and inhospitable" land means in this context.

Line 534: What do the authors mean with "visually"? They state in the previous sentence that differences are not statistically significant.

Fig. 3: Please clarify what the inset figures show.

Reference:

Christensen TR, Johansson T, Åkerman HJ et al. (2004) Thawing sub-arctic permafrost: Effects on vegetation and methane emissions. Geophysical Research Letters, 31, L04501.

Cresto-Aleina F, Brovkin V, Muster S, Boike J, Kutzbach L, Sachs T, Zuyev S (2013) A stochastic model for the polygonal tundra based on Poisson – Voronoi diagrams. Earth System Dynamics, 4, 187–198.

Cresto-Aleina F, Runkle BRK, Brücher T, Kleinen T, Brovkin V (2016) Upscaling methane emission hotspots in boreal peatlands. Geoscientific Model Development, 9, 915–926.

Flessa H, Rodionov A, Guggenberger G et al. (2008) Landscape controls of CH4 fluxes

in a catchment of the forest tundra ecotone in northern Siberia. Global Change Biology, 14, 2040–2056.

Helbig M, Boike J, Langer M, Schreiber P, Runkle BRK, Kutzbach L (2013) Spatial and seasonal variability of polygonal tundra water balance: Lena River Delta, northern Siberia (Russia). Hydrogeology Journal, 21, 133–147.

Helbig M, Chasmer L, Kljun N, Quinton WL, Treat CC, Sonnentag O (2017) The positive net radiative greenhouse gas forcing of increasing methane emissions from a thawing boreal forest-wetland landscape. Global Change Biology, 23, 2413–2427.

Helbig M, Quinton WL, Sonnentag O (in press) Warmer spring conditions increase annual methane emissions from a boreal peat landscape with sporadic permafrost. Environmental Research Letters. doi: 10.1088/1748-9326/aa8c85.

Johnston CE, Ewing S a, Harden JW et al. (2014) Effect of permafrost thaw on CO2 and CH4 exchange in a western Alaska peatland chronosequence. Environmental Research Letters, 9, 85004.

Liljedahl AK, Boike J, Daanen RP et al. (2016) Pan-Arctic ice-wedge degradation in warming permafrost and influence on tundra hydrology. Nature Geoscience, 9, 312–318.

Parmentier FJW, van Huissteden J, van der Molen MK, Schaepman-Strub G, Karsanaev SA, Maximov TC, Dolman AJ (2011) Spatial and temporal dynamics in eddy covariance observations of methane fluxes at a tundra site in northeastern Siberia. Journal of Geophysical Research, 116, G03016.

Sachs T, Giebels M, Boike J, Kutzbach L (2010) Environmental controls on CH4 emission from polygonal tundra on the microsite scale in the Lena river delta, Siberia. Global Change Biology, 16, 3096–3110.

---

## Referee Comment (RC2) · Anonymous Referee #2 · 20 Nov 2017

Overall:

The ms has its focus on regional scale methane dynamic and the modelling of year round dynamics, which is certainly relevant and highly needed. I general there are quite few year round measurements of methane dynamics in the arctic region, which also explains why the modelling studies are even fewer and regional budgets are poorly constrained. Further, the understanding of drivers and exact transport mechanisms in the top soil and soil – snow- atmosphere still in most (not all) cases relies on an interpretation of a net emission, rather that independent quantification of the individual components adding up the net CH4 emission. For that reason the focus of the current

ms is important and timely. Despite that the ms is well written and in general well references, I'm a bit reluctant about the qualities of the ms, because I basically find that it tries to accomplish too much and not in a fully convincing way. As pointed out by reviewer 1, also I have a serious problem with the differences in scaling which are used in the different components of the study. In my perspective, the very coarse spatial scale of the model does not compare well with the highly advanced model approach of partitioning the production and transport of CH4 in the soil and snow. The ms simultaneously tries to solve the issues of the spatial /temporal methane dynamics of the large Sibirian wetlands, the process pathways and comparing all the modelling output to relatively few and very local measurements near Chersky. I basically don't think that the available measurements are well suited to verify the model output of the processes leading to the net CH4 emission at the surface, and the differentiation of pathways of CH4 during different periods of the year. I my opinion the ms could benefit from being divided into two; one with focus on the annual budget for Sibirian and one focused on the process modelling of the different pathways for methane through the soil/snow pack. The later one could benefit from some kind of lab or micro cosmos comparison, where processes could be studied more precisely than what is mostly the case in the field. Regardless of the approach, the issues of differences in scales should be discussed much more detailed and qualified than it is done in the present version of the ms. From my perspective the output of the model and the assessment of the advances in the new "improved" version is not credible as it appears now, despite that the output is in the same ballpark as the measured data, and a number of other studies.

Specific:

L48 -> 66: Maybe a matter of taste, but I'm in general against using these "horror scenarios" which draw lines between the carbon pool of the Arctic soils and potential increase of GHGs. I think we now know that no indications are found that something very dramatic is happening in foreseeable future, and it doesn't add to the understanding of the ms. Consider rephrasing.

L187: Despite that you are obviously aware of the complications of the comparison between scale I'll encourage you to address specifically how the scaling issue between 0,5° modelling grid and EC footprint or chambers is dealt with.

L218: Again please justify, why 11 soil layers are needed, when the horizontal scale is this coarse.

L323: Spun upo for 10.000 years? Please justify further, climate (or C – pools) can not be assumed to have remained constant for this period of time.

L403: I understand that the numbers can be compared, but please argue why the field site measurements can be assumed to be averaging the full 0,5x0,5° modelling pixel.

L445: Differences seems to be substantial please comment.

L470: I basically don't understand how a threshold can be set for proportion of flooded area in a pixel – what is the rational ? Theoretically the whole pixel could be inundated –I assume?

L532: What effect of the snow would you have expected in this context?

L630: there seems to be significant difference in measured and modelled soil temperatures, please comment.

L665: probably why also both absolute values and seasonal pattern seems distinctly different

L710 -723: that differentiation between ebullition and diffusion seems unfounded, and it is hard to see how you verify the different pathways, please elaborate.

Para 3.4.3: could this be merged with the sensitivity study in 3.2? seems to be fundamentally alike.

Fig. S5b: legend does not seem to match.

L916-920: the conclusions here seem somewhat unfounded due to the previously mentioned scaling issues.

---

## Author Comment (AC1) · 21 Jan 2018

**Answer to Reviewer #1**

Thank you very much for your comments and support for the publication of our manuscript. Below we address one by one the comments made during this review. All answers are in blue font.

**Specific comments**

In my opinion, the comparison between observed and simulated methane emissions would however benefit from using an upscaling approach to avoid issues arising from the mismatch of scales. This was done for the chamber measurements, but it remains unclear how representative the flux tower footprint is of the entire grid cell. Comparing flux measurements from a single location to the entire grid cell is only meaningful if the grid cell is characterized by spatially homogeneous methane emissions. This is only rarely the case for such high-latitude landscapes (e.g., Sachs et al., 2010; Parmentier et al., 2011; Helbig et al., 2017).

We agree that the comparison between model methane fluxes and those from observations, specifically from eddy covariance, is a challenge. In our manuscript, we use a scaling factor for the chamber data by considering chamber measurements that were done under exclusively wet and under exclusively dry summer conditions. We then make use of the total fraction of inundated areas in the model grid cell (IF) modeled with the TOPMODEL approach to scale the total chamber fluxes. This scaling approach takes into consideration that the model methane fluxes represent the emissions from only the portion of the grid cell that is inundated, i.e. with water at or above the soils surface.

In the case of the eddy covariance fluxes, following the concerns of the reviewer, we re-evaluated our approach for this comparison. In the revised version of this manuscript we include now a thorough analysis of the footprint area of the eddy covariance fluxes as part of a new Appendix B on "Details on in-situ flux observations". This appendix also includes details on the eddy covariance flux data uncertainty assessment and more detailed results on the chamber measurements, as requested below also by the reviewer. This appendix will be part of the revised manuscript and is attached at the end of this response.

In this new appendix, we analyze the type of vegetation and its coverage in the footprint area of the EC tower, from remote sensing images as a metric to identify wet and dry areas. Areas with dominant cotton grasses, specifically *Eriophorum angustifolium* in our study area, are indicators of predominant wet soils, while tussocks, specifically *Carex appendiculata* in our study area, and shrubs are indicators of predominant dry soil conditions. It is important noting that *C. appendiculata*, can be also found in wet areas, but is predominant in dry areas.

For the model, the vegetation distribution per grid cell is too coarse to consider this metric similar as that for the remote sensing data in the EC footprint area, however the total abundance of C3 grasses in the grid cell A is 33.3 % as given for the model (with the rest of the grid cell dominated by deciduous shrubs and extra tropical evergreen trees), but there is no discrimination between cotton grasses and tussocks.

The footprint of the eddy covariance tower in the Chersky floodplain covers an approximate area of 400 m x 400 m, similar to that one depicted in Fig. 1 of Kittler et al. 2016 (cited in discussion ms) (see new Appendix B at the end of this response for footprint area for the EC tower used in this manuscript). The remote sensing analysis revealed that cotton grasses are present in about 26 % of the footprint area, which would translate into the same portion of the footprint area as fully wet zones during the "wet months": after spring melt in June and until August when most annual precipitation in the region takes place, covering most of the growing season. As will be shown below in this response, $CH_4$ fluxes measured by chambers (footprint of 60 cm x 60 cm) revealed that during the growing season in dry soil areas of the Chersky floodplain that are characterized by a water table below the surface, the emission of methane during the growing season is negligible with even some atm. $CH_4$ uptake by soil (i.e. negative $CH_4$ flux rates) (data shown in new Appendix B). Under this consideration, and as confirmed recently by Helbig et al., 2017, the majority of the $CH_4$ fluxes measured by the EC tower would represent fluxes from fraction of wetland in the footprint area, i.e. 26 %.

In case of the model grid cell where the location of the EC tower falls (grid cell A in Fig. 1 of the discussion ms), the IF for June-July-August during 2014 shows growing inundation values from 17.7 % to 19.9 % (for 10-day mean values for those three months) representing the percentage of total wet areas in the grid cell area. These values are slightly smaller than the 26 % wetness area in the EC footprint, and denote the area of the grid cell where the model methane emissions take place (i.e., no emissions in dry areas, in agreement to the chamber measurements).

With this basis and to make a closer comparison between EC flux measurements and model data for the growing season months, we scaled linearly the 10-day mean EC methane fluxes to the IF from the model, and calculated the standard deviation of the 10-day mean. In the next figure, we show: TOP panel, the original 10-day mean EC methane flux measurements that would represent the emissions of a 26 % wet area between June and August 2014 (black line), the 10-day mean EC methane fluxes scaled to the 10-day mean IF from the model for the same period of time (red line) and 10-day mean model methane emissions for grid cell A, which imply emissions from the IF from the model (blue line). Error bars in all lines are one standard deviation of the 10-day mean flux values. The BOTTOM panel shows the 10-day mean IF from the model used to scale the EC fluxes (blue line), and the constant wetness percentage of the footprint area calculated from the vegetation coverage remote sensing images (i.e., 26 %).

[Figure]

We observe that the scaled EC methane fluxes decreased as a lower IF is considered within the footprint, and those new calculated fluxes become closer to those from the model, and in most cases the latter fall within the 10-day standard deviation of the EC fluxes.

Unfortunately, it is not possible to obtain a temporal varying wetness area for the EC footprint all year, based on our approach of only considering the vegetation cover, thus wouldn't be appropriate to scale all of the EC fluxes for 2014 and 2015 to the IF from the model without any reference for spring and winter wet footprint areas. However, from this analysis we learn that: 1) considering the vegetation cover as indicator of soil wetness, the EC footprint area holds a very similar area to that of the model grid cell through which the majority of the methane is emitted to the atmosphere and 2) the net offsets between methane flux model and EC data can largely be attributed to differences in wetness levels.

Summarizing, we assume that for both the model grid cell and the eddy covariance footprint, methane emissions are not spatially homogeneous, but bound to the distribution of wet (inundated) areas. Accordingly, a meaningful agreement between model and observations can only be obtained if two factors are fulfilled: (i) the fraction of wet surfaces agrees between both data sets, and (ii) the flux rates from wet surfaces agree between both datasets. Through correcting the offsets in inundated fraction, we could demonstrate that the flux rates between model and eddy covariance observations agree very well, emphasizing the sound setup of the

model algorithms and parameter settings. We will add the analysis presented here into the new Appendix B to complement the discussion on scaling fluxes for comparison between EC and model data.

The authors should also address how representative the location of tower and chamber flux measurements is of the entire grid-cell. The authors estimate the fraction of inundated land for the grid-cell and demonstrate how this fraction is an important predictor for methane emissions. The same should apply for flux tower measurements where the fraction of wetlands is tightly coupled to the magnitude of methane emissions (see for example Helbig et al., 2017). How would the wetland fraction at the grid cell-scale compare to the same fraction at a smaller scale at the study sites?

We approached this comment with the answer above. By evaluating the vegetation cover types within the footprint area of the EC tower, we identified the wet areas and assume that the methane fluxes measured with this tower represent the emissions from the wetlands within the footprint. Equivalent to the grid cell area, the inundated fractions determined with the TOPMODEL approach, represent the areas where methane is emitted at grid cell scale. Those are comparable and to show this, a scaling exercise for growing season methane emissions in 2014 was presented above.

The authors report "comparable" (line 30) methane emissions when comparing model and measurements. The analysis could be much stronger if the authors give a quantitative measure for the performance (e.g, Root Mean Square Error or any other suitable metric).

As suggested by the reviewer, we include now in the revised ms the relative RMSE calculation in percentage (e.g. RMSE / mean(CH4_obs) * 100) between model and flux measurements from Eddy Covariance (for 2014 and 2015) and chambers (only for the available three months in 2014). We calculated this error on a monthly basis, using the daily resolution fluxes. Results are shown in the table and figure below.

| Month | Rel. RMSE (%) (model – EC) 2014 | Rel. RMSE (%) (model – chambers) 2014 | Rel. RMSE (%) (model – EC) 2015 |
|-------|---------------------------------|----------------------------------------|---------------------------------|
| Jan | - | - | 99.3 |
| Feb | - | - | 91.9 |
| Mar | - | - | 92.4 |
| Apr | 76.1 | - | 60.7 |
| May | 106.9 | - | 103.6 |
| Jun | 26.9 | 47.8 | 17.1 |
| Jul | 33.0 | 14.2 | 24.7 |
| Aug | 36.7 | 10.5 | 18.3 |
| Sep | 16.2 | - | 26.6 |
| Oct | 24.1 | - | 36.6 |
| Nov | 60.5 | - | - |
| Dec | 91.4 | - | - |

[Figure]

The relative RMSE results show the relative variation between the model and the observations. A larger variation is observed in the first five and last month of the year (winter and spring) between model and measured EC fluxes, while the lowest variations are observed during the growing season and autumn (June to October). The summer variation is larger in 2014 between model and EC data and lowest in July and August between the model and chamber measurements in 2014. This information will be included in the revised MS to quantitatively support the evaluation of the model results.

The authors state that the aim of the work is to "improve our understanding". However, in my opinion, the manuscript mainly focuses on improvements in methane modeling and an evaluation of the performance of a revised methane model. The authors may consider reframing their research objectives and focus results and discussion on the specific research questions.

The reviewer is correct that the stated aim is not reflecting the bottom line of our manuscript. Following this suggestion, we reframed the aim to be clearer and now it reads: "The aim of this work is to analyze the performance of an improved process-based methane model, designed for Arctic tundra and wetlands underlain by permafrost, when applied to a regional domain in Northeast Siberia. Our intention is to evaluate the potential of a refined process-based methane model as a proof of concept, for its application to a larger than site level scales. For this, year-round $CH_4$ emissions are modeled and differentiated among distinct pathways: plant-mediated, ebullition, and diffusion." We also focus the discussion towards this aim in the revised ms.

Large areas in northern Siberia are covered by polygonal tundra. The distinct microtopography of these landscapes has important implications for surface hydrology and thus also surface inundation (see Cresto-Aleina et al., 2013; Helbig et al., 2013; Liljedahl et al., 2016). I was wondering if such polygonal tundra covers a significant proportion of the study area?

And if yes, what would be the consequences of distinct microtopography on the performance of the TOPMODEL and on the simulated methane emissions. Using a mean water table for methane modelling in such heterogeneous landscapes can lead to significant underestimation of methane emissions (Cresto-Aleina et al., 2016).

The reviewer is right that a good portion of the Siberian tundra is characterized as polygonal tundra. However, our area of study does not contain these particular micro-topographic structures since it is mostly located in a floodplain that naturally becomes inundated at the end of the melt season (spring). Towards summer, most of the water recedes to streams and to the Kolyma River and nearby tributaries only to lead to a typical wetland landscape. Still, some polygonal structures are present, but they are not a dominant feature of the landscape within our model domain, as opposed to e.g. the Lena River delta. Therefore, the application of TOPMODEL in the Chersky floodplain is suitable and there is no need to consider polygonal structures.

With the TOPMODEL approach, the authors can distinguish between inundated and non-inundated land. However, many peatlands are characterized by a water table just below the peat surface and are thus not inundated. Nevertheless, they can emit large amounts of methane, which would be neglected in the current modeling approach.

We are aware of the limitations on the use of TOPMODEL in those particular cases where the water table is located below the surface and those were discussed briefly in the discussion manuscript. The study of Kwon et al. (2016) (cited in the discussion ms) reported the flux chamber measurements in the same Chersky floodplain site subject to our study. The authors reported $CH_4$ fluxes measured in plots where the water table was 10 cm below the surface and found negligible contribution of $CH_4$ from these soils. Specifically, areas with water table 5 cm below the surface showed net $CH_4$ emissions but flux rates were not as high as in areas with standing water (see figure below). Also, as shown in the new Appendix B attached at the end of this response, chamber flux measurements of $CH_4$ in dry soils with water tables ca. 10

cm below the surface show none or negligible methane emissions to the atmosphere in this area of study. Taking into account these findings, in our model configuration the fact that no methane emissions take place in dry soils, would not pose a constraint to the total modeled methane fluxes per grid cell in this area of study, however, the role of methane oxidation could be better evaluated.

[Figure]

The figure above shows results of $CH_4$ chamber fluxes (mg $CH_4$ m$^{-2}$ s$^{-1}$) measured from June to August (numbers at the top of the figure indicate the month of the year: 6 is June, 7 is July and 8 is August) in 2014 for the dry and wet plots and their corresponding water table (in x-axis). "Dry" plots have mostly water tables at or below the surface during July and August with mostly uptake of $CH_4$ from the atmosphere (on average 3 mg $CH_4$ m$^{-2}$ d$^{-1}$), whereas the wet plots characterized by water tables located above the surface, showed average emissions of 332 mg $CH_4$ m$^{-2}$ d$^{-1}$ over the same period of time.
Despite this agreement, we are aware that the low CH4 uptake in dry areas might not apply to other tundra areas, e.g. in Zona et al., (2016) in the Alaskan tundra the highest fall and winter $CH_4$ fluxes were observed in upland tundra sites characterized by having a water table below the surface during summer. In future studies, our model scheme should also be tested in other areas such the Alaskan tundra to assess and improve further the model configuration especially in the TOPMODEL scheme.

At the same time, lakes (i.e., inundated land) may be characterized by lower methane emissions than these peatlands due to a lack of fresh organic carbon input. What are the implications of this for the modeling performance? The authors may consider discussing this shortcoming.
This is an interesting idea and we agree with the reviewer that a comparison of lake and peatland model results would be an ideal evaluation of our methane scheme using extreme cases of water table depth. However, we do not see the possibility to perform such study, as explained below.
In our model configuration, the production of methane is considered to take place in mineral soils and does not include peatlands as definition: the layer of soil with > 30 cm of organic rich material (peat) accumulation. A mask containing the distribution of peatlands should be needed to introduce this feature. In addition, as carbon decomposition slows down in permanently anoxic areas of the soil column, the prescribed mask of peatlands should contain added soil C in order to describe deep peat layers characterized by a slow decomposition timescale. These steps are currently been taken for the global context with the JSBACH model and are still pending work for high horizontal-resolution domains such as the regional one presented in this work.
The scheme to model wetland areas using the TOPMODEL approach considers the topographic profile, which is provided as a prescribed compound topographic index in the model domain, and methane emissions take place in areas where the water table is located at or above the soil surface. In this context, the model does not explicitly simulate the location

of "lakes" (inland open water bodies) but rather a dynamic change in the horizontal distribution and accumulation of water at or above the surface, which in turn may consider implicitly inland water bodies at different scales: lakes, wetlands, ponds, etc. With this model, it is not possible to discriminate at this coarse resolution, the type of water bodies, but rather it provides an average portion of the grid cell area where inundation can take place, and only the methane production and ultimately emissions, are linked to the carbon content and environmental conditions of the soil. If by definition there is no consideration of peatlands in our model, in the end all goes down to the available organic carbon in the soil for the production of methane. As requested by the reviewer, we will discuss this shortcoming in the revised manuscript.

In the current manuscript, the authors "decreased or increased [the parameters] by a fixed value" (line 343). Could the authors use a Monte-Carlo approach instead to assess the parameter sensitivity?

The purpose of the parameter permutation is to know, to which parameter the model is most sensitive, as this identifies which parameter need to be better constrained to reduce model uncertainty. The purpose of a Monte-Carlo approach is the identification of the uncertainty of a model given a known probability distribution function of parameter values for a combination of parameters. MC approaches are not primarily designed to identify model sensitivities to specific parameters, even though some approaches such as LHS allow interpreting MC approaches in terms of model sensitivities; however only at very high computational costs. One-at-a-time schemes (OAT) as the one applied here directly target the model sensitivity, are computational cost efficient, and are deemed fully sufficient for the purpose (Saltelli et al. 2000). The identification of compensating effects between parameters or non-linear effects, which would require an MC approach area, is beyond the scope of this paper. We therefore consider an MC approach to assessing model sensitivity as unnecessary.

The authors mention "reported values in the literature". Could they specifically discuss/show the observational constraints on the individual parameters?

We thank the reviewer for this comment. We improved the description of our selection of parameters for the sensitivity study in the revised ms. The selected parameters are those that are prescribed in the model and are considered uncertain. Specifically, the selected values for $\phi$ (snow porosity) and $f$CH4anox (fraction of anoxic decomposed carbon that becomes methane) were those kept within ranges of values previously discussed in the published literature, whereas for the other four selected parameters (see below) we chose a range of values around the defined values for the control simulation.

Thus, the selected parameters are characterized by at least one of the two criteria: 1) it is a parameter with large uncertainty because it is not provided in current published literature or its values are still controversial as reported in published literature, and 2) it is possible to test a range of values based on reported values in literature. The last criterion is only true for two of the selected parameters ($\phi$ and $f$CH4anox) as mentioned above. As given in the discussion ms, the six selected parameters for our sensitivity studies are:

In the TOPMODEL scheme:

1) $\chi$min_cti, minimum compound topographic index threshold value. This parameter fulfills criterion 1 since it is a model parameter that is exclusively part of the TOPMODEL scheme, therefore there is no literature reference and rather is a given value that has to be adjusted.

In the plant-mediated transport scheme:

2) $d$r, root diameter. This is a highly uncertain value in literature with only few reported values. Few studies have reported the diameter of vascular plants in boreal ecosystems. In Wania et al., 2010 (cited in discussion ms and after Schimmel, 1995) the authors report a diameter for *Eriphorum angustifolium* of 3.95 mm, while for *Carex aquatilis* a value of 3.8 mm. Chapin and Slack (1979) reported a diameter for *Eriophorum vaginatum* of 0.8 mm, while Wang et al., 2016 reported a value of 1 mm for the same species. For our model set up, we use a value of 2 mm in the control run

considering an average value between those reported in the literature. For the sensitivity study, we selected higher root diameters experiments: 5 and 8 mm.

3) *Rf*r, principal fraction of the pore-free soil volume occupied by roots. This is also a highly uncertain value that is not reported in literature; therefore, we assume in our control experiment a fraction of 40 % (i.e., in a certain volume of soil, 40 % is occupied by plants roots). For the sensitivity studies, we decreased and increased this reference value by 50 % of the control value, i.e., 20 % and 60 % respectively.

In the diffusion of gas through snow:

4) $h_{snow}$, snow depth threshold. The studies of Pirk et al., (2016) and Smagin and Shnyrev (2015) (both cited in the discussion ms) measured $CH_4$ emissions through snowpacks under different conditions. These studies evidence the transport of gas through snow layers as thick as 1.4 m. However, regarding the thinner snowpack the authors only show results from layers 10 cm thin. For our purpose, the lower limit of snow thickness is simply a model metric that allows us to differentiate between emissions in the presence or absence of snow. We selected thinner snow layers to test the model response, and the changes on this threshold thus mainly determine the timing of the emissions, which ultimately influences the magnitude of the total emissions through snow by allowing an earlier or later release of gas trapped in the soil.

5) $\phi$, snow porosity. This parameter has been previously reported in literature and is derived from snow density measurements, which ultimately controls the amount of gas that can be diffused through the snow layer. This was discussed in the ms. Based on observations, Pirk et al., (2016) measured methane emissions through snow with densities that ranged between ca. 250 kg m$^{-3}$ (at the surface of the snowpack) to 420 kg m$^{-3}$ (at about 80 cm depth of the 1.4 m snowpack). According to our model results, the snow depths in the model domain did not exceed 30 cm during the peak of the snow accumulation (shown in Figure S4c of the discussion ms), thus is unlikely to find dense snowpacks. We chose a maximum density of 330 kg m$^{-3}$ that corresponds to a porosity of 0.64 as our control value and tested for the sensitivity experiments less dense snowpacks with increasing porosities of 0.71 (for a density of 263 kg m$^{-3}$ characteristic of aged snow) and 0.86 (for a density of 128 kg m$^{-3}$ for fresh snow).

In the overall methane module:

6) *f*CH4anox or the fraction of anoxic decomposed carbon that becomes methane. This is a highly uncertain parameter in literature with some reported values in literature. In the discussion ms, we thoroughly discussed it (Lines 882 to 914), therefore we refrain to include here this discussion. However, we summarize by arguing that despite there are some values reported in literature, these are still uncertain and in our sensitivity tests we chose those values that have been reported and are characteristic of specific field conditions.

In the revised ms, we will improve the description of the selected parameters, especially those that are obtained from observational constraints.

Line 406-408: Why do the authors only show one adjacent cell? What is the justification to compare a neighboring grid cell to the ground-based observations? To demonstrate the spatial heterogeneity the authors could consider using more than just two grid cells.

Thank you for this suggestion. Our intention to show a neighboring grid cell was to demonstrate the spatial heterogeneity in the model results. Showing other grid cells in the model domain indeed can complement this. We believe that the maps showing spatial variability in flux rates provide already a good overview on the overall spatial variability. This larger scale variability is a superposition of many environmental factors, the most important of these being inundation fraction and coverage fraction of C3 grasses. The closer analysis for these two cells that the reviewer refers to was mainly performed to emphasize that even moderate variations in these factors (and others, such as e.g. soil depth) can lead to systematic differences in simulated fluxes. As we see it, extending this kind of analysis also to other cells would not add to this message, but rather confuse the reader by providing too

much information. We suggest, however, to extend the discussion related to the spatial heterogeneity in the modeled methane emissions by showing results of mean total methane fluxes in the eight grid cells surrounding grid cell A.

[Figure]

The figure on the left shows the time series of the total CH4 fluxes at daily resolution for the eight grid cells surrounding grid cell A (shown in black). One of those surrounding grid cells is grid cell B (red line).

In the tables below, the mean±std. of the total methane fluxes during summer (June, July and August) from grid cell A (given as values in black at the center cell of each table), and the surrounding eight grid cells for 2014 (left table) and 2015 (right table). Left side grid cell from the center cell, corresponds to the values for grid cell B (values in red).

| June, July and August in 2014 | | | | June, July and August in 2015 | | |
|---|---|---|---|---|---|---|
| 25.4 ± -7.1 | 56.0 ±-12.1 | 23.7± -3.7 | | 25.2±-8.3 | 56.9±-14.7 | 24.9±-4.8 |
| 72.8±-15.9 | 48.6±-5.5 | 27.8±-4.0 | | 75.5±-18.3 | 50.1±-7.3 | 28.2±-4.8 |
| 57.2±-6.8 | 57.9±-6.9 | 33.4±-4.7 | | 59.2±-8.7 | 59.5±-10.4 | 34.3±-5.7 |

In line 464-465, the authors mention the "parameter adjustment", but do not elaborate how exactly the parameter for the TOPMODEL was adjusted. Did the authors use an objective (cost) function to optimize this parameter?

There was no optimization of these parameters based on a cost function. The parameter adjustment for the TOPMODEL was also done in the same fashion as for the sensitivity studies: by varying each of the parameters of the TOPMODEL and analyzing the response by comparing the output to the chosen remote sensing data. This model parameter adjustment can only be done in this way within the current model structure. A more sophisticated optimization of parameters falls into a model data assimilation type, which is not implemented in this model configuration and goes beyond the scope of this work.

The authors demonstrate in their sensitivity analysis that the threshold TOPMODEL parameter and "allocation-of-decomposition-to-CH4" are the most important parameters determining the magnitude of simulated methane emissions. In my opinion, the authors should strengthen these results throughout the manuscript. It appears as if their results indicate that methane emissions mainly depend on methane production dynamics (i.e., fCH4anox) and on inundation as "on-off" switch of methane emissions.

The threshold TOPMODEL parameter and the allocation of C decomposition to methane are the parameters that, under the current model configuration, settings and for the selected groups of parameters for sensitivity tests, the most influential to the simulated methane emissions. This test was aimed to identify which of the most selected uncertain parameters have the highest influence to the results and with that, identify where the model is more sensitive and where it needs further improvements and evaluations, i.e. especially in those processes where the most influential parameters play a role in the model as in this case in the hydrology and carbon decomposition.

The methane emissions in our process-based model, not only depend on the methane that is produced based on the available carbon decomposed in the soil, but also depend on the available volumetric soil pore space, moisture, soil temperature and ice content in the soil as driving processes. Indeed, once methane is available in the soil to be emitted to the atmosphere, the inundated areas simulated with the TOPMODEL approach, determine the magnitude of the emissions. Our discussion regarding the sensitivity studies is based solely on the parameters chosen for the sensitivity experiments, and those are the threshold parameters in TOPMODEL and the fraction of available carbon to be decomposed into methane. We will improve this discussion to emphasize this result in this section.

Transport pathways and methane oxidation appear to be less important (merely changing the timing of emissions). Are these modelling results supported by observations in the field? The authors may consider discussing this in more detail.

The methane transport pathways are the result of the process-based methane calculations in the model according to, among others, the changes methane and oxygen concentrations in the soil and in the soil pore space that varies in relation to the freezing and thawing soil cycles, influencing directly the methane concentration in the soil.

The timing of the emissions is linked to the changes mostly in the soil physical state and speed of transport processes by their definition, e.g. diffusion of gas in air is faster than in water, resistance to molecular gas diffusion through the exodermis of plants, all in a process-based design. The model still lacks of a proper hydrology representation that allows inundation without having to set the soil moisture to saturated conditions, and that overall has an impact in the e.g. diffusion and oxidation of the methane. In the parts of the grid cell that are not water saturated because inundation cannot take place, the methane processes are not taken into account. Thus, the still not well-represented methane processes are not less important, but are only part of the limitations of the current model configuration and these results hint to the next steps to improve the model.

As presented in the discussion manuscript, some field studies have conducted experiments to measure independently the different pathways of methane emissions into the atmosphere. Through isotopic quantification of $\delta^{13}C$, Knoblauch et al., 2016 (cited in discussion ms) measured the amount of methane emitted through plants; Kwon et al., 2016 using chambers in the Chersky floodplain, also measured the gas emitted through plants. These studies were discussed in the discussion manuscript, and we demonstrated that, in agreement to field studies, the most dominant methane transport pathway from the total annual emissions (ca. 70 – 90 %), is through vascular plants when they are present. In the case of ebullition, this is a more difficult process to measure in field studies, because of its episodic nature. Despite some studies have attempted to measure methane emitted exclusively through ebullition (Tokida et al., 2007; Jammet et al., 2015 both cited in discussion ms), for models it is difficult to evaluate this process against observations.

In the case of methane oxidation, in our model configuration the oxygen content is explicitly taken into account, enabling two process-based oxidation processes: bulk soil methane oxidation and rhizospheric methane oxidation. After methane is produced in the soil (from available decomposed carbon), the bulk soil methane oxidation can take place considering the available oxygen in the soil pore spaces. The other oxidation pathway considers the available oxygen in plants. Only part of the oxygen in the soil is available for methane oxidation, and this discrimination relates to the amount of carbon dioxide produced during heterotrophic respiration, which has a maximum value of 40 % of the total oxygen content in the soil. An additional 10 % of the available oxygen is assumed to be unavailable because it is used in other processes (e.g. respiration by microbes). This leads to only 50 % of the total oxygen in the soil to be available for $CH_4$ oxidation. The methane processes in the model (oxidation and emission) take place in the inundated area, and this also restricts the magnitude of the oxidation. The daily methane oxidation rates for the two oxidation pathways for grid cells A and B in 2014 are shown in the figure below.

[Figure]

The bulk soil CH$_4$ oxidation accounts for about 1 % of the total methane production during the growing season for grid cell A and B, and an even smaller percentage (average 0.6 % for grid cell A and B during summer) for the rhizospheric CH$_4$ oxidation. These leads to most of the methane that is produced to be emitted to the atmosphere through the different transport pathways. Past observational and laboratory studies have estimated the methane oxidation in boreal and tundra soils. Whalen and Reeburgh (2000) showed that about 55 % of the CH$_4$ diffusing from the saturated boreal soils, were oxidized while reaching the surface. Through bottle incubations, Knoblauch et al. (2016) measured the volumetric CH$_4$ oxidation potential of soil and moss samples collected from ponds of the Lena Delta. The fraction of produced CH$_4$ that is oxidized before it is emitted was then calculated following three different approaches. Their results show a mean fraction of produced CH$_4$ that was oxidized between 61 to 78 % estimated from a stable isotope approach, while slightly different values were found in samples from pond areas without vascular plants: up to 90 % of the CH$_4$ that was produced, was completely oxidized following a potential methanogenesis approach, and between 63 % to 94 % calculated from diffusive CH$_4$ fluxes into the bottom water.

Berestovskaya et al. (2005), measured CH$_4$ oxidation rates of different soil samples from the Russian Arctic tundra and found that generally the rates of methane oxidation exceeded those to the rates of methane production especially at temperatures of 5 degC. For this to happen, methane-oxidizing bacteria rapidly consumes the methane released from the freshly thawed tundra soils and the methane already deposited in the unfrozen soil, and this takes place even before methanogens produce new methane.

Based on these scarce observations in boreal soils, the oxidation processes in our model are still robust and need to be revisited in order to improve the contribution of the methane oxidation processes into the total methane emissions. We will discuss this in more detail in the revised ms.

It is important noting that the process-based model presented in this manuscript explicitly considers physical drivers such as soil moisture, inundated area, soil temperature and substrate availability for methane production and emissions, and is potentially one of the few models that include explicitly methane oxidation processes. However, despite our efforts of improving the process-based representation, intrinsic model shortcomings are still present, and those are related to setting a fixed criterion in the soil moisture to allow the accumulation of water above the surface, which leads to a loss in the connection to the soil temperature. This has been clearly stated in our manuscript. As such, a one-to-one comparison between the model results and observations can be hardly expected. Still, we demonstrated advances in the process-model based approach which lead to methane emissions results that are comparable in temporal variation in magnitude to those measured on site.

Line 61-62: Perhaps the authors could mention another important permafrost thaw effect on methane emissions here: increasing surface wetness due to surface subsidence of ice-rich soils (see for example Christensen et al., 2004; Johnston et al., 2014, Helbig et al., 2017).
OK, we will mention this process in the revised manuscript. Thanks for this suggestion.

Line 94-100: Wintertime methane emissions have also been reported by Helbig et al. (2017) for a boreal peat landscape in northwestern Canada, where they found winter emissions to contribute about 25 % to the annual budget.
Thank you for adding this citation that we overlooked because by the time our manuscript was submitted, this paper was still not published, but now it will be added.

Line 121: Could the authors discuss here the most important "shortcomings in the parameterization" of the state-of-the-art methane models?
The biggest limitations for modeling methane emissions in boreal regions are related to the complex network of processes with highly variable influences that are difficult to disentangle with temporally and spatially scarce field measurements. The available published literature is also scarce and focused only on fine scale site-level studies or at very coarse global scale. Several models that attempt to simulate methane emissions from soils are rather coarse and not well documented or evaluated, which leads the process-based principle remaining rather incomplete. These shortcomings are well documented in the cited paper of Xu et al., 2016. According to the methane models intercomparison exercise WETCHIMP (Bohn et al., 2015; cited in the discussion manuscript) the authors conclude that process-based methane models are limited by the following factors: availability of a valid and highly resolved wetland map, account for methane emissions and uptake in dry non-wetland areas, limited soil thermal physics that do not contain freeze and thaw processes, lack of a snow scheme and consequently, gas transport in the presence of snow cover, lack of peat soils.
From our work presented in this manuscript and our model development efforts in this topic, we have taken into account some of this shortcomings and improved our model tool, however still limitations exist. We still conclude that in boreal regions influenced by permafrost, process-based modeling for methane emissions is challenged by the lack of the observational measurements that can contribute e.g. to understand better the dynamics of soil moisture and temperature, wetlands distribution, as well as the distribution and temporal variation of roots in vascular plants. Additionally, the land surface models that serve as framework have intrinsic limitations in their design, e.g. in the case of JSBACH, the hydrology scheme does not allow the accumulation or horizontal redistribution of water and other tools such as TOPMODEL had to be implemented. Still after the inclusion of the TOPMODEL approach, the final methane emissions are restricted exclusively to the areas where standing water takes place, leaving out the dry areas to come into play, and TOPMODEL does not feedback the soil thermal physics. Finally, the carbon decomposition scheme in our JSBACH model version is only dependent on 15-day mean of air temperature and precipitation, which leads to an absence of permafrost carbon in this model version.
This piece of discussion will be completed in the revised ms to clarify better out statement of L121.

Line 133: Perhaps the work by Cresto-Aleina et al. (2013, 2016) on microtopography effects on surface water and methane emission dynamics could be mentioned here too.
OK, we will add this citation and process as suggested.

Line 500-501: Only mineral soils are considered for the methane modelling? How common are organic soil in the study area? I would assume that at least top-soils in the floodplain would be organic-rich. How would "considering" organic soils change the results?
The lacking representation of organic soils is a shortcoming JSBACH has in common with many other land surface models. The authors are only aware of two peatland-enabled versions of the LPJ (Lund-Potsdam-Jena) model in the published literature. In addition, a small number of further modelling studies have been published, where organic layers were

considered, though mainly for their thermal properties (e.g. Ekici et al., 2014, cited in discussion ms). This lacking representation is mainly due to the difficulties of coupling sub-gridscale hydrology and carbon cycle in a holistic way. The reviewer is right that organic soils are common in the study area. From measurements in the Chersky floodplain reported in Kwon et al., 2016, the soil layer has a top organic peat layer about 15-25 cm thick on top of alluvial material composed of silty clay. In the model configuration, only mineral soils are considered and indeed the organic carbon pools might be depressed in contrast to the organic carbon in a peat layer. This was discussed earlier above in this response.

Line 577-579: The authors may consider supporting this statement with information on the exact magnitude of interannual variability.
We calculated the magnitude of the interannual variability of the fluxes from eddy covariance and the model by comparing the standard deviation of the monthly values from 2014 to those in 2015. We summarize these results in the figure below.

[Figure]

The statement in the discussion ms is now supported by showing that largest interannual variability in the model grid cell A takes place in May and July with 7.9 and 5.5 mg $CH_4$ m$^{-2}$ d$^{-1}$ when compared the standard deviations from the monthly fluxes between 2014 and 2015, while for grid cell B, the largest variability between the two years took place in June and July (10.9 and 5.6 mg $CH_4$ m$^{-2}$ d$^{-1}$, respectively). Still, the largest interannual variability was observed in June for the Eddy covariance data with 12.6 mg $CH_4$ m$^{-2}$ d$^{-1}$ difference in their monthly standard deviation between both years.
For practical reasons, we will complete the lines mentioned by the reviewer with the quantities obtained, and we will not show the figure as well.

Line 589-592: What is the uncertainty in the eddy covariance flux measurements?
Could the authors quantify uncertainties due to random errors, gap-filling, u*-threshold, and footprint heterogeneity? An uncertainty quantification of eddy covariance fluxes would further strengthen the model-observation comparison.
The uncertainty analysis for the eddy-covariance flux data consists of random and systematic errors and is assessed based on well-established concepts (Aubinet et al., 2012).
Random errors linked to the turbulent sampling error and instrument error are given as standard output of the flux processing software TK3 (Mauder and Foken, 2015) for each 30 min flux value. Footprint uncertainties are not quantified, since there are no major transitions in biome types within the core areas of the flux footprints. Random errors are combined and considered as independent variables.

Systematic errors can occur due to unmet assumptions and methodological challenges, instrument calibration and data processing. Instruments are calibrated in regular intervals, and in comparison to a second eddy-covariance tower close by (~ 600 m) no systematic offset in the frequency distributions of wind speed, sonic temperature, and methane mixing ratios between towers was observed. The standardized software TK3 (Mauder and Foken, 2015) contains all the required processing steps for the flux data processing, as well as conversions and corrections, and yielded good agreement in a recent comparison with EddyPro (Fratini and Mauder, 2014). The post-processing quality control and flagging system scheme was based on stationarity and a well-developed turbulence scheme proposed by Foken and Wichura (1996) followed by additional tests applied to flag implausible data points in the resulting flux time series. Data coverage of methane fluxes was 86 % during the growing season and 67 % during the winter (Kittler et al., 2017).

The gap-filling method is based on a moving window that is centered in the gap and a 10-day window length, i.e. 5 days before and 5 days after the gap. The uncertainties were quantified as standard deviation for the corresponding window, similar to the gap-filling uncertainties for the $CO_2$ flux via the MDS routine (Reichstein et al., 2005).

No u*-threshold was applied to the flux dataset, since we determined the stationarity of the signal and integral turbulence characteristics also for nighttime conditions. This information facilitates identifying datasets with regular turbulent exchange also during stable stratification, therefore producing fewer gaps compared to a bulk exclusion of data during stable nighttime stratification through the u*-filter method. Random errors decrease with averaging and were calculated according to Rannik et al. (2016).

Averaged over both years (2014 and 2015) the $CH_4$ flux uncertainty based on 30 min data is $5.1 \pm 8.8$ nmol m$^{-2}$ s$^{-1}$ ($7 \pm 12.1$ mg $CH_4$ m$^{-2}$ d$^{-1}$). This result is not considering gap-filling techniques to the quality-checked signal (bulk uncertainty). The mean value considering also gap-filling is: $7.4 \pm 8.3$ nmol m$^{-2}$ s$^{-1}$ ($10.2 \pm 11.5$ mg $CH_4$ m$^{-2}$ d$^{-1}$).

For a fen ecosystem, it has been reported an uncertainty of $4.7 \pm 3.8$ nmol m$^{-2}$ s$^{-1}$. This result considers quality-checked data without applying a gap-filling technique (Jammet et al., 2017). After considering monthly averaging of the gap-filling and with a quality checked signal, the uncertainties of the $CH_4$ fluxes measured from EC for 2014 and 2015 are reduced to $0.35 \pm 0.22$ mg $CH_4$ m$^{-2}$ d$^{-1}$. Monthly uncertainty values will be included in Figure 5 for the revised ms as error bars of the mean monthly values. The updated figure is presented below. Details on data uncertainty assessment as outlined above, will be provided in a new Appendix B on "Details on in-situ flux observations". References cited in this section are listed at the end of this response.

[Figure]

Line 691-711: I am not sure how this section contributes to the research questions of this manuscript? Perhaps the authors could mention differences in environmental characteristics of grid-cell A and B briefly in the manuscript and move figure 9 to the supplementary material?

We will shorten this section and instead merge it with the discussion of methane fluxes, in this way we could move figure 9 to the supplementary material. This suggestion certainly will make the manuscript more focused on the main aim. Thanks for this suggestion.

Line 808-810: The impact of cooler early summer temperatures on soil warming and methane emissions has been demonstrated recently using multi-year methane observations in a boreal peat landscape (see Helbig et al., in press). The authors may consider discussing their modelling results in relation to these observations.

We thank the reviewer for making us aware of this new publication. We will add it in the revised ms and cite it accordingly. Helbig et al., 2017 shows that between years 2013 and 2016, during May of each year the one in 2014 was colder compared to the other years. This finding was based on a meteorological record of an area in northwestern Canada. As a result of temperature shifts, soil temperatures varied and influenced the year-to-year methane fluxes, specially variations in spring soil temperature were influential. The findings of Helbig et al. are in good agreement with our model observations regarding the interannual variability in air and soil temperature and their influence in methane emissions. We will complete our model results with this nice comparison.

Line 847-851: The authors may consider starting the discussion mentioning the parameters that actually made a difference and not with the parameters that did not change the results. It should be highlighted what process/parameter matters in the model.

Thank you for this suggestion. We will re-structure the discussion based on this suggestion.

Line 991-992: Few studies have shown that non-inundated upland areas may take up methane (e.g., Flessa et al., 2008). As far as I understand, such uptake is not considered in the current work. How could uptake in the drier areas of the model domain change simulation results? There are large areas in the model domain that appear to be characterized by upland landscapes and thus potential methane uptake (see Fig. 1).

Indeed atmospheric $CH_4$ uptake should not be neglected when considering a regional $CH_4$ budget, especially when the majority of areas are predominantly aerobic. With plot-based observations in dry areas of the Chersky floodplain, the $CH_4$ emissions where negative, indicating uptake (average of -3 mg $CH_4$ m$^{-2}$ d$^{-1}$ during summer of 2014) and this value was considerably smaller compared to that of the $CH_4$ emissions measured in wet plots (on average 332 mg $CH_4$ m$^{-2}$ d$^{-1}$ in summer 2014) (see response above for figure of results). Based on this result and the consideration of an inundated fraction of 20 % during summer in the model grid cell A (Fig. 9d in discussion manuscript), about 66.4 mg $CH_4$ m$^{-2}$ d$^{-1}$ (in wet plots: 332 mg $CH_4$ m$^{-2}$ d$^{-1}$ * 0.2 = 66.4 mg $CH_4$ m$^{-2}$ d$^{-1}$) are emitted and 2.4 mg $CH_4$ m$^{-2}$ d$^{-1}$ are loss by uptake (oxidized) (given by the dry plots results: -3 mg $CH_4$ m$^{-2}$ d$^{-1}$ * 0.8 = -2.4 mg $CH_4$ m$^{-2}$ d$^{-1}$ equivalent to 8.7 % of the total methane emissions), leading to the net $CH_4$ emission of 64 mg $CH_4$ m$^{-2}$ d$^{-1}$ according to chamber measurements. The mean $CH_4$ emission in grid cell A during June-July-August 2014 (Fig. 5a of discussion manuscript) is 48.6 mg $CH_4$ m$^{-2}$ d$^{-1}$ in the inundated areas. The mean methane soil and plant oxidation for the same period of time, given by the model for grid cell A, is 0.63 mg $CH_4$ m$^{-2}$ d$^{-1}$ (0.3 % of the total emission) which is low compared to the uptake estimation for the chamber measurements in dry areas. However, these are only representing oxidation processes in saturated soils, which are not predominant in contrast to dry soils. As mentioned before, by not considering non-inundated areas in the modeling of methane processes, the methane uptake is ultimately underestimated because the conditions for methane oxidation are limited. The model can be further improved in the $CH_4$ oxidation scheme, but this can only be possible after a thorough observation of $CH_4$ uptake rates and their controlling factors in this area, and also as the

hydrology scheme is also improved. More on the methane oxidation in the model is discussed in this response and will be also emphasized in the revised ms.

Line 1134-1141: The authors may consider not to introduce a new concept (e.g., anaerobic microsites) at the very end of the conclusions. I would recommend to only refer here to what has been shown in the manuscript so far.
OK, we will improve this section in the revised manuscript.

Line 1252-1255: What would happen if the model would run with the old order of processes? Shouldn't this be part of the uncertainty analysis?
The old order of processes was presented in the paper by Kaiser et al., 2017; there, it was shown that the order of processes was selected based on the velocity that they physically can exhibit, with ebullition first and the slowest transport at last which was plant mediated transport due to the resistance of the plants exodermis. Observational evidences however, as discussed here and in the manuscript, show that in the presence of vascular plants, wetland annual methane emissions are mainly from the transport of gas through plants. Due to the structure of the model, it is not possible to run parallel processes and instead, a sequential flow of processes has to be computed. For this reason, the solution to improve the individual share of the transport processes was to re-arrange the processes by expected priority. We do not think this should be part of the sensitivity studies for this manuscript since this is a purely computational design and not due to the inherent processes in the model.

Fig. 1: Why did the authors use such a large study area, if ground-based observations were only available for a very small fraction of the model domain? How can the model performance be evaluated for the other non-floodplain grid cells that appear to be characterized by different landscape characteristics?
We agree with the reviewer that a smaller study area could have been shown, especially for the area near the grid cell where ground-based observations take place.
Plot level model simulation have been performed in the past, particularly with a similar version of the model presented in this manuscript for a site in Samoylov (Kaiser et al., 2017). Model development for methane emissions has not only focused on the improvement in the mechanisms represented in the model for the production and transport of methane, but also in the scaling with the intention of understanding better the contribution of $CH_4$ processes over larger spatial scales. Regional scales still pose a challenge but certainly models need to be aimed to be applied to larger scares rather than only plot level. After a plot level application, we improved the description of some processes in the model and aim to test it in a rather larger, but still, regional spatial scale.
Thus, the intention of selecting a larger regional domain is two-fold: 1) to test and apply the process-based methane model in a larger than site-level domain and 2) to identify the heterogeneity in the methane processes linked to different soil and vegetation conditions, this is important since sub-grid soil heterogeneity is still not represented in the model, and is also particularly relevant for large-scale inundation evaluation.
We agree with the reviewer that no observational data is available to evaluate more than one model grid cell, and indeed one should be careful with the interpretation of the non-floodplain areas of the model domain, however, our contribution here is also aimed to be used for even larger domains and for future predictions, so testing such model already in larger scale and been showing its computational capability and overall realistic performance is a step forward towards that aim. As more observational efforts will be done in the future, in other areas near the Kolyma region and Chersky floodplain for our own practical purposes, the model will be able to be evaluated for those other areas. Also, an intercomparison between JSBACH results and atmospheric inverse modeling at the regional scale is in preparation. Until then, we still believe in the value of our scientific contribution to evidence the applicability of a refined process-based methane model.

Fig. 6: Why do the authors compare the mean grid-cell soil temperature profile to measured wet and dry soil temperature profiles? Physical soil properties differ drastically between wet and dry soils and consequently strongly determine soil temperature dynamics (see end of discussion). Wouldn't it be therefore necessary to at least model soil temperature dynamics of the inundated and non-inundated land surface separately?

We present these data in Fig. 6 to show the existing model physical state that was used for the calculation of methane emissions in the model. We agree with the reviewer that ideally, the model should be able to produce results separately for the dry and for the wet soil areas. However and unfortunately, this is not possible with the current model configuration and this is due to the basic model structure of JSBACH. Each model grid cell is subdivided into tiles that only serve to describe different vegetation types, however the soil properties remain the same for the entire grid cell and average soil state variables are considered. Thus, the soil temperature dynamics actually represent the entire grid cell and these are independent of the TOPMODEL interactions, i.e. inundated and non-inundated areas. This is obviously a shortcoming in this version which was presented in the discussion ms and which is true for many other land surface models. In order to represent sub-grid heterogeneity of soil properties the model configuration would need to be completely restructured which we hope can be done in the near future with the new developments of the JSABCH 4.0.

Fig. 7: Methane emissions increase considerably in the model at sub-zero soil temperatures. In contrast, measured methane emissions appear to be quite insensitive to soil temperature below 0_C. The authors mention this mismatch in lines 655-659. Perhaps the authors can discuss this mismatch between temperature-emission responses in more detail. How is it possible that such cold simulated soil temperatures result in emissions of > 30 mg CH4 m-2 day-1?

We agree with the reviewer that wintertime processes are still not well captured with our current model configuration. This goes down basically to the soil moisture that had to be artificially modified to allow the accumulation of water at the soil surface according to the topographic profile. The results presented in Fig. 7 show that, the high model methane emissions mentioned by the reviewer, take place mostly during October and May (grey circles and triangles for grid cells A and B, respectively) and this reflect the gradual transition of the emissions as the soil starts to freeze towards December and also as it starts to melt before summer. Comparing with the observations, this result seems implausible, however, we think is not also impossible to happen. In the work of Zona et al. (2016), the authors demonstrated the emissions of methane during the zero curtain period. In their Figure 3, panel B, high methane emissions take place still at subzero temperatures (on average 7.8 mg CH$_4$ m$^{-2}$ d$^{-1}$ at - 5 degC) between September and December in 2014, while in panel A, the methane fluxes behave more similarly to our observations in the Chersky floodplain (barely changing < 0 degC). Still the magnitude of the observed emissions is not as large as what we observe with JSBACH and here the model parameters and schemes might play the role. The zero curtain period presented in Zona et al. reflects the release of CH4 still in autumn, due to the production of CH4 in sub-soil warm layers. To investigate if the results of our model reflect somehow this process as well, still other schemes in the model must be revisited and improved such as: Q10 and water impact in carbon decomposition, and processes such as soil freezing under moisture limitation and thermal soil response.

Fig. 8: Here, an uncertainty estimate for the measured cumulative methane emissions would help interpreting the comparison between simulated and measured fluxes.

In order to include uncertainty estimates to the cumulative methane emissions presented in Fig. 8, we calculated the monthly cumulative fluxes in panel e and added the error bars as standard deviation of the monthly cumulative fluxes. Despite our discussion regarding the total cumulative fluxes when comparing the eddy covariance record to the model grid cells results, we observe that the uncertainty in the monthly fluxes is larger in all of the data sets during October 2014 and generally decreases toward April 2015. The uncertainty ranges are

also generally larger in the eddy covariance data and this is due to the high intrinsic signal daily variability.

We updated this figure in the revised ms and discussed the uncertainty values. The new figure is:

[Figure]

Fig. 11: I am not sure how this figure contributes to the research questions. The seasonality of different methane emission pathways is already shown in Fig. 10. How does a representation of the spatial distribution of the methane emissions add to the manuscript?

We still think that showing the total model domain methane emissions from the different pathways evidence the skill of the model for its regional application. In Fig. 10, only grid cells A and B are shown. Furthermore, by following the suggestion of the reviewer of moving Fig. 10 to supplementary material, we still would like to keep Fig. 11 as part of the main ms.

**Technical comments**

Line 149: Remove "done".

OK

Line 150: Remove "are".

OK

Line 196: Please define what "hospitable and inhospitable" land means in this context.

We have completed this paragraph by adding the following lines: "A prescribed fraction of each grid cell is used to discriminate between land hospitable and inhospitable to vegetation. In JSBACH, each grid cell has a designated fraction where vegetation cover types across tiles can be assigned, hence is the fraction hospitable to vegetation. The remaining fraction of the grid cell is then associated to a land cover type that represents areas where vegetation does not grow, such as rocky surfaces and deserts; hence it is considered inhospitable to vegetation (Reick et al., 2013).

Line 534: What do the authors mean with "visually"? They state in the previous sentence that differences are not statistically significant.

We refer here to the time shift in the mean methane emissions signal when the sensitivity experiments are compared. However, indeed the statistical analysis showed that there is no significant difference between the results of the sensitivity tests for the individual and total emissions. We rephrased this sentence to avoid confusion, it now reads: "A time shift is seen however, in the $CH_4$ emissions from mid-October until mid-November (Fig. 3, column 4 of row e), with the larger emissions through snow taking place earlier if $h_{snow}$ is thinner.

Nevertheless, this temporal shift in the $CH_4$ emissions through the snow is not observed in the total $CH_4$ emissions."

Fig. 3: Please clarify what the inset figures show.
Thank you for pointing this out. Now we added the following sentence to the caption: "The inset figures in some of the panels are zooms to periods of time where larger difference between signals is depicted."

**References cited in this response that are not included in the original discussion manuscript:**
Aubinet, M., T. Vesala, and D. Papale (2012), Eddy Covariance - A practical guide to measurement and data analysis, 449 pp., Springer, Dordrecht; Heidelberg; London; New York, doi:10.1029/2002JD002055.

Berestovsakaya, Y.Y., I.I. rusanov, l.V. Vasil'eva and N.V. Pimenov (2005), The processes of methane production and oxidation in the soils of the Russian Arctic tundra, Microbiology, 74(2), 221-229, 10.1007/s11021-005-0055-2.

Chapin III, F.S. and M. Slack (1979), Effect of defoliation upon root growth, phosphate absorption and respiration in nutrient-limited tundra graminoids, Oecologia, 42, 1, 67-79, 10.1007/BF00347619.

Foken, T., and B. Wichura (1996), Tools for quality assessment of surface-based flux measurements, Agr Forest Meteorol, 78(1-2), 83-105, doi:10.1016/0168-1923(95)02248-1.

Fratini, G., and M. Mauder (2014), Towards a consistent eddy-covariance processing: an intercomparison of EddyPro and TK3, Atmos Meas Tech, 7(7), 2273-2281, doi:10.5194/amt-7-2273-2014.

Jammet, M., Crill, P., Dengel, S., and Friborg, T.: Large methane emissions from a subarctic lake during spring thaw: mechanisms and landscape significance, Journal of Geophysical Research: Biogeosciences, 120, 2289-2305, 2015, 10.1002/2015JG003137.

Jammet, M., S. Dengel, E. Kettner, F. J. W. Parmentier, M. Wik, P. Crill, and T. Friborg (2017), Year-round $CH_4$ and $CO_2$ flux dynamics in two contrasting freshwater ecosystems of the subarctic, Biogeosciences, 2017, 14, 5189-5216, doi:10.5194/bg-14-5189-2017.

Kittler, F., M. Heimann, O. Kolle, N. Zimov, S. Zimov, and M. Göckede (2017), Long-term drainage reduces $CO_2$ uptake and $CH_4$ emissions in a Siberian permafrost ecosystem, Global Biogeochem. Cy., 31, 1704-1717, doi:10.1002/2017GB005774.

Mauder, M., and T. Foken (2015), Eddy-Covariance Software TK3, doi:10.5281/zenodo.20349.

Rannik, U., O. Peltola, and I. Mammarella (2016), Random uncertainties of flux measurements by the eddy covariance technique, Atmos Meas Tech, 9(10), 5163-5181, doi:10.5194/amt-9-5163-2016.

Reichstein, M., et al. (2005), On the separation of net ecosystem exchange into assimilation and ecosystem respiration: review and improved algorithm, Glob Change Biol, 11(9), 1424-1439, doi:10.1111/j.1365-2486.2005.001002.x.

Reick, C., T. Raddatz, V. Brovkin, V. Gayler, (2013), Representation of natural and anthropogenic land cover change in MPI-ESM, Journal of Advances in Modeling Earth Systems, 5, 459-482, 10.1002/jame.20022.

Saltelli, A., Chan, K. & Scott, E. M. (2000), *Sensitivity Analysis*. *Wiley Series in Probability and Statistics*, John Wiley & Sons, ltd.

Tokida, T., Mizoguchi, M., Miyazaki, T., Kagemoto, A., Nagata, O., and Hatano, R.: Episodic release of methane bubbles from peatland during spring thaw, Chemosphere, 70, 165-171, 2007, 10.1016/j.chemosphere.2007.06.042.

Wang, P., L. Mommer, J. van Ruijven, F. Berendse, T. C. Maximov, M. M. P. D. Heijmans, 2016, Seasonal changes and vertical distribution of root standing biomass of graminoids and shrubs at a Siberian tundra site, Plan Soil, 10.1007/s11104-016-2858-5.

Whalen, S.C. and W.S. Reeburgh (2000), Methane oxidation, production, and emission at contrasting sites in a Boreal Bog. Geomicrobiology Journal, 17, 237-251.

**References recommended by reviewer (and added also to the revised ms).**

Christensen TR, Johansson T, Åkerman HJ et al. (2004) Thawing sub-arctic permafrost: Effects on vegetation and methane emissions. Geophysical Research Letters, 31, L04501.

Cresto-Aleina F, Brovkin V, Muster S, Boike J, Kutzbach L, Sachs T, Zuyev S (2013) A stochastic model for the polygonal tundra based on Poisson – Voronoi diagrams. Earth System Dynamics, 4, 187–198.

Cresto-Aleina F, Runkle BRK, Brücher T, Kleinen T, Brovkin V (2016) Upscaling methane emission hotspots in boreal peatlands. Geoscientific Model Development, 9, 915–926.

Flessa H, Rodionov A, Guggenberger G et al. (2008) Landscape controls of CH4 fluxes in a catchment of the forest tundra ecotone in northern Siberia. Global Change Biology, 14, 2040–2056.

Helbig M, Boike J, Langer M, Schreiber P, Runkle BRK, Kutzbach L (2013) Spatial and seasonal variability of polygonal tundra water balance: Lena River Delta, northern Siberia (Russia). Hydrogeology Journal, 21, 133–147.

Helbig M, Chasmer L, Kljun N, Quinton WL, Treat CC, Sonnentag O (2017) The positive net radiative greenhouse gas forcing of increasing methane emissions from a thawing boreal forest-wetland landscape. Global Change Biology, 23, 2413–2427.

Helbig M, Quinton WL, Sonnentag O (2017) Warmer spring conditions increase annual methane emissions from a boreal peat landscape with sporadic permafrost. Environmental Research Letters, 12, 115009, doi: 10.1088/1748-9326/aa8c85.

Johnston CE, Ewing S a, Harden JW et al. (2014) Effect of permafrost thaw on CO2 and CH4 exchange in a western Alaska peatland chronosequence. Environmental Research Letters, 9, 85004.

Liljedahl AK, Boike J, Daanen RP et al. (2016) Pan-Arctic ice-wedge degradation in warming permafrost and influence on tundra hydrology. Nature Geoscience, 9, 312–318.

Parmentier FJW, van Huissteden J, van der Molen MK, Schaepman-Strub G, Karsanaev SA, Maximov TC, Dolman AJ (2011) Spatial and temporal dynamics in eddy covariance observations of methane fluxes at a tundra site in northeastern Siberia. Journal of Geophysical Research, 116, G03016.

Sachs T, Giebels M, Boike J, Kutzbach L (2010) Environmental controls on CH4 emission from polygonal tundra on the microsite scale in the Lena river delta, Siberia. Global Change Biology, 16, 3096–3110.

**Appendix B: Details on in-situ flux observation program**

*Eddy-covariance flux data uncertainty assessment*

Following well-established procedures in literature (Aubinet et al., 2012), our uncertainty analysis for the eddy-covariance flux data has been split up into random and systematic errors. The major sources for random errors, associated with the turbulent sampling and instrument issues, have been quantified for each 30 min flux value through the flux processing software TK3 (Mauder and Foken, 2015). Errors related to footprint uncertainties were not quantified, since there are no major transitions in biome types within the core areas of the flux footprints.

Systematic errors can be introduced by unmet theoretical assumptions and methodological challenges, as well as by instrument calibration and data processing issues. In the context of the Chersky observations, instruments were maintained and calibrated in regular intervals, therefore minimizing this potential error component. Moreover, data intercomparisons with a second eddy-covariance tower close by (~ 600 m) yielded no systematic offset in the frequency distributions of wind speed, sonic temperature, and methane mixing ratios between towers. Regarding flux data processing, the TK3 software package contains all the required processing steps, conversions and corrections for the flux data processing, and yielded good agreement in a recent comparison with EddyPro (Fratini and Mauder, 2014). To avoid methodological issues that may bias flux data results, we employed a rigid post-processing quality control and flagging system scheme, with the well-established analyses for stationarity and well-developed turbulence originally proposed by Foken and Wichura (1996) at its core, supplemented by additional tests (absolute range and spikes) to flag implausible data points in the resulting flux time series. Based on the quality assessment and control tools outlined above, we excluded systematic errors from the uncertainty quantification of flux data that were assigned a high to medium data quality (QF 1-6 based on the scheme proposed by Foken et al., 2005; 2012) and subsequently used for assessing long-term $CH_4$ flux budgets.

No u*-threshold was applied to the flux dataset, since we determined stationarity of the signal and integral turbulence characteristics also for nighttime conditions. This information facilitates identifying datasets with regular turbulent exchange also during stable stratification, therefore producing fewer gaps compared to a bulk exclusion of data during stable nighttime stratification through the u*-filter method. After filtering out low-quality fluxes, the data coverage of methane fluxes was 86 % during the growing season and 67 % during the winter from the original full 30 min flux data set (Kittler et al., 2017). To produce a continuous flux record for quantification of long-term $CH_4$ budgets, we filled the remaining gaps by averaging existing flux data within a moving window of 10-day length centered on the gap. Uncertainties for gap-filled values were quantified as standard deviation within the corresponding window, similar to the definition of gapfilling uncertainties for the $CO_2$ flux via the well-established marginal distribution sampling routine by Reichstein et al. (2005).

To produce aggregated uncertainty values for longer time periods, we applied the procedures suggested by Rannik et al. (2016). All random errors were combined by considering them as independent variables, and normally decrease with the length of the averaging period. Averaged over both data years used within the context of this study (2014 and 2015), the $CH_4$ flux uncertainty based on the 30 min data is 7.4±8.3 nmol $m^{-2}$ $s^{-1}$, a result comparable to 4.7±3.8 nmol $m^{-2}$ $s^{-1}$ reported for a fen ecosystem by Jammet et al. (2017).

*Source weight function of the eddy-covariance flux data*

We conducted a source weight analysis, also called footprint analysis, to determine the fractional contribution of different land cover types within the field of view of the eddy-covariance flux tower. Source weight functions for each 30min flux measurement were computed based on the Lagrangian Stochastic footprint model by Rannik et al. (2003). Footprints were accumulated, analyzed and interpreted using an approach presented by Göckede et al. (2006; 2008). We projected these footprints onto a WorldView2 land cover map at 2 m horizontal resolution (see also Figure A1). In the context of the presented study, we aggregated the original 22 land cover classes into 9 classes to concentrate on the dominant elements of the vegetation community structure (see also Table A1).

[Figure]

**Figure A1: Accumulated source weight function for the control tower within the Chersky study site, based on data from the growing season (mid June – mid September) in 2014. Solid white isolines indicate the 80, 60, 40, and 20% levels, while the dashed line gives the 10% level. Background colors give aggregated land cover classes based on WorldView2 data.**

Since the tower is situated on a slightly elevated patch of tundra, tussocks and shrubs featuring various levels of wetness (red and orange colors in Fig. A1) dominate the immediate surroundings. Even though inundated parts of the study area, in this case identified by the prevalence of the cotton grass *Eriophorum angustifolium* (blue-ish colors in Fig. A1), are dominating the area encircled by the 10% isoline that is used here to mark the boundary of the cumulative footprint area, they are mostly present in the outer reaches, therefore combining just about 26% of the total flux signal sampled by the eddy system. Another 31% is contributed by wet to moist tussock tundra with some shrubs. Overall coverage fractions within the major wetness categories (see also Table A1) remain approximately constant between tower footprint and two larger regions covered by the same WorldView dataset, indicating that this composition of wetness levels is typical for the Kolyma floodplain ecosystems analyzed within the context of this study.

**Table A1: Fractional coverage of aggregated WorldView land cover classes within the control tower footprint of the Chersky study site. Background color coding was used to categorize the classes into wetness levels. The rightmost two columns give fractional coverage of these classes within the area immediately surrounding the towers (1.2 x 1.2 km) and within the entire WorldView scene analyzed (5 x 5 km).**

| Land cover class | category | tower footprint | 1.2x1.2km | 5x5km |
|---|---|---|---|---|
| water | open water | 0.001 | 0.035 | 0.134 |
| cotton grass, wet continuously | | 0.111 | 0.043 | 0.053 |
| cotton grass, partially dry | wetland | 0.067 | 0.153 | 0.147 |
| cotton grass with tussocks | | 0.081 | 0.063 | 0.038 |
| tussocks with some shrubs | wet to moist | 0.312 | 0.418 | 0.280 |
| tussocks with higher shrubs | | 0.388 | 0.165 | 0.182 |
| higher shrubs, with tussocks | moist to dry | 0.031 | 0.097 | 0.115 |
| trees | | 0.001 | 0.006 | 0.017 |
| undefined | | 0.008 | 0.020 | 0.035 |

*Flux chamber observations*

The Chersky study site features two transects of 10 permanently installed PVC collars for flux chamber measurements. With distances of approximately 25 m between individual microsites, both transects cover a distance of ~225 m within the drained and control sections, of this permafrost site. Site locations were selected quasi-randomly to reflect the dominant microsite characteristics (e.g. vegetation composition, wetness level) found at each of the target locations. With a chamber footprint of 60 cm x 60 cm, this technique allowed studying microsites with rather homogeneous environmental conditions, as compared to the eddy-covariance fluxes with often heterogeneous footprint areas. Details on the chamber program, overall methane flux rates observed, and functional relationships with e.g. soil temperature, vegetation and wetness levels, are provided by Kwon et al. (2016; 2017).

[Figure]

**Figure A2: Daily methane flux rates aggregated from flux chamber measurements within the growing season of 2014. Measurements are separated into drained (1 wet microsite, 9 dry microsites) and control (8 wet microsites, 2 dry microsites) transects.**

Figure A2 displays average flux rates for wet and dry microsites observed within the drained and control transects during sampling campaigns in summer 2014. These flux chamber results clearly demonstrate that methane release rates were virtually zero in the absence of standing water. At some of the dry microsites (results not shown), even negative $CH_4$ flux rates were observed, indicating the oxidation of methane under highly aerobic conditions within these predominantly wet tussock tundra ecosystems in Northeastern Siberia.

**Cited literature**

Aubinet, M., T. Vesala, and D. Papale (Eds.) (2012), Eddy Covariance - A practical guide to measurement and data analysis, 438 pp., Springer, The Netherlands.

Foken, T., R. Leuning, S. P. Oncley, M. Mauder, and M. Aubinet (2012), Corrections and data quality, in Eddy Covariance - A practical guide to measurement and data analysis, edited by M. Aubinet, T. Vesala and D. Papale, pp. 85-131, Springer, Dordrecht; Heidelberg; London; New York.

Foken, T., M. Göckede, M. Mauder, L. Mahrt, B. Amiro, and W. Munger (2005), Post-Field Data Quality Control, in Handbook of Micrometeorology: A Guide for Surface Flux Measurement and Analysis, edited by X. Lee, W. Massman and B. Law, pp. 181-208, Springer Netherlands, Dordrecht, doi:10.1007/1-4020-2265-4_9.

Foken, T., and B. Wichura (1996), Tools for quality assessment of surface-based flux measurements, Agr. Forest Meteorol., 78(1-2), 83-105, doi:10.1016/0168-1923(95)02248-1.

Fratini, G., and M. Mauder (2014), Towards a consistent eddy-covariance processing: an intercomparison of EddyPro and TK3, Atmos Meas Tech, 7(7), 2273-2281, doi:10.5194/amt-7-2273-2014.

Göckede, M., et al. (2008), Quality control of CarboEurope flux data - Part 1: Coupling footprint analyses with flux data quality assessment to evaluate sites in forest ecosystems, Biogeosciences, 5(2), 433-450.

Göckede, M., T. Markkanen, C. B. Hasager, and T. Foken (2006), Update of a footprint-based approach for the characterisation of complex measurement sites, Bound.-Lay. Meteorol., 118(3), 635-655, doi:10.1007/s10546-005-6435-3.

Jammet, M., S. Dengel, E. Kettner, F. J. W. Parmentier, M. Wik, P. Crill, and T. Friborg (2017), Year-round CH4 and CO2 flux dynamics in two contrasting freshwater ecosystems of the subarctic, Biogeosciences, 14(22), 5189-5216, doi:10.5194/bg-14-5189-2017.

Kittler, F., M. Heimann, O. Kolle, N. Zimov, S. Zimov, and M. Göckede (2017), Long-term drainage reduces CO2 uptake and CH4 emissions in a Siberian permafrost ecosystem, Glob. Biogeochem. Cy., (online first), doi:10.1002/2017GB005774.

Kwon, M. J., M. Heimann, O. Kolle, K. A. Luus, E. A. G. Schuur, N. Zimov, S. A. Zimov, and M. Göckede (2016), Long-term drainage reduces $CO_2$ uptake and increases $CO_2$ emission on a Siberian floodplain due to shifts in vegetation community and soil thermal characteristics, Biogeosciences, 13(14), 4219-4235, doi:10.5194/bg-13-4219-2016.

Kwon, M. J., et al. (2017), Drainage enhances surface soil decomposition but stabilizes old carbon pools in tundra ecosystems Nat. Clim. Change, (submitted).

Mauder, M., and T. Foken (2015), Documentation and Instruction Manual of the Eddy-Covariance Software Package TK3Rep., University of Bayreuth, Bayreuth.

Rannik, U., T. Markkanen, J. Raittila, P. Hari, and T. Vesala (2003), Turbulence statistics inside and over forest: Influence on footprint prediction, Bound.-Lay. Meteorol., 109(2), 163-189, doi:10.1023/a:1025404923169.

Rannik, Ü., O. Peltola, and I. Mammarella (2016), Random uncertainties of flux measurements by the eddy covariance technique, Atmos. Meas. Tech., 9(10), 5163-5181, doi:10.5194/amt-9-5163-2016.

Reichstein, M., et al. (2005), On the separation of net ecosystem exchange into assimilation and ecosystem respiration: review and improved algorithm, Glob Change Biol, 11(9), 1424-1439, doi:10.1111/j.1365-2486.2005.001002.x.

---

## Author Comment (AC2) · 21 Jan 2018

Thank you very much for your comments to our discussion manuscript. Below we address one by one the comments made during this review. All answers are in blue font.

Overall:

The ms has its focus on regional scale methane dynamic and the modelling of year round dynamics, which is certainly relevant and highly needed. I general there are quite few year round measurements of methane dynamics in the arctic region, which also explains why the modelling studies are even fewer and regional budgets are poorly constrained. Further, the understanding of drivers and exact transport mechanisms in the top soil and soil – snow-atmosphere still in most (not all) cases relies on an interpretation of a net emission, rather that independent quantification of the individual components adding up the net CH4 emission. For that reason the focus of the current ms is important and timely. Despite that the ms is well written and in general well references, I'm a bit reluctant about the qualities of the ms, because I basically find that it tries to accomplish too much and not in a fully convincing way. As pointed out by reviewer 1, also I have a serious problem with the differences in scaling which are used in the different components of the study. In my perspective, the very coarse spatial scale of the model does not compare well with the highly advanced model approach of partitioning the production and transport of CH4 in the soil and snow. The ms simultaneously tries to solve the issues of the spatial /temporal methane dynamics of the large Sibirian wetlands, the process pathways and comparing all the modelling output to relatively few and very local measurements near Chersky. I basically don't think that the available measurements are well suited to verify the model output of the processes leading to the net CH4 emission at the surface, and the differentiation of pathways of CH4 during different periods of the year.

In agreement with reviewer 1, we detailed further our approach for the scaling between the model output at grid cell level and the available observations, especially those from Eddy covariance measurements. These are also added in this response below.

We agree with the reviewer 2, and also highlighted by reviewer 1, that the observational data to validate our model output is few. On the other hand, boreal wetlands, especially those in permafrost regions as in far Northeast Siberia are quite understudied due to the difficulty to reach those places and perform measurements all year round. The data presented in this manuscript shows a synergistic and unique study between the first year-round greenhouse gas emission measurements and summer chamber fluxes in a site of the Kolyma (Chersky) floodplain and a process-based methane model embedded in a land surface model. Site level comparisons are achieved by comparing the two years of continuous Eddy covariance methane flux measurements and summer flux chamber measurements to the model grid cell output. Many fluxes simulated by models in other world regions cannot be adequately evaluated due to the lack of measurements on the study site and have to rely in measurements done in other areas, or from data that has been collected during different periods of time, e.g. only summer measurements. In contrast, our study benefits greatly from the simultaneous temporal and spatial (at grid cell scale possible) synergy where the model development has directly benefited from the year-round greenhouse gas fluxes observations. Finally, is not our aim to been able to evaluate all model grid cells but rather demonstrate that a process-based methane model can achieve in an Arctic tundra region and this is already a scientific contribution per se to the scientific community.

I my opinion the ms could benefit from being divided into two; one with focus on the annual budget for Sibirian and one focused on the process modelling of the different pathways for methane through the soil/snow pack. The later one could benefit from some kind of lab or micro cosmos comparison, where processes could be studied more precisely than what is mostly the case in the field.

We appreciate this suggestion made by the reviewer. The reviewer suggests to have one ms focused solely in the annual methane budget for a larger study region, i.e. Siberia. In fact, we plan to work on such a larger scale (but still high-resolution, process-based) study in the

future. However, at this point our intention is mainly to provide a first proof-of-concept of the applicability of our modeling framework at a still relatively small regional domain, and we believe that this manuscript needs a strong focus on the background description of the used model configuration. Such description has not been published elsewhere. The reviewer also suggests to have such process modeling description supported by e.g. micro cosmos experiments, in a separate ms. We agree that such a study would be an excellent addition to the study we already completed; however, developing micro cosmos experiments is out of the scope of our current project and this step could be done for future investigations in order to refine the current model configuration. Instead of splitting our work into two separate manuscripts, we therefore suggest instead to improve the flow of the current ms: we will shorten the revised ms and move extra useful information to the supplementary material, and clarify further the issues mentioned through this review in the new revised ms.

Regardless of the approach, the issues of differences in scales should be discussed much more detailed and qualified than it is done in the present version of the ms. From my perspective the output of the model and the assessment of the advances in the new "improved" version is not credible as it appears now, despite that the output is in the same ballpark as the measured data, and a number of other studies.

As mentioned earlier, we refined the analysis for the comparison between the methane fluxes from eddy covariance and those from the model, regarding the difference in scales, this also helps to sustain better our results presented in the ms. The difference in scales between EC data and model output is also a comment made by reviewer #1. Our answer to this concern is (same as for Rev. #1):

We agree that the comparison between model methane fluxes and those from observations, specifically from eddy covariance, is a challenge. In our manuscript, we use a scaling factor for the chamber data by considering chamber measurements that were done under exclusively wet and under exclusively dry summer conditions. We then make use of the total fraction of inundated areas in the model grid cell (IF) modeled with the TOPMODEL approach to scale the total chamber fluxes. This scaling approach takes into consideration that the model methane fluxes represent the emissions from only the portion of the grid cell that is inundated, i.e. with water at or above the soils surface.

In the case of the eddy covariance fluxes, following the concerns of the reviewer, we re-evaluated our approach for this comparison. In the revised version of this manuscript we include now a thorough analysis of the footprint area of the eddy covariance fluxes as part of a new Appendix B on "Details on in-situ flux observations". This appendix also includes details on the eddy covariance flux data uncertainty assessment and more detailed results on the chamber measurements, as requested below also by the reviewer. This appendix will be part of the revised manuscript and is attached at the end of this response.

In this new appendix, we analyze the type of vegetation and its coverage in the footprint area of the EC tower, from remote sensing images as a metric to identify wet and dry areas. Areas with dominant cotton grasses, specifically *Eriophorum angustifolium* in our study area, are indicators of predominant wet soils, while tussocks, specifically *Carex appendiculata* in our study area, and shrubs are indicators of predominant dry soil conditions. It is important noting that *C. appendiculata*, can be also found in wet areas, but is predominant in dry areas.

For the model, the vegetation distribution per grid cell is too coarse to consider this metric similar as that for the remote sensing data in the EC footprint area, however the total abundance of C3 grasses in the grid cell A is 33.3 % as given for the model (with the rest of the grid cell dominated by deciduous shrubs and extra tropical evergreen trees), but there is no discrimination between cotton grasses and tussocks.

The footprint of the eddy covariance tower in the Chersky floodplain covers an approximate area of 400 m x 400 m, similar to that one depicted in Fig. 1 of Kittler et al. 2016 (cited in discussion ms) (see new Appendix B at the end of this response for footprint area for the EC tower used in this manuscript). The remote sensing analysis revealed that cotton grasses are present in about 26 % of the footprint area, which would translate into the same portion of the footprint area as fully wet zones during the "wet months": after spring melt in June and until

August when most annual precipitation in the region takes place, covering most of the growing season. As will be shown below in this response, $CH_4$ fluxes measured by chambers (footprint of 60 cm x 60 cm) revealed that during the growing season in dry soil areas of the Chersky floodplain that are characterized by a water table below the surface, the emission of methane during the growing season is negligible with even some atm. $CH_4$ uptake by soil (i.e. negative $CH_4$ flux rates) (data shown in new Appendix B). Under this consideration, and as confirmed recently by Helbig et al., 2017, the majority of the $CH_4$ fluxes measured by the EC tower would represent fluxes from fraction of wetland in the footprint area, i.e. 26 %.

In case of the model grid cell where the location of the EC tower falls (grid cell A in Fig. 1 of the discussion ms), the IF for June-July-August during 2014 shows growing inundation values from 17.7 % to 19.9 % (for 10-day mean values for those three months) representing the percentage of total wet areas in the grid cell area. These values are slightly smaller than the 26 % wetness area in the EC footprint, and denote the area of the grid cell where the model methane emissions take place (i.e., no emissions in dry areas, in agreement to the chamber measurements).

With this basis and to make a closer comparison between EC flux measurements and model data for the growing season months, we scaled linearly the 10-day mean EC methane fluxes to the IF from the model, and calculated the standard deviation of the 10-day mean. In the next figure, we show: TOP panel, the original 10-day mean EC methane flux measurements that would represent the emissions of a 26 % wet area between June and August 2014 (black line), the 10-day mean EC methane fluxes scaled to the 10-day mean IF from the model for the same period of time (red line) and 10-day mean model methane emissions for grid cell A, which imply emissions from the IF from the model (blue line). Error bars in all lines are one standard deviation of the 10-day mean flux values. The BOTTOM panel shows the 10-day mean IF from the model used to scale the EC fluxes (blue line), and the constant wetness percentage of the footprint area calculated from the vegetation coverage remote sensing images (i.e., 26 %).

[Figure]

We observe that the scaled EC methane fluxes decreased as a lower IF is considered within the footprint, and those new calculated fluxes become closer to those from the model, and in most cases the latter fall within the 10-day standard deviation of the EC fluxes.

Unfortunately, it is not possible to obtain a temporal varying wetness area for the EC footprint all year, based on our approach of only considering the vegetation cover, thus wouldn't be appropriate to scale all of the EC fluxes for 2014 and 2015 to the IF from the model without any reference for spring and winter wet footprint areas. However, from this analysis we learn that: 1) considering the vegetation cover as indicator of soil wetness, the EC footprint area holds a very similar area to that of the model grid cell through which the majority of the methane is emitted to the atmosphere and 2) the net offsets between methane flux model and EC data can largely be attributed to differences in wetness levels.

Summarizing, we assume that for both the model grid cell and the eddy covariance footprint, methane emissions are not spatially homogeneous, but bound to the distribution of wet (inundated) areas. Accordingly, a meaningful agreement between model and observations can only be obtained if two factors are fulfilled: (i) the fraction of wet surfaces agrees between both data sets, and (ii) the flux rates from wet surfaces agree between both datasets. Through correcting the offsets in inundated fraction, we could demonstrate that the flux rates between model and eddy covariance observations agree very well, emphasizing the sound setup of the model algorithms and parameter settings. We will add the analysis presented here into the new Appendix to complement the discussion on scaling fluxes for comparison between EC and model data.

**Specific:**

L48 -> 66: Maybe a matter of taste, but I'm in general against using these "horror scenarios" which draw lines between the carbon pool of the Arctic soils and potential increase of GHGs. I think we now know that no indications are found that something very dramatic is happening in foreseeable future, and it doesn't add to the understanding of the ms. Consider rephrasing.

Thank you for this suggestion. We will consider rephrasing these lines. However, we think is still important to mention them, such as that changes in air temperature, soil topography and projected shifts in precipitation in Arctic tundra ecosystems will influence in the future the soil hydrologic regime in permafrost areas which in turn will affect future emissions of CO2 and CH4 into the atmosphere from Arctic terrestrial ecosystems. This projected scenario is part of current literature discussions, which draw the framework of studies like ours presented in this manuscript.

L187: Despite that you are obviously aware of the complications of the comparison between scale I'll encourage you to address specifically how the scaling issue between 0,5_ modelling grid and EC footprint or chambers is dealt with.

We have added the response to this important suggestion above.

L218: Again please justify, why 11 soil layers are needed, when the horizontal scale is this coarse.

The coarse horizontal resolution in the model does not influence the need for a refined vertical discretization of soil processes. In particular, a fine vertical resolution is required to find numerically stable solutions of the gas diffusion equation.

L323: Spun up for 10.000 years? Please justify further, climate (or C – pools) can not be assumed to have remained constant for this period of time.

Thank you for this question. The idea behind the so called spin up approach is to initialize state variables, such as temperature, moisture or carbon content based on the process representation (the differential equations) and environmental conditions during a pre-industrial time when we can neglect a human-induced disturbance of ecosystems and climate. This is important in prognostic modelling in order to reliably isolate effects of anthropogenic actions and related climate change on ecosystems. Usually, soil carbon pools have a mean residence time of less than 1000 years in the aerobic case and hence, this slowest carbon pool will reach a steady state with pre-industrial climate after 1000 years of iteration. For example, for a temperate terrestrial ecosystem we would assume a stable climate over 1000 years round 1700-1800 and substitute the pre-industrial climate by an observation-based climatology from 1901-1930. Climate variations in the past (e.g. little ice age) are usually neglected because future climate change will be much stronger.

Soil organic matter in permafrost regions is additionally stabilized by soil freezing, even in the active layer, i.e. the OM is either frozen over long time periods in permafrost or the decomposition season is reduced to a few months. That is an additional stabilization which leads to much higher effective mean residence times and hence we need to spin-up the model longer to reach the pre-industrial steady state, usually 10000 years are valid (McGuire et al., 2016; Chadburn et al., 2017). In Chadburn et al. (2017) it was shown that such approach leads

to soil organic matter pools comparable to observations at several Arctic stations. At the Cherskii site, we unfortunately do not count with observed carbon stocks. In the following figure however, we show the total carbon (sum of woody, green and reserve) in the soil at the end of this spin up period for the entire model domain and it can be seen that these have reached equilibrium after this period.

[Figure]

L403: I understand that the numbers can be compared, but please argue why the field site measurements can be assumed to be averaging the full 0,5x0,5_ modelling pixel.

In this context, we approach a scaling factor considering the wet and dry areas of the grid cell vs. the wet and dry plots in the chamber sites (model grid cell and chamber plots heterogeneity). This was documented in the discussion manuscript. Briefly, to obtain the total flux from chamber measurements, the values measured from fully wet sites and fully dry sites were scaled to the daily-inundated fractions as given from the model, leading to Eq. 8 of the discussion manuscript. Taking this approach in consideration, the final fluxes from chamber measurements represent the $CH_4$ emissions per $m^2$ per day under heterogeneous (wet and dry) soil conditions, similar to those at grid cell scale.

L445: Differences seems to be substantial please comment.

The differences in the wetland extent from the model compared to those from the high-resolution remote sensing data might seem substantial ($\sim$ 6 %), however, it is important noting the following:

- No other remote sensing data exclusive for this area at such high resolution (150 m) has been available for the Boreal Arctic region such as that one used for our study (Reschke et al., 2012; cited in discussion ms). Other available wetland extent remote sensing products are only at global scale with spatial resolutions in the order of 25 km. By having a better-resolved reference wetland extent, ensures that the uncertainties in the model wetland extent are limited mostly to the model technique and spatial resolution used to simulate wetlands. In addition, high latitude wetlands still pose a challenge in remote sensing, due to the long periods of darkness during the year.
- The addition of the TOPMODEL scheme in the land surface model JSBACH allows the representation of standing water following the topographic profile of the region of interest. Without this scheme, it is not possible at all with this model to allow accumulation of water at the soil surface. Furthermore, this is not an exclusive characteristic of the JSBACH model, as most land surface models lack of an explicit and fully functional hydrology model where also the dynamics of inland waters, such as runoff, are possible to be simulated. From the modeling point of view and within the inherent limitations of the model structure, the possibility of simulating wetland extent in a land surface model and having a remote sensing data with sufficient resolution is an excellent combination of first steps achievements to improve process-based modeling for greenhouse gases at high latitudes. For this reasons, we argue that a 6 % difference between the remote sensing wetland data and the modeled data provides a first good approximation.

L470: I basically don't understand how a threshold can be set for proportion of flooded area in a pixel – what is the rational? Theoretically the whole pixel could be inundated –I assume?

The proportion of flooded area in a pixel follows exclusively the topographic profile and the location of the water table, in a way that if there is enough water content in the soil (basically close to saturation) it will be possible to accumulate water at the surface and only if the topography structure allows it. For this reason, not all the pixel could be inundated. The sequence of the TOPMODEL scheme in our model configuration follows the next steps: 1) selection of soil layer where the water table will be positioned according to the soil water content, 2) by defining a water table threshold, the model locates the position of the water table, 3) a fixed TOPMODEL parameter defines the dependence of flooding on the water table variation in a way that the lower the value of this parameter means that larger areas with the same water table will be flooded (parameter $f$ of equation 1, exponential decay of transmissivity with depth, in the discussion manuscript), 4) a fixed threshold in the TOPMODEL scheme limits the area of floodability ($\chi_{min\_cti}$, given in L242-245 in the discussion ms), this is used to avoid the occurrence of running water and is dependent on soil types. This TOPMODEL parameter is used in the general TOPMODEL scheme to allow runoff, which in our model configuration should not be taken into account. The lower the value the larger the flooded area e.g., limits of the horizontal extent of the inundated area. We find this level of detail on the model configuration can only be added in the supplementary material of our manuscript and we will consider doing so.

L532: What effect of the snow would you have expected in this context?

As shown in the sensitivities exercise, by describing thinner snow layers (3 and 1 cm) than the value in the control simulation (5 cm) allows only in a temporal shift of the emissions without affecting the magnitude of the total annual $CH_4$ emissions. The intention of doing this test is to analyze the response of the model exclusively to this parameter and our hypothesis (thinner snow layers allows faster diffusion of gas than thicker layers with constant density values) has been confirmed.

L630: there seems to be significant difference in measured and modelled soil temperatures, please comment.

The effect of having higher soil moisture in the soil pores influences the soil thermal regime in organic-rich soils both during summer and winter. When the soil pores are predominantly filled with water, the water promotes a high thermal capacity, and when pores are predominantly filled with air, the thermal soil capacity decreases and more energy is required to heat the soil. Also, near-surface vegetation in these tundra environments, such as mosses and lichens (Porada et al., 2016) plays an important role as effective thermal insulator but also would help to insulate the surface soil layers from the warm surface temperatures from atmospheric influence during summer. As well, snow cover serves as thermal insulator, and a further snow layer evaluation from the model and measurements in the site needs to be done as measurements become available.

Besides the need to consider the previous factors, our results evidence the effect of the soil moisture variation, which in general is quite low, to the soil thermal regime. The soil hydrology, as mentioned extensively in the manuscript, poses still limitations in our current model configuration and it requires further improvements, also hopefully based, on available soil moisture measurements in this study region, which at the moment are unavailable.

L665: probably why also both absolute values and seasonal pattern seems distinctly different

It is unclear to us what this comment from the reviewer is referring to. We ask for a further clarification in a way that a suitable response can be given from our side.

L710 -723: that differentiation between ebullition and diffusion seems unfounded, and it is hard to see how you verify the different pathways, please elaborate.

In our methane module, emissions of methane via ebullition and diffusion are explicitly modeled and are based in fundamental principles of gas motion. Diffusion of a gas is a molecular motion process and its speed relies on the medium where it takes place: it is slow in water and faster in air. It works independently of a water table with the net movement of molecules following a concentration gradient from high to low concentrations in order to achieve equilibrium. In the case of ebullition, this takes place when a certain volume of water gets saturated with a specific gas and oversaturation allows the formation of bubbles that, due to pressure effects, are released through available pathways, such as interstitial water in the soil. While diffusion is a continuous but rather slow molecular process, ebullition is fast and highly sporadic. These are well known physical processes in gas dynamics, and an excellent review on the explicit diffusion and ebullition processes for methane in soils is provided in Le Mer and Roger (2001), Eur. J. Soil Biol. Vol. 37, doi: 10.1016/S1164-5563(01)01067-6.

Para 3.4.3: could this be merged with the sensitivity study in 3.2? seems to be fundamentally alike.

This recommendation by the reviewer is unclear. Section 3.2 is about presenting the results of the sensitivity experiments, while section 3.4.3 contain results on the environmental controls related to the methane fluxes and their temporal variation. Thus, these sections are not alike and therefore cannot be merged as suggested. Perhaps there was some confusion in the number of sections that the reviewer is referring. We ask for a clarification on this comment to better assess a response.

Fig. S5b: legend does not seem to match.

Ok, panels a and b were inverted and this is now corrected. Thank you for identifying this mistake.

L916-920: the conclusions here seem somewhat unfounded due to the previously mentioned scaling issues.

The lines the reviewer here is referring to are part of the discussion and not of the conclusion section (starting in L1098 of discussion ms). Related to the lines referred here from the discussion, we wrote: "We simulated for the first time year-round methane emissions in a Northeast Siberian region centered on the city of Chersky, including emissions during the non-growing season. Our results showcase the ability of the improved JSBACH-methane model to reproduce seasonality in the $CH_4$ emissions when compared to fluxes measured by eddy covariance and chambers in a study site near Chersky."

In these lines of the discussion, we refer also explicitly to the ability of the model to reproduce the seasonality of the methane emissions (lower in winter, higher in summer months) independent of their magnitude, which we presented and discussed accordingly in the discussion ms. Regarding the scaling between EC measurements and model results for the comparison of their magnitude, we hope that with the clarification above this argument is adequately answered.

References.

Chadburn, S.E., et al., (2017), Carbon stocks and fluxes in the high latitudes: using site-level data to evaluate Earth system models, Biogeosciences, 14, pag. 5143-5169, doi:10.5194/bg-14-5143-2017.

Helbig, M., Quinton, W.L., Sonnentag O., (2017) Warmer spring conditions increase annual methane emissions from a boreal peat landscape with sporadic permafrost. Environmental Research Letters, 12, 115009, doi: 10.1088/1748-9326/aa8c85.

McGuire, A.D. et al., (2016), Variability in the sensitivity among model simulations of permafrost and carbon dynamics in the permafrost region between 1960 and 2009. Global Biogeochemical Cycles, 30(7), pag. 1015-1037, doi:10.1002/2016GB005405.

Porada, P., A. Ekici and C. Beer (2016), Effects of bryophyte and lichen cover on permafrost soil temperature at large scale, The Cryosphere, 10, 2291-2315, 10.5194/tc-10-2291-2016.

**Appendix B: Details on in-situ flux observation program**

*Eddy-covariance flux data uncertainty assessment*

Following well-established procedures in literature (Aubinet et al., 2012), our uncertainty analysis for the eddy-covariance flux data has been split up into random and systematic errors. The major sources for random errors, associated with the turbulent sampling and instrument issues, have been quantified for each 30 min flux value through the flux processing software TK3 (Mauder and Foken, 2015). Errors related to footprint uncertainties were not quantified, since there are no major transitions in biome types within the core areas of the flux footprints.

Systematic errors can be introduced by unmet theoretical assumptions and methodological challenges, as well as by instrument calibration and data processing issues. In the context of the Chersky observations, instruments were maintained and calibrated in regular intervals, therefore minimizing this potential error component. Moreover, data intercomparisons with a second eddy-covariance tower close by (~ 600 m) yielded no systematic offset in the frequency distributions of wind speed, sonic temperature, and methane mixing ratios between towers. Regarding flux data processing, the TK3 software package contains all the required processing steps, conversions and corrections for the flux data processing, and yielded good agreement in a recent comparison with EddyPro (Fratini and Mauder, 2014). To avoid methodological issues that may bias flux data results, we employed a rigid post-processing quality control and flagging system scheme, with the well-established analyses for stationarity and well-developed turbulence originally proposed by Foken and Wichura (1996) at its core, supplemented by additional tests (absolute range and spikes) to flag implausible data points in the resulting flux time series. Based on the quality assessment and control tools outlined above, we excluded systematic errors from the uncertainty quantification of flux data that were assigned a high to medium data quality (QF 1-6 based on the scheme proposed by Foken et al., 2005; 2012) and subsequently used for assessing long-term $CH_4$ flux budgets.

No u*-threshold was applied to the flux dataset, since we determined stationarity of the signal and integral turbulence characteristics also for nighttime conditions. This information facilitates identifying datasets with regular turbulent exchange also during stable stratification, therefore producing fewer gaps compared to a bulk exclusion of data during stable nighttime stratification through the u*-filter method. After filtering out low-quality fluxes, the data coverage of methane fluxes was 86 % during the growing season and 67 % during the winter from the original full 30 min flux data set (Kittler et al., 2017). To produce a continuous flux record for quantification of long-term $CH_4$ budgets, we filled the remaining gaps by averaging existing flux data within a moving window of 10-day length centered on the gap. Uncertainties for gap-filled values were quantified as standard deviation within the corresponding window, similar to the definition of gapfilling uncertainties for the $CO_2$

flux via the well-established marginal distribution sampling routine by Reichstein et al. (2005).

To produce aggregated uncertainty values for longer time periods, we applied the procedures suggested by Rannik et al. (2016). All random errors were combined by considering them as independent variables, and normally decrease with the length of the averaging period. Averaged over both data years used within the context of this study (2014 and 2015), the $CH_4$ flux uncertainty based on the 30 min data is $7.4\pm8.3$ nmol m$^{-2}$ s$^{-1}$, a result comparable to $4.7\pm3.8$ nmol m$^{-2}$ s$^{-1}$ reported for a fen ecosystem by Jammet et al. (2017).

*Source weight function of the eddy-covariance flux data*

We conducted a source weight analysis, also called footprint analysis, to determine the fractional contribution of different land cover types within the field of view of the eddy-covariance flux tower. Source weight functions for each 30min flux measurement were computed based on the Lagrangian Stochastic footprint model by Rannik et al. (2003). Footprints were accumulated, analyzed and interpreted using an approach presented by Göckede et al. (2006; 2008). We projected these footprints onto a WorldView2 land cover map at 2 m horizontal resolution (see also Figure A1). In the context of the presented study, we aggregated the original 22 land cover classes into 9 classes to concentrate on the dominant elements of the vegetation community structure (see also Table A1).

[Figure]

**Figure A1: Accumulated source weight function for the control tower within the Chersky study site, based on data from the growing season (mid June – mid September) in 2014. Solid white isolines indicate the 80, 60, 40, and 20% levels, while the dashed line gives the 10% level. Background colors give aggregated land cover classes based on WorldView2 data.**

Since the tower is situated on a slightly elevated patch of tundra, tussocks and shrubs featuring various levels of wetness (red and orange colors in Fig. A1) dominate the immediate surroundings. Even though inundated parts of the study area, in this case identified by the prevalence of the cotton grass *Eriophorum angustifolium* (blue-ish colors in Fig. A1), are dominating the area encircled by the 10% isoline that is used here to mark the boundary of the cumulative footprint area, they are mostly present in the outer reaches, therefore combining just about 26% of the total flux signal sampled by the eddy system. Another 31% is contributed by wet to moist tussock tundra with some shrubs. Overall coverage fractions within the major wetness categories (see also Table A1) remain approximately constant between tower footprint and two larger

regions covered by the same WorldView dataset, indicating that this composition of wetness levels is typical for the Kolyma floodplain ecosystems analyzed within the context of this study.

**Table A1: Fractional coverage of aggregated WorldView land cover classes within the control tower footprint of the Chersky study site. Background color coding was used to categorize the classes into wetness levels. The rightmost two columns give fractional coverage of these classes within the area immediately surrounding the towers (1.2 x 1.2 km) and within the entire WorldView scene analyzed (5 x 5 km).**

| Land cover class | category | tower footprint | 1.2x1.2km | 5x5km |
|---|---|---|---|---|
| water | open water | 0.001 | 0.035 | 0.134 |
| cotton grass, wet continuously | | 0.111 | 0.043 | 0.053 |
| cotton grass, partially dry | wetland | 0.067 | 0.153 | 0.147 |
| cotton grass with tussocks | | 0.081 | 0.063 | 0.038 |
| tussocks with some shrubs | wet to moist | 0.312 | 0.418 | 0.280 |
| tussocks with higher shrubs | | 0.388 | 0.165 | 0.182 |
| higher shrubs, with tussocks | moist to dry | 0.031 | 0.097 | 0.115 |
| trees | | 0.001 | 0.006 | 0.017 |
| undefined | | 0.008 | 0.020 | 0.035 |

*Flux chamber observations*

The Chersky study site features two transects of 10 permanently installed PVC collars for flux chamber measurements. With distances of approximately 25 m between individual microsites, both transects cover a distance of ~225 m within the drained and control sections, of this permafrost site. Site locations were selected quasi-randomly to reflect the dominant microsite characteristics (e.g. vegetation composition, wetness level) found at each of the target locations. With a chamber footprint of 60 cm x 60 cm, this technique allowed studying microsites with rather homogeneous environmental conditions, as compared to the eddy-covariance fluxes with often heterogeneous footprint areas. Details on the chamber program, overall methane flux rates observed, and functional relationships with e.g. soil temperature, vegetation and wetness levels, are provided by Kwon et al. (2016; 2017).

[Figure]

**Figure A2: Daily methane flux rates aggregated from flux chamber measurements within the growing season of 2014. Measurements are separated into drained (1 wet microsite, 9 dry microsites) and control (8 wet microsites, 2 dry microsites) transects.**

Figure A2 displays average flux rates for wet and dry microsites observed within the drained and control transects during sampling campaigns in summer 2014. These flux chamber results clearly demonstrate that methane release rates were virtually zero in the absence of standing water. At some of the dry microsites (results not shown), even negative $CH_4$ flux rates were observed, indicating the oxidation of methane under highly aerobic conditions within these predominantly wet tussock tundra ecosystems in Northeastern Siberia.

**Cited literature**

Aubinet, M., T. Vesala, and D. Papale (Eds.) (2012), Eddy Covariance - A practical guide to measurement and data analysis, 438 pp., Springer, The Netherlands.

Foken, T., R. Leuning, S. P. Oncley, M. Mauder, and M. Aubinet (2012), Corrections and data quality, in Eddy Covariance - A practical guide to measurement and data analysis, edited by M. Aubinet, T. Vesala and D. Papale, pp. 85-131, Springer, Dordrecht; Heidelberg; London; New York.

Foken, T., M. Göckede, M. Mauder, L. Mahrt, B. Amiro, and W. Munger (2005), Post-Field Data Quality Control, in Handbook of Micrometeorology: A Guide for Surface Flux Measurement and Analysis, edited by X. Lee, W. Massman and B. Law, pp. 181-208, Springer Netherlands, Dordrecht, doi:10.1007/1-4020-2265-4_9.

Foken, T., and B. Wichura (1996), Tools for quality assessment of surface-based flux measurements, Agr. Forest Meteorol., 78(1-2), 83-105, doi:10.1016/0168-1923(95)02248-1.

Fratini, G., and M. Mauder (2014), Towards a consistent eddy-covariance processing: an intercomparison of EddyPro and TK3, Atmos Meas Tech, 7(7), 2273-2281, doi:10.5194/amt-7-2273-2014.

Göckede, M., et al. (2008), Quality control of CarboEurope flux data - Part 1: Coupling footprint analyses with flux data quality assessment to evaluate sites in forest ecosystems, Biogeosciences, 5(2), 433-450.

Göckede, M., T. Markkanen, C. B. Hasager, and T. Foken (2006), Update of a footprint-based approach for the characterisation of complex measurement sites, Bound.-Lay. Meteorol., 118(3), 635-655, doi:10.1007/s10546-005-6435-3.

Jammet, M., S. Dengel, E. Kettner, F. J. W. Parmentier, M. Wik, P. Crill, and T. Friborg (2017), Year-round CH4 and CO2 flux dynamics in two contrasting freshwater ecosystems of the subarctic, Biogeosciences, 14(22), 5189-5216, doi:10.5194/bg-14-5189-2017.

Kittler, F., M. Heimann, O. Kolle, N. Zimov, S. Zimov, and M. Göckede (2017), Long-term drainage reduces CO2 uptake and CH4 emissions in a Siberian permafrost ecosystem, Glob. Biogeochem. Cy., (online first), doi:10.1002/2017GB005774.

Kwon, M. J., M. Heimann, O. Kolle, K. A. Luus, E. A. G. Schuur, N. Zimov, S. A. Zimov, and M. Göckede (2016), Long-term drainage reduces CO2 uptake and increases CO2 emission on a Siberian floodplain due to shifts in vegetation community and soil thermal characteristics, Biogeosciences, 13(14), 4219-4235, doi:10.5194/bg-13-4219-2016.

Kwon, M. J., et al. (2017), Drainage enhances surface soil decomposition but stabilizes old carbon pools in tundra ecosystems Nat. Clim. Change, (submitted).

Mauder, M., and T. Foken (2015), Documentation and Instruction Manual of the Eddy-Covariance Software Package TK3Rep., University of Bayreuth, Bayreuth.

Rannik, U., T. Markkanen, J. Raittila, P. Hari, and T. Vesala (2003), Turbulence statistics inside and over forest: Influence on footprint prediction, Bound.-Lay. Meteorol., 109(2), 163-189, doi:10.1023/a:1025404923169.

Rannik, Ü., O. Peltola, and I. Mammarella (2016), Random uncertainties of flux measurements by the eddy covariance technique, Atmos. Meas. Tech., 9(10), 5163-5181, doi:10.5194/amt-9-5163-2016.

Reichstein, M., et al. (2005), On the separation of net ecosystem exchange into assimilation and ecosystem respiration: review and improved algorithm, Glob Change Biol, 11(9), 1424-1439, doi:10.1111/j.1365-2486.2005.001002.x.

---

## Author Response (AR2)

**Minor comments of final report for manuscript bg-2017-310**

The authors basically addressed two referees' comments and revised manuscript accordingly. But I still have some comments below:

1) The author needs to carefully cite other studies. P2, 50: In Saunois et al. (2016), 4% is the ratio of the total emissions in the north of 60 °N to the global total emission from top-down inversions. P3, 80: Strictly, the cold season, instead of late autumn, CH4 emissions account for up to 50% of the annual emission, according to Zona et al. (2016).
Thank you for pointing this out. This was as suggested, rephrased accordingly.

2) P2, 50: The description of 'soil topography' is uncommon.
We removed the word soil, and replaced it with "surface topography". Here we refer to changes in topography e.g. related to the formation of polygons due to the formation of ice wedges.

3) In my understanding, the methane production, oxidation and emissions were simulated for the saturated portion of each grid cell (P8, 275). But when the author calculated the chamber measurement, the emissions are from both from wet soil and dry soil (p13, 430). The author may explain. What is the portion of the emission from dry soil?
The scaling approach used for the methane fluxes measured with chambers following Eq. 1, was done to account for the emissions from exclusively wet soils, and to keep the agreement with the emissions from exclusively water near-saturated soils in the model. This step was necessary because, as was presented in the Appendix A (section *Flux chamber observations* and in now Fig. A3), the methane emissions measured with the chambers in dry soils were negligible. Thus, the scaling approach for the chamber measurements serves to avoid considering the measured fluxes from a 100 % wet area, but rather from a portion of wet soils located in an heterogeneous dry and wet total area covered by all of the chambers, finding an equivalent to the also heterogeneous grid cell area. We added a sentence on this regard when explaining this scaling approach, also making reference to the Appendix A. This paragraph now reads:

"Thus, the total $F_{ch}$ from the chamber plots was averaged separately for the wet plots ($F_{ch\_wet}$) and for the dry plots ($F_{ch\_dry}$). This heterogeneity in the data finds its equivalent in the model grid cell heterogeneity as estimated by TOPMODEL, where on average only a portion of the grid cell area is inundated and the rest remains dry during a specific period of time. The modeled methane emissions correspond exclusively to the portion of the grid cell with near-water saturated soils. Similarly, the chamber flux measurements evidenced predominant emissions of this gas in plots with wet soils, whereas the emissions in dry plots were negligible. Thus, to obtain the total $F_{ch}$ the chamber flux measurements, and to account for emissions predominantly from wet soils, $F_{ch\_wet}$ and $F_{ch\_dry}$ were scaled to the daily-inundated fractions $w_{mod}$ for the corresponding model grid cell A. More details are presented in Appendix A."

4) The author addressed the footprint comment by reviewer #1 in 455-460. This inundated fraction between 17.7%-19.9% will result in a much lower EC CH4 contributed by saturated area. However, in the CH4 comparison, the author did not apply these fractions (Figure 4 and 5). How much will this affect the model-EC bias?

We calculated the change in EC methane fluxes after the scaling exercise presented in Appendix A and considering the smaller fraction of wet areas in the model grid cell vs. the larger wet area identified in the EC footprint through the coverage of cotton grasses. At the end of section *Source-weight function of the EC flux data and scaling for model flux evaluation* in Appendix A. We added the following paragraph (accompanied by Fig. A2, which was presented in the response to Rev. #1):

"Furthermore, C3 grasses cover 33.3 % of the model grid cell A, whereas the inundated fraction in that same grid cell ranges between 17.7 % and 19.9 %, calculated as a 10-day mean during June, July and August 2014. Thus, to improve the comparison between EC and model $CH_4$ fluxes, we corrected the 10-day EC mean fluxes (related to 26 % of the footprint wet cotton grass area), through a linear scaling to obtain the fluxes from a smaller wet area i.e. corresponding to the 10-dat mean inundated fraction of the model grid cell. The results of this scaling exercise are shown in Fig. A2. The non-scaled 10-day EC fluxes for the period of analysis are on average 65 mg $CH_4$ $m^{-2}$ $d^{-1}$, and the scaled fluxes decreased on average by 24 % (mean 49.6 mg $CH_4$ $m^{-2}$ $d^{-1}$) after a correction by considering a smaller wet area within the footprint, reaching a magnitude similar to the 10-day mean fluxes from the model (48.6 mg $CH_4$ $m^{-2}$ $d^{-1}$). This exercise emphasizes that wetness is the dominant control for total methane emissions in these ecosystems."

5) Usually, the process-based CH4 modeling is constrained by observations globally or regionally. Even so, we can still see the large bias between observation and simulation in the local. Under this local condition, the biases of the inundation fraction and total CH4 emissions are remarkable. It is hard to convince the readers about the portion of CH4 emission via different pathways, as CH4 transport pathways are even weakly constrained. The author may not highlight the fraction of each pathway emission.
We agree that it is difficult to quantify the portion of $CH_4$ emissions via different pathways given the scale of the area of study and uncertainty in the fluxes associated with evaluating net methane emissions at the local level. Still, the authors provide the fraction of the total emissions that corresponds to each pathway as implemented in the process-based $CH_4$ model of our study. This explicit representation is necessary in such model, particularly for forecasts regarding Arctic methane emissions under future climate scenarios. Also, some observational data for specific pathways can be found in literature and can be used as a robust reference for evaluation purposes, but this is not sufficient, and more efforts should be undertaken to collect new datasets that allow validating these individual processes.
To implement the concerns of the reviewer, two paragraphs in the Conclusion section were modified and completed in the following way:

"The seasonal transition of the four $CH_4$ transport pathways is mainly controlled by changes in the soil temperature, and only indirectly linked to soil moisture. The majority of the annual emissions take place through vascular plants. Given the relatively large scale of the model regional domain and uncertainties in the methane fluxes associated with forcing the model with reanalysis data and evaluating net emissions at the local level, it is difficult to quantify the emissions through the individual emission pathways. However, this explicit representation is necessary in process-based modeling, particularly for forecast regarding Arctic methane emissions under future climate scenarios.
The findings of this study demonstrate that to improve the understanding of the interannual variability of $CH_4$ fluxes form wetlands in boreal permafrost areas, to improve process model evaluation, and contribution of the individual emission pathways, more highly resolved

temporal observational data is required, specially of year-round CH$_4$ EC fluxes and soil temperatures which are generally scarce and challenging for boreal and tundra areas."

6) In this study, there are not different sites to represent the spatial heterogeneity to match climate reanalysis at a half-degree resolution. As we know, the climate reanalysis is one of the sources to the prediction uncertainty. Is it possible to try single-point simulation by using the climate forcing at the EC site?

We understand the concern from the reviewer and the potential added uncertainty to the modeled methane emissions due to climate reanalysis data. In principle, it is possible to use site level simulations with the climate at the EC site. This exercise was demonstrated a site level in Lena Delta in the paper by Kaiser et al., 2017 (GMD). We therefore believe that adding such experiment in this manuscript falls out of the scope where the main aim is to demonstrate the application of this model at a regional scale. However, to address the concerns of the reviewer, we analyzed the site level climate data available from the EC tower vs. the corresponding fields, from the reanalysis data used to drive our model simulations, for the model grid cell where the EC tower is located. The correlation between the forcing fields and the daily means are shown in the figure below for years 2013, 2014 and 2015. A figure of these comparisons is shown below. As can be observed, a very good agreement is generally observed between the fields in the two data sets, except in the most uncertain fields: precipitation and wind speed. The former is known to contain large uncertainties in both measurements and models, especially when large shares of the annual budget of total precipitation fall in the form of snow. Insufficient heating of devices in the Eddy Covariance system, also can lead to low bias in precipitation measurements. As for wind speed, the disagreement between the reanalysis data and the measurements done with the anemometer in the Eddy tower can explained mainly due to the different altitudes at which measurements are performed: while the reanalysis data is a scaled at 10 m reference height, the wind speed from the Eddy tower is measured at 4.7 m height. Despite these disagreements, the comparison between fields in the driving climate for the model and the observations done by the Eddy tower are in close agreement, thus the uncertainty introduced in the simulated methane emissions due to the forcing data do not pose a considerable contribution, at least for the grid cell where the EC measurements were done. In the paragraph of Conclusions modified above, we also add the potential added uncertainties due to the reanalysis data.

[revised manuscript text omitted]